# Complex regulatory networks influence pluripotent cell state transitions in human iPSCs

Timothy D. Arthur[1,2], Jennifer P. Nguyen[2,3], Agnieszka D'Antonio-Chronowska[4], Hiroko Matsui[5], Nayara S. Silva[6], Isaac N. Joshua[5], iPSCORE Consortium*, André D. Luchessi[6,7], William W. Young Greenwald[3], Matteo D'Antonio[2,5], Martin F. Pera [8] & Kelly A. Frazer [4,5] ✉

Stem cells exist in vitro in a spectrum of interconvertible pluripotent states. Analyzing hundreds of hiPSCs derived from different individuals, we show the proportions of these pluripotent states vary considerably across lines. We discover 13 gene network modules (GNMs) and 13 regulatory network modules (RNMs), which are highly correlated with each other suggesting that the coordinated co-accessibility of regulatory elements in the RNMs likely underlie the coordinated expression of genes in the GNMs. Epigenetic analyses reveal that regulatory networks underlying self-renewal and pluripotency are more complex than previously realized. Genetic analyses identify thousands of regulatory variants that overlapped predicted transcription factor binding sites and are associated with chromatin accessibility in the hiPSCs. We show that the master regulator of pluripotency, the NANOG-OCT4 Complex, and its associated network are significantly enriched for regulatory variants with large effects, suggesting that they play a role in the varying cellular proportions of pluripotency states between hiPSCs. Our work bins tens of thousands of regulatory elements in hiPSCs into discrete regulatory networks, shows that pluripotency and self-renewal processes have a surprising level of regulatory complexity, and suggests that genetic factors may contribute to cell state transitions in human iPSC lines.

Characterizing the molecular mechanisms underlying self-renewal and pluripotency cell state transitions (i.e., molecular differences between closely related pluripotency states) is essential for advancing our understanding of development and disease processes. Human pluripotent stem cells (hiPSCs) exhibit a spectrum of interconvertible pluripotent cell states that have different transcriptional and epigenetic profiles and hence present an excellent model to study how regulatory networks govern these biological processes[1,2]. However, the molecular characterization of self-renewal and pluripotency cell states in hiPSCs has been impeded because most studies examine a limited number of lines, and each line is composed of subpopulations in different pluripotency cell states[1,3]. Therefore, conducting gene

[1]Biomedical Sciences Graduate Program, University of California, San Diego, La Jolla, CA 92093, USA. [2]Division of Biomedical Informatics, University of California, San Diego, La Jolla, CA 92093, USA. [3]Bioinformatics and Systems Biology Graduate Program, University of California, San Diego, La Jolla, CA 92093, USA. [4]Department of Pediatrics, University of California San Diego, La Jolla, CA 92093, USA. [5]Institute of Genomic Medicine, University of California San Diego, 9500 Gilman Dr, La Jolla, CA 92093, USA. [6]Northeast Biotechnology Network (RENORBIO), Graduate Program in Biotechnology, Federal University of Rio Grande do Norte, Natal, Brazil. [7]Department of Clinical and Toxicological Analysis, Federal University of Rio Grande do Norte, Natal, Brazil. [8]The Jackson Laboratory, Bar Harbor, ME 04609, USA. *A list of authors and their affiliations appears at the end of the paper. ✉e-mail: kafrazer@health.ucsd.edu

co-expression and regulatory element co-accessibility network analyses across hundreds of hiPSC lines could provide important biological insights into self-renewal and pluripotency cell state transitions including how genetic background contributes to variability in the molecular mechanisms underlying their regulation.

While it is clear that under conventional culturing conditions, each hiPSC line is composed of subpopulations of cells in different pluripotency states, how these proportions vary from cell line to cell line has not yet been investigated in depth. Most cells in a conventional hiPSC culture resemble late post-implantation epiblast stem cells, referred to as 'primed' pluripotency (Supplementary Fig. 1). Primed pluripotent cells are poised for rapid transition from a bivalent to an active state of lineage-specific genes. Additionally, an intermediate cellular state, referred to as 'formative' pluripotency, exists as a subpopulation in conventional hiPSC cultures[1,4]. Formative pluripotency represents the early post-implantation epiblast and is characterized by an enrichment of self-renewal processes, an absence of lineage priming, and the capacity for direct conversion into somatic or germline lineages[1]. Under specific culture conditions, hiPSCs can be maintained in the 'naïve' state resembling the preimplantation epiblast[5], or a totipotent state resembling the even earlier developmental 8-cell morula stage[3]. Understanding how pluripotent subpopulations vary across hiPSC lines under conventional culturing could improve their utility for studying developmental processes and regenerative medicine.

Gene co-expression network analysis is a powerful method that has been widely used to analyze RNA-seq data to identify modules of co-expressed genes that are members of the same biological pathways[6–8]. But thus far, applications of these algorithms to analyze ATAC-seq data have been more limited. For example, studies have used paired bulk ATAC-seq and RNA-seq to examine how regulatory elements impact gene expression networks under dynamic conditions, including comparing cells at baseline and post-stimulation[9] or undergoing differentiation[10]. There has also been considerable effort studying patterns of co-accessibility in single-cell ATAC-seq under static conditions to explore local *cis* interactions (~500 kb) between regulatory elements both in single tissues[11] and simultaneously across multiple tissues[12]; and a recent study analyzed paired single-cell ATAC-seq and RNA-seq data to study local *cis*-regulation of gene expression under environmental stimuli[13]. However, genes that are members of the same biological pathways are frequently encoded on different chromosomes, and there have been limited studies aimed at examining co-accessible ATAC-seq peaks distributed across the human genome to understand regulatory processes underlying the co-expression of gene modules. For example, in hiPSCs, the maintenance of pluripotency relies on the expression of pluripotency-related transcription factors, such as NANOG, OCT4, and SOX2, which create global epigenomic regulatory networks that enable self-renewal through the repression of developmental genes[14], regulation of cell cycle transitions[1,15], and promotion of autoregulation[16,17]. The ability to bin the precise locations and epigenetic profiles of all regulatory elements distributed across the genome into discrete regulatory networks in hiPSCs would provide a valuable resource for further investigations of these important cellular processes.

In this study, we sought to determine if regulatory network modules in hiPSC lines could be identified by examining genome-wide chromatin co-accessibility across samples from hundreds of individuals in the iPSCORE cohort[18]. We hypothesized that these regulatory network modules could be identified due to heterogeneity arising from variation in the proportions of pluripotent subpopulations (Fig. 1A, Supplementary Fig. 1) across hiPSC lines, and would underlie the coordinated expression of gene network modules. We generated 150 ATAC-seq samples from 143 hiPSC lines in iPSCORE and applied the unsupervised Leiden community detection algorithm which enabled us to discover genome-wide regulatory network modules

(RNMs) comprised of co-accessible regulatory elements. We integrated these data with gene network modules (GNMs) that we similarly generated from the RNA-seq dataset[19,20] for 213 hiPSC lines also in iPSCORE. We demonstrated that both the RNMs and GNMs were associated with the differential expression of marker genes defining hiPSC pluripotency cell states, and showed that their discovery is due to differences in the proportion of these transitory pluripotency states across the hiPSC samples. The analyses of these datasets coupled with whole-genome sequencing data enabled us to identify and functionally characterize the TF binding and chromatin state profiles of elements in high-resolution hiPSC regulatory networks associated with distinct pluripotency states; and characterize mechanisms by which regulatory variants affect TF binding and regulatory networks.

## Results
### Relative proportions of pluripotent subpopulations vary across 213 hiPSC lines
Previous studies have demonstrated that human induced pluripotent stem cell (hiPSC) lines are composed of subpopulations of interconvertible pluripotent cell states (Fig. 1A, Supplementary Fig. 1)[1,4]. We set out to determine if the relative proportion of these cell states varied across 213 hiPSC lines with bulk RNA-seq data (Supplementary Data 1, Supplementary Data 2) using cellular deconvolution[21]. Deconvolution typically uses gene signatures obtained from cell type clusters in single-cell data; however, due to the high similarity between hiPSC pluripotency states, they don't separate into distinct clusters. Therefore, we generated gene signatures using bulk RNA-seq data for FACS-sorted formative and paired unsorted (e.g. primed) cells[1]. We applied the CIBERSORT deconvolution algorithm with a signature matrix containing the 300 most differentially expressed genes to obtain estimates for the relative proportions of cells in the formative and primed states in the 213 hiPSC lines (Supplementary Fig. 2A, Supplementary Data 3). We examined the expression of formative-specific and primed-specific genes in hiPSCs with the highest ($n = 15$) and lowest ($n = 69$) estimated proportions of formative cells and observed notable expression differences of key regulators of pluripotency, such as *DUSP5*, *LEFTY1*, *FZD5*, *FST*, and others (Fig. 1B–E, Supplementary Fig. 2B, C). These data indicate that cellular deconvolution can efficiently rank hiPSC lines by their relative fraction of the formative cell state. Taken together, our results show that remarkable variation in pluripotency cell state composition exists across the 213 hiPSC lines under conventional culture conditions.

### Identification of gene networks associated with pluripotency state
Pluripotent cell states have traditionally been defined by the expression of genes underlying self-renewal, pluripotency, and cell cycle regulation[16,22,23]. To identify co-expressed gene modules for these key biological processes, we analyzed the 213 hiPSC RNA-seq samples by calculating the pairwise correlation between 16,110 expressed autosomal genes (TPM ≥ 1 in at least 20% of samples) using a linear mixed model (LMM)[18,20,24] resulting in the identification of 3,533,609 co-expressed gene pairs (adjusted *P*-value < 0.05) with positive associations. We created a global gene network (GN) using these gene pairs as edges and identified gene co-expression network modules (GNMs) by applying the unsupervised Leiden community detection algorithm[25,26] to the GN (Supplementary Note 1 shows our approach for detecting gene modules works better for the iPSCORE cohort, which contains related individuals, than the commonly used WGCNA[6] approach; Supplementary Fig. 3, Supplementary Fig. 4). In total, we identified 13 gene co-expression network modules (GNMs) consisting of between 118 to 1854 genes (Supplementary Data 2, Supplementary Data 4). Biological networks are scale-free[27] and follow the Pareto Principle[28], which states that 20% of the nodes are responsible for 80% of a

 

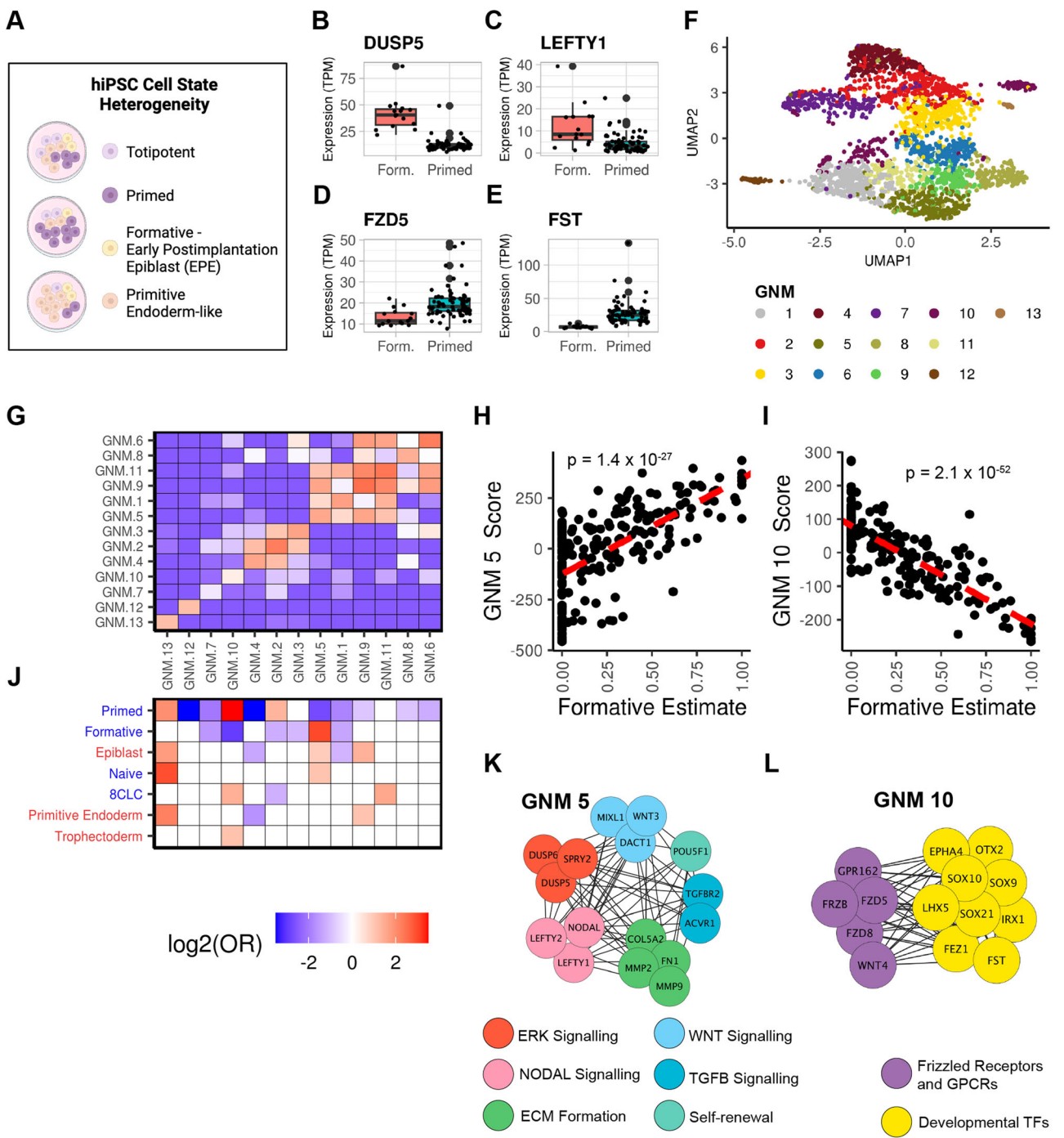

network's connectivity. Therefore, we examined the expression of 2964 GNM-specific Pareto genes (top 20 percentile of intra-GNM degree connectivity) in UMAP space (Fig. 1F), and observed that the genes within the same GNM clustered together, indicating that they were highly co-expressed across the 213 hiPSC lines. To further validate that the GNMs captured genes enriched for being co-expressed across the 213 hiPSC lines, we first determined the number of co-expressed genes between each pairwise combination of GNMs. We then performed Fisher's Exact tests on the number of co-expressed genes within each GNM compared with the number of co-expressed genes between each set of paired GNMs, which showed that genes within a GNM were significantly more likely to be co-expressed with one another (Fig. 1G). These results show that the Leiden community detection algorithm identified 13 GNMs consisting of genes enriched for being co-expressed across the 213 hiPSC lines.

We next sought to determine if the GNMs were enriched for different pluripotency cell states. We first examined whether certain GNMs were associated with the estimated fraction of formative state cells (Fig. 1A, Supplementary Fig. 2A). For each of the 213 RNA-seq samples, we calculated a GNM score for each of the 13 GNMs by summing the inverse normal transformed TPM expression of the corresponding GNM-specific Pareto genes (each RNA-seq sample had 13 GNM scores). To identify associations between the estimated fraction of formative state cells and GNM scores across the 213 hiPSC lines, we ran a linear model and observed that GNM 5 was positively associated with the estimated proportion of formative state cells, while GNM 10 exhibited a strong negative association (Fig. 1H, I). Next, we utilized marker genes for hiPSC pluripotency states defined in previous studies (Fig. 1J, Supplementary Data 5). We discovered that two GNMs (10 and 11) were associated with genes upregulated in the 8-cell

**Fig. 1 | Community detection algorithm identifies hiPSC gene networks.**
**A** Cartoon depicting our hypothesis that hiPSCs are composed of varying proportions of pluripotent cell state subpopulations. Created with BioRender.com.
**B**−**E** Boxplots of gene expression in the 15 hiPSCs with the highest (>80%) estimated proportions versus the 69 hiPSCs with the lowest (0%) estimated proportions of formative-state cells (Supplementary Fig. 2A). **B** *DUSP5*, (**C**) *LEFTY1*, (**D**) *FZD5*, and (**E**) *FST*. The boxplot maxima and minima are set by the samples with the highest and lowest expression of the corresponding gene, the center line represents the median gene expression, the upper and lower hinges correspond to the 25th and 75th percentiles, and the upper and lower whiskers extend to the highest and lowest value within 1.5 times the interquartile range (IQR). Source data are provided as a Source Data file. **F** UMAP plot displaying the expression of the 2946 Pareto genes (colored by GNM membership) across 213 hiPSC RNA-seq samples. Source data are provided as a Source Data file. **G** Heatmap showing that genes within a GNM are enriched for co-expression. Pairwise Fisher's Exact tests were performed to validate that genes within the same GNM were more co-expressed with each other than with genes in other GNMs. Each cell is filled with the $\log_2$(Odds Ratio). For plot legibility, the enrichment range was set to −3.5 to 3.5. Source data are provided as a Source Data file. **H** Scatter plot showing the association between GNM 5 and the estimated proportion of formative-state cells across the 213 RNA-seq

samples. To calculate the nominal *P*-value, a linear model was used to examine the association between the GNM 5 Score and the estimated relative formative population. Source data are provided as a Source Data file. **I** Scatter plot showing the association between GNM 10 and the estimated proportion of formative-state cells across the 213 RNA-seq samples. To calculate the nominal *P*-value, a linear model was used to examine the association between the GNM 10 Score and the estimated relative formative population. Source data are provided as a Source Data file. **J** Heatmap showing the enrichment of pluripotency cell state-associated genes in the 13 GNMs. A Fisher's Exact Test was performed on seven published gene sets (Supplementary Data 5) associated with hiPSC cell states on each GNM to calculate the Odds Ratio. Published gene set labels on the *y*-axis were colored to indicate whether they were curated from in vivo (red) or in vitro (blue) experiments. The GNMs on the *x*-axis are ordered the same as in the above panel (**G**). Each cell is filled with the $\log_2$(Odds Ratio) for significant GNM cell state associations (red = enrichment, blue = depletion, white = non-significant). For plot legibility, the enrichment range was set to −3.5 to 3.5 and the "Primitive Endoderm" label represents the primitive endoderm-primed founder cells. Source data are provided as a Source Data file. **K** Network graph showing a subset of the co-expressed genes in GNM 5. **L** Network graph showing a subset of the co-expressed genes in GNM 10.

like cells (8CLC) totipotent cell state;[3] GNM 5 was enriched for gene sets upregulated in the naïve[29], formative (EPE)[1], epiblast[29] states; GNM 9 and GNM 13 were both enriched for genes upregulated in the primitive endoderm-primed founder cells (PrE)[29]; while GNM 10 was strongly depleted for genes associated with the formative state[1] and enriched for genes associated with the primed[1], 8CLC[3], and trophectoderm[29] states (Fig. 1J). Examining GNM 5 and 10 networks, we observed that they consisted of genes in molecular pathways characteristic of different pluripotent cell states. For example, GNM 5 consists of genes in Nodal signaling, ERK signaling, self-renewal processes, extracellular (ECM) matrix formation, and Wnt signaling pathways (Fig. 1K), which are characteristic of formative and naïve pluripotent cell states[1]. While GNM 10 consists of frizzled receptors, and developmental transcription factors (TFs) (Fig. 1L), characteristic of the primed pluripotent stem cell state[1,4]. These findings show that certain GNMs are enriched for genes upregulated or downregulated in specific pluripotency states and that their discovery is due to differences in the proportion of the pluripotency cellular states across the hiPSC samples.

Altogether, we identified 13 modules of co-expressed genes differentially enriched for the expression of marker genes defining the continuum of hiPSC pluripotency cell states.

### Identification of genome-wide regulatory network underlying the co-expression of gene modules

After discovering gene co-expression modules associated with self-renewal and interconvertible pluripotency cell states, we sought to identify the underlying regulatory modules (Fig. 2A). We generated 150 independent ATAC-seq libraries from 143 hiPSC lines derived from 133 iPSCORE individuals (7 lines were cultured independently twice and 5 individuals each had two or three independent clones) (Supplementary Data 6).

To identify modules of co-accessible regulatory elements, we calculated the accessibility of 56,978 reference peaks across the 150 ATAC-seq samples (Supplementary Fig. 5, Supplementary Data 8) and applied an LMM to identify pairs of co-accessible ATAC-seq peaks. We created a genome-wide regulatory network (RN) using 8,696,814 co-accessible autosomal ATAC-seq peak pairs (*P*-value < $5 \times 10^{-8}$, Effect Size > 0) as edges. We applied the unsupervised Leiden community detection algorithm[24,25] to the RN and identified 13 major regulatory network modules (RNMs) each consisting of at least 500 ATAC-seq peaks (mean = 3673.9 ± 1261.9 peaks) (Fig. 2B). Of the 56,978 reference peaks, 47,761 were present in these 13 RNMs. We analyzed the accessibility of the most

interconnected peaks within each module (top 10% intramodular degree) and observed RNM-specific clustering in the UMAP space (Fig. 2C), indicating that the RNMs capture ATAC-seq peaks with similar varying accessibility across the hiPSC lines. We also calculated the intramodular co-accessibility enrichment between each pairwise combination of the 13 RNMs, which showed that peaks within an RNM were significantly more likely to be co-accessible compared with peaks in different RNMs (Fig. 2D, Supplementary Data 13). These findings validate that the RNMs capture highly co-accessible peaks.

We sought to determine whether the RNMs were associated with specific pluripotency states. We initially performed cellular deconvolution on the 150 ATAC-seq libraries using the 200 most differentially accessible ATAC-seq peaks[30,31] from FACS-sorted formative and primed cells[1] to generate estimates of the relative proportions of each cell state across the hiPSC lines. Only a small fraction of cells (range 0−36.6%) in each ATAC-seq sample were estimated to be in the formative state (Fig. 2E, Supplementary Data 6, Supplementary Data 7). We calculated RNM scores by summing the inverse normal transformed accessibility of 9545 Pareto peaks (top 20% intramodular connectivity) for each ATAC-seq sample (each sample had 13 RNM scores) (Supplementary Data 8). We then ran a linear model to test for associations between the estimated proportion of formative state cells and RNM scores (Fig. 2F−H). We observed that RNM 3 had a strong positive correlation with the estimated formative proportion, while RNMs 2 and 8 had weaker positive correlations (Fig. 2F−H). Finally, we examined whether genome-wide ATAC-seq peak co-accessibility is associated with the coordinated gene expression in different pluripotency states. We annotated 32,327 ATAC-seq peaks in the 13 RNMs with 12,078 neighboring candidate target genes (see Methods) and then performed a Fisher's Exact test to calculate enrichments of RNMs in GNMs. We found that all RNMs had an association with at least one GNM (Fig. 2I). For example, five RNMs (2, 3, 5, 8, and 13) were positively enriched for the formative GNM 5 (Fig. 1G, H, J, K), suggesting that co-accessibility of ATAC-seq peaks across the genome mechanistically underlie the differential expression of genes between pluripotency states.

Altogether, we discovered 13 regulatory network modules composed of highly co-accessible peaks across 143 hiPSCs. We show that the RNMs are associated with different pluripotency states and demonstrate considerable variability in their proportions between hiPSC lines. Additionally, we show that these regulatory modules were strongly associated with, and likely mechanistically underlie, the coordinated expression within gene network modules.

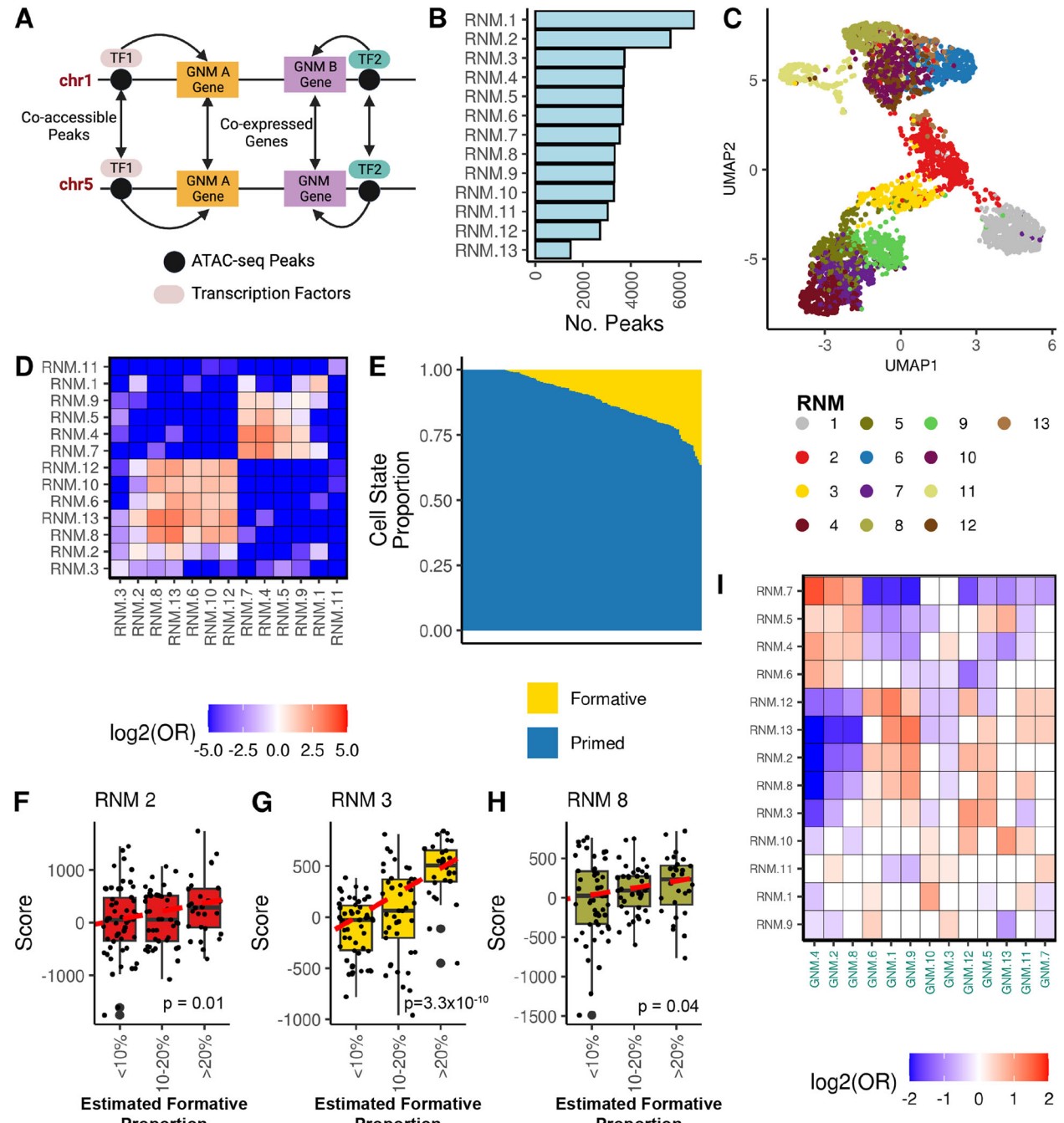

**Fig. 2 | Community detection algorithm identifies hiPSC regulatory networks.** **A** Diagram of proposed molecular mechanisms underlying gene co-expression networks. Co-expression of genes located on different chromosomes but within the same GNM (**A** or **B**) is mediated via regulatory elements that are co-accessible because they bind the same transcription factors (TF1 or TF2). Created with BioRender.com. **B** Histogram displaying the number of peaks in each of the 13 major RNMs. **C** UMAP plot of 4772 ATAC-seq peaks in the 13 major RNMs. We plotted ATAC-seq peaks with the top 10% intramodular co-accessibility in each RNM, as opposed to the Pareto peaks ($n = 9545$) for plot legibility. Each point represents a peak colored by its corresponding RNM. Source data are provided as a Source Data file. **D** Heatmap showing the pairwise associations between 13 RNMs based on the co-accessibility enrichment. Each cell is filled with the $\log_2$(Odds Ratio) for the corresponding RNM pair. Source data are provided as a Source Data file. **E** Barplot showing the estimated cell state proportions across 150 ATAC-seq samples. Previously published ATAC-seq peaks from FACs sorted cells representing formative and primed pluripotency states[1] were used to perform cellular deconvolution on the 150 iPSCORE hiPSC ATAC-seq samples. Each stacked barplot corresponds to an ATAC-seq sample and the colors correspond to the estimated formative and

primed cell states. Source data are provided as a Source Data file. **F–H** Boxplots demonstrating RNM 2 (**F**), 3 (**G**), and 8 (**H**) associations with the estimated proportion of cells in the formative state. A linear model was used to calculate the association between RNM scores (summed inversed normal transformed accessibility of RNM Pareto peaks) and the estimated formative proportion of 123 ATAC-seq samples (<100% estimated formative high proportion). Each point represents an ATAC-seq sample. Samples were binned by estimated proportion for boxplot legibility. The boxplot maxima and minima are set by the samples with the highest and lowest RNM score, the center line represents the RNM score, the upper and lower hinges correspond to the 25th and 75th percentiles, and the upper and lower whiskers extend to the highest and lowest value within 1.5 times the interquartile range (IQR). Source data are provided as a Source Data file. **I** Heatmap showing GNM-RNM associations. 32,327 ATAC-seq peaks in the 13 major RNMs were annotated with a putative target gene (see "Methods"). We performed pairwise Fisher's Exact test to calculate enrichments ($\log_2$(Odds Ratio)) of RNM peaks in GNMs. Non-significant associations are filled in white. Source data are provided as a Source Data file.

## Functional annotation of hiPSC ATAC-seq peaks

We hypothesized that the co-accessible ATAC-seq peaks within a module have similar epigenetic profiles and molecular functions. However, unlike genes, which have been annotated with regard to their expression profiles and biological functions, regulatory elements have not been well characterized. Therefore, in order to functionally characterize the 13 RNMs, we annotated the ATAC-seq peaks with three epigenetic annotations (Supplementary Fig. 6A): (1) hiPSC-specific chromatin states[32–34], (2) TF binding sites[35–37], and (3) formative (EPE)-associated ATAC-seq peaks[1].

We initially annotated each of the 56,978 reference ATAC-seq peaks with chromatin states collapsed into five main categories (See "Methods", Supplementary Data 8) and observed that 46.8% of the ATAC-seq peaks were in enhancers, 22.0% were in active promoters, 13.9% were in bivalent or poised chromatin, 5.8% were in repressed polycomb regions, and 11.45% were in transcribed regions (Supplementary Fig. 6B). Our annotations were consistent with previous characterizations of hiPSC regulatory elements[1,38–40], specifically; (1) the relatively large fraction of the peaks in bivalent chromatin indicating open but inactive regulatory elements, and (2) the presence of peaks in polycomb regions.

We identified 187 TFs that were expressed in the hiPSCs and predicted their binding across the 56,978 reference ATAC-seq peaks using TOBIAS, a digital footprinting method[35] (Supplementary Fig. 6C, Supplementary Data 10, Supplementary Data 11). To validate the TOBIAS predictions, we used ENCODE ChIP-seq data for 18 TFs generated using H1 embryonic stem cells (ESCs)[37,41,42] (See "Methods", Supplementary Fig. 7, Supplementary Data 9). We observed that TOBIAS-predicted binding sites were strongly enriched in corresponding TF ChIP-seq peaks (Fisher's Exact test; maximum nominal $p = 5.5 \times 10^{-188}$), indicating that TOBIAS accurately predicted TF binding across the epigenome. Since TOBIAS often predicted that TFs with similar motifs bound at the same site, we collapsed the 187 TFs based on their predicted overlap of bound sites (See "Methods", Supplementary Fig. 8) into 92 TF groups (49 single motifs, 24 same TF family, 19 contained TFs from different families and are referred to as "Complexes") (Supplementary Data 10, Supplementary Data 11). Approximately 31% of the 56,978 reference ATAC-seq peaks were predicted to be not bound by TFs.

Finally, we annotated 938 ATAC-seq peaks as associated with the formative state because they overlapped peaks specific to the FACs sorted hiPSC formative subpopulation[1] (Supplementary Fig. 6D, Supplementary Data 8). We then annotated 2981 peaks as associated with the primed state because they overlapped peaks specific to the FACs sorted hiPSC primed subpopulation[1].

## Functional characterization of regulatory network modules

To further characterize the 13 RNMs we calculated their enrichments for formative and primed state ATAC-seq peaks (Fig. 3A), the 5 collapsed chromatin states (Fig. 3B), and the 92 TF groups including "Not Bound" peaks (Fig. 3C; Supplementary Fig. 9, Supplementary Data 12).

We initially examined the three RNMs (2, 3, and 8) (Fig. 3A) enriched for the formative-specific peaks. RNM 2 was enriched for pluripotency TFs (NANOG-OCT4 complex, POU2F2/OCT2, SOX-LEF1 complex, TEAD Family) in enhancers, suggesting that it represents the hallmark hiPSC pluripotency regulatory network active in the formative state (Fig. 3A–C). RNM 3 was highly enriched for enhancers and the TEAD Family (TEAD1 and TEAD4) (Fig. 3A–C). TEAD signaling is strongly implicated in the differentiation of hiPSC to the trophoblast lineage[43]; and recently Dattani et al.[44] showed that suppression of YAP/TEAD signaling was critical to the insulation of naive hiPSC from trophoblast differentiation. Though the capacity for conventional hiPSC to undergo trophoblast differentiation has been the subject of considerable controversy[43,45], it is certainly possible that formative state cells, closer to the naïve-state, might be capable of entering this

differentiation pathway, and that regulatory elements in RMN 3 are in some way primed for activation in naïve and formative state cells[46]. RNM 8 was strongly enriched for $G_1$ cell cycle-associated TFs (E2F Family, E2F2, E2F5, SP-E2F Complex) which is consistent with the fact that the formative state has a highly proliferative phenotype and an abbreviated $G_1$ phase[1,47] (Fig. 3A–C). This suggests that RNM 8 represents a network that may underlie cell cycle regulation in formative pluripotency.

The three primed RNMs (6, 7, and 10) exhibited similar enrichments for E2F cell cycle-associated TFBSs (Fig. 3A–C), which is consistent with the primed state being in the $G_1$ phase[1]. Interestingly, the primed associated RNMs displayed distinct chromatin state enrichments (Fig. 3A, B). RNMs 6 and 7 were strongly enriched with active promoters and actively transcribed regions, respectively, (Fig. 3B), further supporting that the primed state is more transcriptionally active than the formative state[1]. RNM 10 is enriched with both bivalent chromatin and repressed polycomb complexes (Fig. 3B) and likely captures a primed-specific regulatory mechanism for the repression of developmental genes. These observations suggest that repressed polycomb complexes are likely activated at the transition from the formative to the primed state. Overall, these analyses show that the RNMs encompass well-known epigenomic and cell state-specific features present in formative to primed states and importantly captured the coordinated activity of the majority of regulatory elements underpinning the spectrum of pluripotency traits in hiPSCs.

Seven RNMs (1, 4, 5, 9, 11, 12, and 13) were not enriched with formative or primed-associated peaks. While RNM 1 is not cell state associated, its sole enrichment is with NANOG-OCT4 and OTX2 TFs (Fig. 3C). The fact that the loss of NANOG-OCT4 mediated self-renewal initiates neuronal differentiation[48,49], and that REST (RE1-Silencing Transcription factor), which is involved in the repression of neural genes in non-neuronal cells, was depleted[50], suggests RNM 1 could represent a regulatory network underlying neural specification. RNM 9 has similar chromatin state and TF enrichments as the canonical pluripotency RNM 2 but is not enriched with the formative or primed-associated peaks (Fig. 3A–C), suggesting that there may be a shared NANOG-OCT4-mediated regulatory network active across different pluripotent cell states.

To further demonstrate the utility of annotating the ATAC-seq peaks in the RNMs (pluripotency state, chromatin state, and predicted TFBSs), we plotted a 2 MB interval on chromosome 4 surrounding SMAD1, which is a transcription factor involved in early cell fate decisions during specification of trophoblast and amnion, and later, gastrulation[43,51,52] via regulation of bone morphogenic proteins (BMPs). We observed nine formative-state associated peaks and multiple peaks binding SMAD Family (SMAD2 and SMAD4) and SOX-LEF1 Complex (SOX2, SOX3, SOX4, LEF1) TFs (Fig. 3D). Focusing on the -15 kb interval harboring the SMAD1 promoter, we observed an RNM 8 ATAC-seq peak overlapping a formative-state associated peak, bound by a SOX-LEF1 Complex TF (Fig. 3E). RNM 8 was enriched with formative-state peaks and bivalent chromatin (Fig. 3A, B), suggesting that the regulatory elements in this network are not only associated with $G_1$ cell cycle-associated TFs (see above) but also repressing developmental processes and differentiation. Altogether these observations demonstrate the utility of annotating RNM ATAC-seq peaks with epigenomic and cell state-specific features.

In summary, these findings present important biological predictions for hiPSC regulatory networks and biology. Foremost we demonstrate that it is possible to capture the coordinated activity of the majority of the regulatory elements in hiPSCs and bin these elements into discrete functionally characterized regulatory networks that most likely mechanistically underlie self-renewal, pluripotency, and cell state transitions.

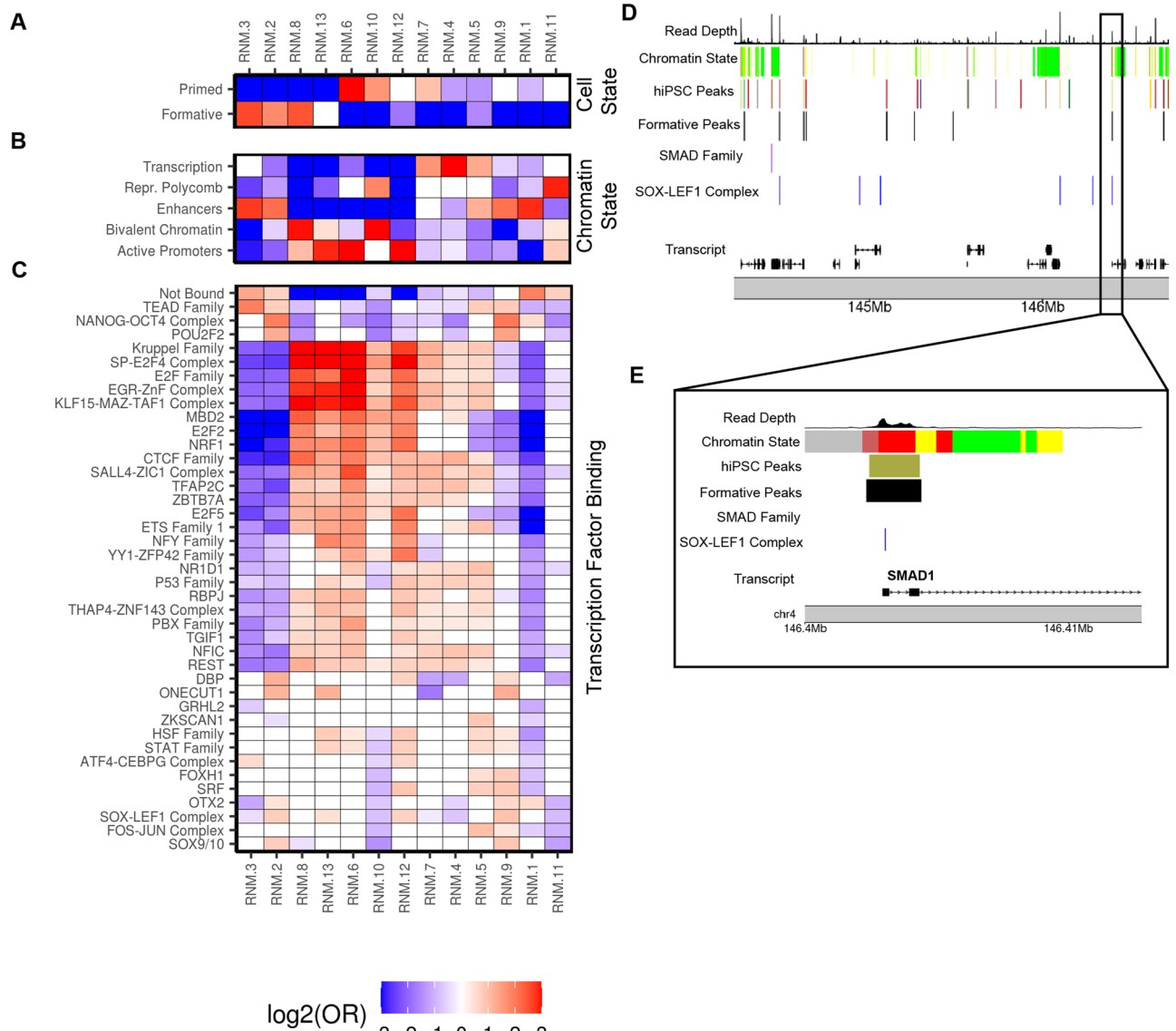

log2(OR)
-3 -2 -1 0 1 2 3

**Fig. 3 | RNM functional characterization. A** Heatmap displaying the enrichment (Odds Ratio) of formative and primed-state associated peaks in each RNM calculated using Fisher's Exact tests. For Fig. 3A–C, the following features are consistent; (1) the RNM order on the *x*-axis, (2) each cell is filled with the log₂(Odds Ratio) for the corresponding RNM (red = enrichment, blue = depletion, white = non-significant), and (3) the enrichment range was set from −3 to 3 for plot legibility. Source data are provided as a Source Data file. **B** Heatmap showing enrichment (Odds Ratio) of the annotations for 5 collapsed iPSC chromatin states in each RNM calculated using Fisher's Exact tests (Supplementary Fig. 6B). Source data are provided as a Source Data file. **C** Heatmap displaying enrichment (Odds Ratio) of 41 selected TF groups in each RNM calculated using Fisher's Exact tests. A heatmap reporting the RNM enrichment for all 92 TF groups and "Not Bound" is shown in

Supplementary Fig. 9. Source data are provided as a Source Data file. **D** Genome browser plot exhibiting the distribution of the three epigenetic annotations in the *chr4:144,200,000:146,700,000* locus. The top track is the read depth of the merged BAM file of 34 reference samples (see "Methods"), the track below is the five collapsed chromatin states (colored by chromatin state membership) (Supplementary Fig. 6B), the following four tracks respectively show co-accessibility of hiPSC ATAC-seq peaks (colored by RNM membership), the formative-state ATAC-seq peaks reanalyzed from Lau et al[1], predicted binding sites for SMAD Family and SOX-LEF1 Complex, and transcript hg19 coordinates. **E** Detailed genome browser view highlighting the *SMAD1* region of the *chr4:146,400,000:146,415,000* locus. As indicated by the black box, the plot is a subset of the locus displayed in Fig. 3D, and has concordant track ordering.

## hiPSC regulatory networks are conserved in fetal cell types

After establishing the architecture of regulatory networks in hiPSCs, we sought to examine if the co-expression of genes in other cell types could use the same regulatory networks (i.e., the same ATAC-seq peaks would show co-accessibility) or if new regulatory networks would be established. To determine if the hiPSC regulatory networks were conserved in other cell types we examined 54 sets of fetal cell type-specific ATAC-seq peaks from the Descartes Human Chromatin Accessibility Atlas[53]. Of the 47,761 hiPSC reference ATAC-seq peaks in the 13 major RNMs we identified 11,830 that overlapped cell type-specific peaks for at least one of the 54 fetal cell types (Fig. 4A, B,

Supplementary Data 13). For each RNM, we then examined if the overlapping peaks were enriched in one or more of the 54 fetal cell types (Fig. 4C).

Based on the RNM functional annotations (Fig. 3), we hypothesized that RNM 3, and RNM 1 could represent regulatory networks respectively underlying the early emergence of fate commitment to the placental trophoblast and neural lineages, respectively. We observed that RNM 3 was enriched for cell type-specific peaks in villous cytotrophoblasts, extravillous trophoblasts, trophoblast giant, and eight other cell types (Fig. 4C) consistent with its enrichment in TEAD elements and their role in trophoblast differentiation[54]. On the other

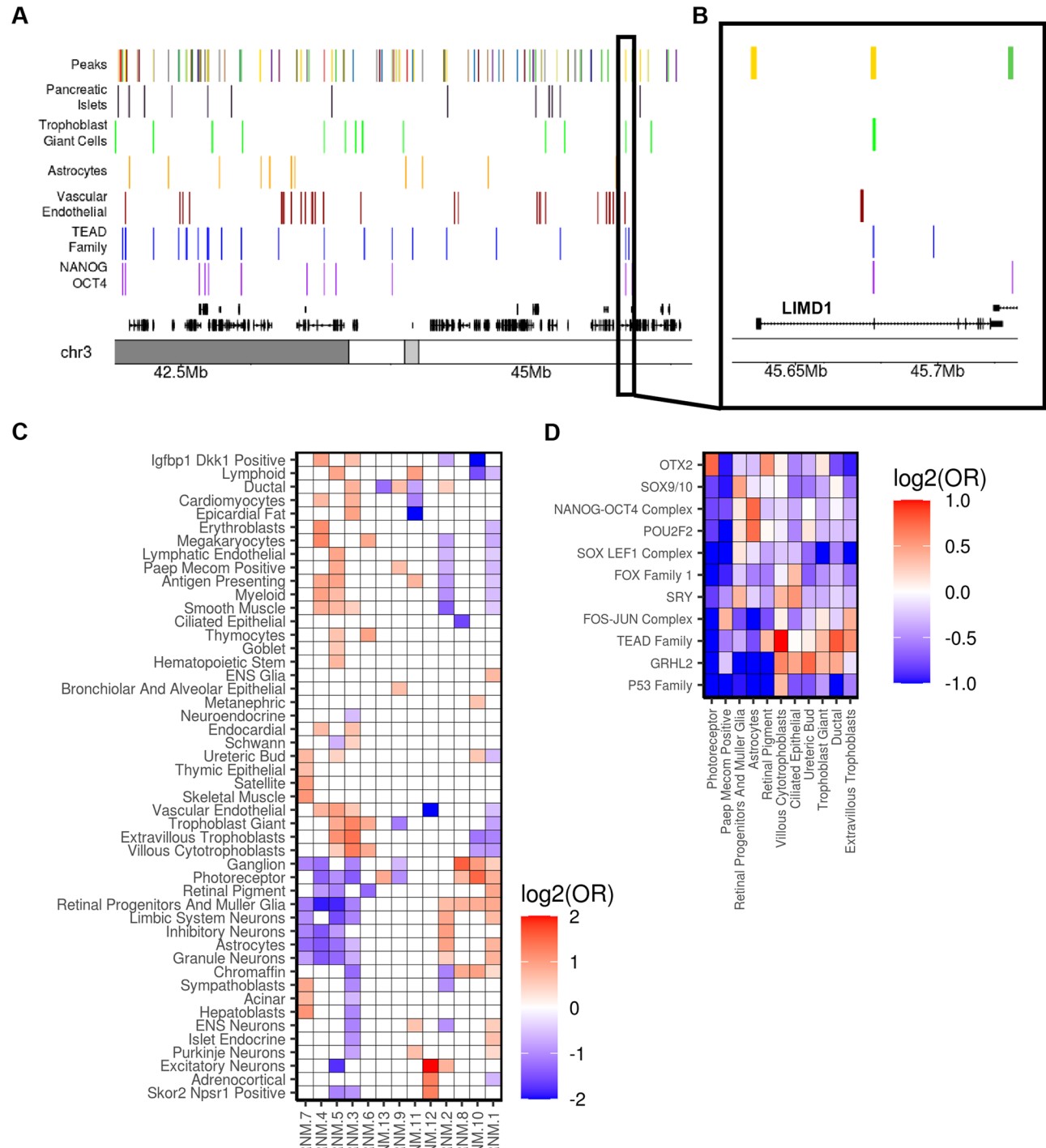

**Fig. 4 | RNM enrichments for fetal cell type-specific ATAC-seq peaks. A** Genome browser visualization showing (from the top): (1) hiPSC ATAC-seq peaks (colored by RNM); fetal cell type-specific peaks from the Descartes Atlas for (2) pancreatic islets, (3) trophoblast giant cells, (4) astrocytes, (5) vascular endothelial cells; TF footprints for (6) TEAD Family, and (7) NANOG-OCT4 Complex; and (8) transcript hg19 coordinates. **B** As indicated by the black box, the view is a section of Fig. 4A at higher resolution and demonstrates that an RNM 3 hiPSC ATAC-seq peak bound by NANOG-OCT4 and TEAD Family TFs overlaps a trophoblast giant cell-specific peak near an *LIMD1* exon. **C** Heatmap displaying enrichment of the 13 RNMs in fetal cell type-specific peaks. Fisher's Exact tests were used to calculate the enrichment

(Odds Ratio) of the cell type-specific peaks for 54 fetal cell types in each RNM. 5 fetal cell type-specific peaks were omitted from the plot because they were not enriched an RNM. Each cell is filled with the log$_2$(Odds Ratio) for the corresponding fetal cell types and RNM (red = enrichment, blue = depletion, white = non-significant). Source data are provided as a Source Data file. **D** Heatmap showing TF enrichment in the ATAC-seq peaks underlying enrichments in RNM 10. Fisher's Exact tests were used to calculate the enrichment (Odds Ratio) of the 92 TF groups in the hiPSC peak underlying the fetal cell type associations. Each cell is filled with the log$_2$(Odds Ratio) for the indicated fetal cell type and TF group (red = enrichment, blue = depletion, white = non-significant). Source data are provided as a Source Data file.

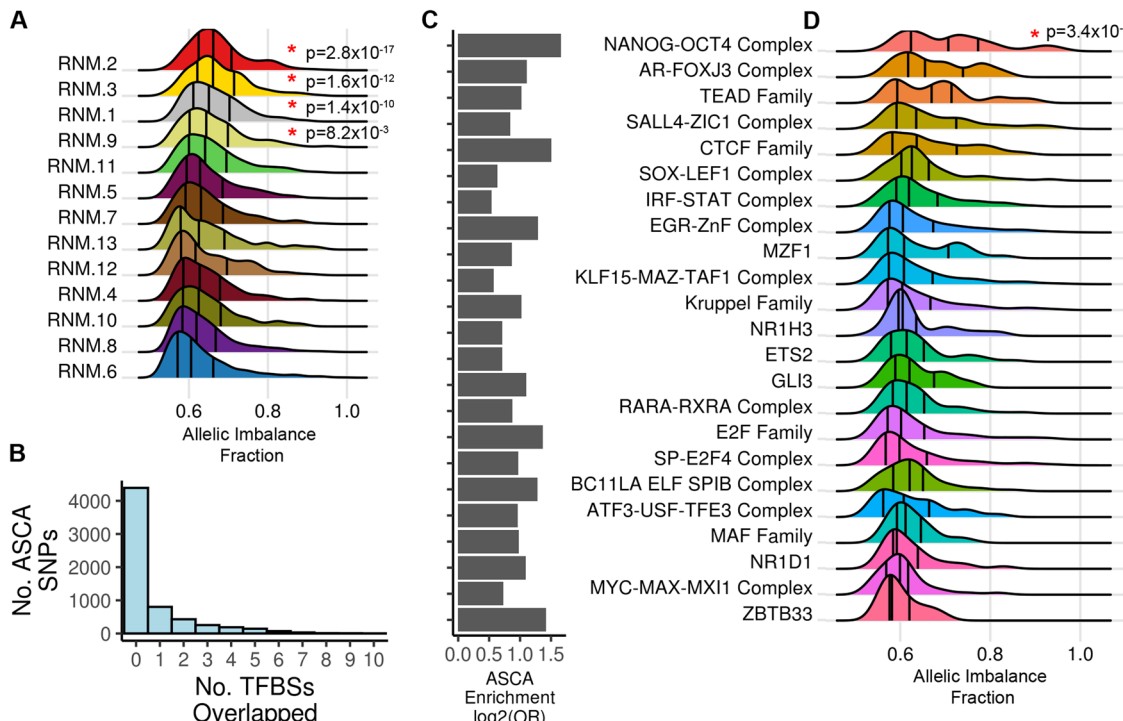

**Fig. 5 | RNMs and TFBSs exhibit differential allelic imbalance. A** Density plot showing the allelic imbalance fraction (AIF) of ASCA SNPs in the 13 RNMs. The AIF densities (x-axis) for the 13 RNMs (y-axis) demonstrate that 4 RNMs (indicated by red asterisks) contain ASCA SNPs that have significantly greater allelic imbalance compared to ASCA SNPs in other RNMs (one-sided Mann-Whitney U Test, Benjamini-Hochberg adjusted P-value < 0.05). Source data are provided as a Source Data file. **B** Histogram showing the number of ASCA SNPs (y-axis) that overlapped predicted TFBSs (x-axis). 6323 SNPs in 4241 ATAC-seq peaks exhibited allele-specific chromatin accessibility (ASCA). ASCA SNPs were intersected with predicted TBFSs for all 92 TF groups. 4299 ASCA SNPs did not overlap a TFBS and 1933 overlapped one or more TFBSs. Source data are provided as a Source Data file. **C** Barplot demonstrating enrichment for ASCA SNPs in 23 TF groups. A two-tailed Fisher's Exact test was used to calculate enrichment (Odds Ratio) for ASCA SNPs in the 92 TF groups (Supplementary Data 12). The barplot displays the enrichment (log$_2$(Odds Ratio)) for the 23 TF groups that were significant (adjusted P-value < 0.05). The TF groups are ordered based on their AIF densities in Fig. 5D. Source data are provided as a Source Data file. **D** Density plot showing the allelic imbalance fraction (AIF) of ASCA SNPs in the 23 TF groups. The AIF densities (x-axis) for the 23 TF groups (y-axis) demonstrate that NANOG-OCT4 Complex binding sites (indicated by red asterisk) contain ASCA SNPs that have greater allelic imbalance compared to ASCA SNPs in the other 22 TF groups (one-sided Mann-Whitney U Test, Benjamini-Hochberg adjusted P-value < 0.05). Source data are provided as a Source Data file.

hand, RNM 1 was enriched in multiple neural cell types including astrocytes, granule neurons, limbic system neurons, ENS glia, and ENS neurons, consistent with observations above on its likely occupancy by NANOG-OCT4, factors that probably play a role in suppressing regulatory elements important in the differentiated neuronal state[55]. These findings suggest that some of the RNMs present in the hiPSCs could represent networks important for lineage specification and are conserved in the derived cell types.

We reasoned that one possible mechanism underlying RNM conservation during development would involve binding of the same TF group to the same regulatory elements in both hiPSCs and the derived fetal cell type. We directly calculated the enrichment of the 92 hiPSC TF groups in the 54 fetal cell-type specific ATAC-seq peaks (Fig. 4D, Supplementary Data 14). TEAD Family hiPSC TFBSs exhibited a strong enrichment in the three trophoblast cell type-specific peaks (Fig. 4D), recapitulating their importance in the development of placental lineages (Fig. 4C)[44,56]. Trophoblast giant cells and villous cytotrophoblasts were also enriched with GRHL2, which has been reported to be a master regulator of placental branching morphogenesis[57]. NANOG-OCT4 Complex and POU2F2 were strongly enriched in astrocyte-specific peaks, which further supports the NANOG-mediated repression of neural tissue differentiation during pluripotency[55]. Altogether, these results suggest that some hiPSC regulatory networks are conserved in fetal cell types and that the molecular mechanism underlying conservation can either be the binding of the same TF Family to the same regulatory elements in both stages of

development or NANOG-mediated repression of regulatory elements active in early tissue differentiation.

## Characterization of allele-specific chromatin accessibility SNPs (ASCA-SNPs)

Previous studies have shown that both genetic and epigenetic variation affect hiPSC phenotypes[58–60], however, the underlying mechanisms are poorly understood. One likely mechanism through which genetic variation influences hiPSC phenotypes[58–60] is by affecting chromatin accessibility. We examined 105,055 SNPs (MAF ≥ 0.05, HWE p-value > 1 × 10$^{-6}$) present in 35,614 ATAC-seq peaks in the 13 major RNMs and determined that 6323 displayed allele-specific chromatin accessibility (ASCA, Supplementary Data 15). To determine if specific RNMs harbored ASCA SNPs with large effects, we performed Mann-Whitney U tests on the allelic imbalance fraction (AIF) of ASCA SNPs in each of the 13 RNMs using the other 12 RNMs as background (Fig. 5A). We observed that four RNMs 1, 2, 3, and 9 (all enriched for enhancers Fig. 3B), contained ASCA SNPs with a greater allelic imbalance fraction (AIF) than the ASCA SNPs in the other 9 RNMs (Fig. 5A). Of note, RNMs 2 and 3 were associated with the formative state (Fig. 3A) and represent the NANOG-OCT4 and TEAD-mediated (Fig. 3C) regulatory networks, respectively. While RNMs 1 and 9 were depleted for formative state regulatory elements, and represented NANOG-OCT4-mediated regulatory networks most likely shared across different pluripotent state(s). Our findings show that compared with the other regulatory networks active in hiPSCs, the regulatory elements in the pluripotency-

associated networks harbor genetic variants that exert large effects on chromatin accessibility.

Genetic variation can influence chromatin accessibility by affecting the binding specificity of TF motifs in regulatory elements[61–64]. We examined if any specific group of TF groups were enriched for having ASCA SNPs overlapping their binding sites. Of the 6323 ASCA SNPs, 1933 (30.6%) overlapped at least one TFBS, while the majority did not overlap any TFBSs ($n = 4299$, Fig. 5B). We calculated the enrichment of the ASCA SNPs in TFBSs by performing a Fisher's Exact test for all 92 TF groups (Fig. 5C). Regulatory elements predicted to bind 23 TF groups were enriched with ASCA SNPs (adjusted two-tailed $P$-value < 0.05 and Odds Ratio > 1). These 23 groups included many pluripotency-associated TFs, such as NANOG-OCT4, TEAD Family, POU2F2, and SOX-LEF1 Complex (Fig. 5C). We examined the allelic imbalance fraction (AIF) of the ASCA SNPs in each of the 23 TF groups and observed that NANOG-OCT4 contained ASCA SNPs with significantly higher AIF indicating that they had large effects on chromatin accessibility (Fig. 5D). Six other TF groups (AR-FOXJ3, TEAD Family, CTCF Family, SALL4-ZIC1 Complex, and SOX-LEF1 Complex) exhibited increased AIF, albeit not significantly.

Taken together, these results show that pluripotency-associated TF binding sites and regulatory networks are enriched with genetic variants that have large effects on chromatin accessibility. While previous studies have shown that pluripotency TFs have a remarkably high degree of evolutionary constraint[65], our findings show that the regulatory elements to which they bind have a high amount of genetic variability suggesting that they may influence pluripotency state transitions in hiPSCs.

## Discussion

We sought to determine if regulatory network modules in hiPSCs underlying self-renewal and pluripotency could be discovered by examining genome-wide chromatin co-accessibility across samples from hundreds of iPSCORE individuals. We generated bulk ATAC-seq samples and employed cellular deconvolution to estimate the proportion of cells in the formative and primed states across 143 hiPSCs, which showed that the lines were composed of varying proportions of cells in these two pluripotency states. Using the bulk ATAC-seq data we calculated accessibility for each hiPSC at 56,978 reference peaks, and then using data from the 143 lines we determined genome-wide co-accessibility between all pairwise combinations of the reference peaks. Likewise, for 213 RNA-seq samples, we calculated co-expression between 16,110 genes. We applied an unsupervised community detection algorithm independently to each dataset which detected 13 regulatory network modules (RNMs) and 13 gene network modules (GNMs) and demonstrated that the RNMs and GNMs were strongly correlated with each other, suggesting that the coordinated co-accessibility of regulatory elements in the RNMs most likely underlie the coordinated expression of genes in the GNMs.

To functionally characterize the RNMs we annotated the individual ATAC-seq peaks with chromatin states, transcription factor (TF) binding sites[35–37], and for association with the formative and primed pluripotency subpopulations. We showed that regulatory elements in each of the 13 RNMs tended to share similar predicted TF binding, chromatin state profiles, and pluripotency state enrichments. Additional analyses suggest that some of the hiPSC regulatory networks are shared with derived fetal cell types (i.e., the same ATAC-seq peaks show co-accessibility).

Our findings highlight certain understudied regulatory networks that may be critical to hiPSC pluripotent cell state transitions. The formative cell state is enriched for self-renewal processes[1], compared to primed cells which can exhibit emergent features of lineage commitment[22]. Historically, epigenomic regulation in pluripotent cells has been characterized by a single regulatory network primarily mediated by NANOG, OCT4, and SOX2 binding (RNM 2)[23]. Our study suggests that there are several regulatory networks with distinct epigenetic profiles (chromatin state, TF binding), that mediate biological processes that are indispensable for maintaining the formative and primed pluripotency states (Fig. 3). For example, a TEAD4-mediated regulatory network (RNM 3, Fig. 3A–C) is present in the early post-implantation epiblast-like formative state and absent in the late post-implantation epiblast-like primed state. We demonstrate that trophoblast-specific peaks are enriched with RNM 3 peaks and TEAD Family binding sites (Fig. 4C, D), which is consistent with previous observations that TEAD4 is a pioneering factor for the placental lineage commitment[54]. Recently much effort has been put into developing efficient trophoblast differentiation protocols to study common pregnancy-related diseases, such as preeclampsia[43,45,46,56]. RNM 3 may serve as a resource to identify regulatory elements that are early determinants of trophoblast cell fate commitment. Bivalent chromatin and repressed polycomb complexes are well-known pluripotency-associated epigenomic features that repress the expression of genes involved in differentiation[14,40,66]. We show that the primed-associated RNM 10 is enriched in both polycomb repressed regions and bivalent chromatin, while the formative-associated RNM 8 is primarily enriched with bivalent chromatin, indicating that the two pluripotency states have discrete regulatory networks involving different epigenomic mechanisms for the repression of developmental genes (Fig. 3A, B). These analyses revealed regulation in hiPSCs is more complex than the canonical NANOG, OCT4, and SOX2 regulatory networks and that leveraging large sample sizes can resolve independent functional distinct pluripotency networks.

Our study also shows that RNMs are differentially enriched with 49 fetal cell type-specific peaks. These results indicate that distinct early embryonic regulatory networks are reused in later stages of fetal development. It also indicates that lineage-specific regulatory elements are regulated by distinct networks during pluripotency.

Our study also addresses a dearth of information about the role of genetic regulatory variation in stem cells because there are only a handful of labs actively investigating the topic. In addition to heritable, complex diseases that have known developmental origins (i.e. autism spectrum disorder, schizophrenia), there is mounting evidence that common diseases, such as type 2 diabetes[67] and cardiovascular disease[19], are influenced by regulatory variation that is active during fetal development. It is paramount to expand exploration into these areas of research to characterize developmental regulatory variation. In this study, we identify thousands of SNPs with allele-specific chromatin accessibility (ASCA). Despite the importance of maintaining pluripotency, we observed that the regulatory elements in the pluripotency-associated networks harbor genetic variants that exert large effects on chromatin accessibility in human stem cells in vitro. We also observed that compared with the other TF groups the binding sites for the NANOG-OCT4 complex were enriched for ASCA SNPs. Our findings suggest that variability in the regulatory elements in the pluripotency networks could play an important role in the observed varying proportions of pluripotency states between hiPSCs. These observations have remarkable implications for evolution and speciation.

In summary, our study suggests that epigenomic regulation of pluripotency and self-renewal processes are more complex than previously thought. It classifies hiPSC ATAC-seq peaks into 13 networks, 6 of which are associated with the formative or primed cell states, and 7 of which are likely associated with different pluripotent cell states. It also proposes potential mechanisms for how genetic variation influences TF binding and pluripotency regulatory mechanisms. These network classifications and ASCA SNPs could be a useful resource for researchers investigating, various aspects of pluripotency and cell state transitions, such as; (1) epigenomic regulation in stem cells, (2) cell fate commitment, (3) fetal developmental processes,

(4) embryonic-specific regulatory variation, (5) influence of regulatory variation on regulatory element co-accessibility, and (6) the development iPSC tissue derivation protocols.

## Methods

### Subject information

We used hiPSC lines from 219 individuals (Supplementary Data 1) recruited as part of the iPSCORE project[18,20,68]. There were 140 individuals belonging to 40 families composed of two or more subjects (range: 2–14 subjects) and 134 genetically unrelated individuals (some individuals were in the same family but only related by marriage). The iPSCORE_ID (i.e, iPSCORE_4_1) indicates family (4) and individual number (1). Each subject was assigned a Universal Unique Identifier (UUID). Sex, age, and self-reported race/ethnicity were reported at the time of enrollment of each subject. We previously estimated the ancestry of each participant by comparing their genomes to those of individuals in the 1000 Genomes Project (KGP)[69]. Recruitment of these individuals was approved by the Institutional Review Boards of the University of California, San Diego, and The Salk Institute (project no. 110776ZF).

### Molecular data sources

We used the following datasets from the iPSCORE resource:
- Previously published[18,20] 50X WGS (Illumina; 150-bp paired-end) generated from the blood or skin fibroblasts of the 219 individuals in this study. Of the 219 individuals, 127 had both RNA-seq and ATAC-seq libraries, 86 only had RNA-seq and 6 only had ATAC-seq.
- 150 ATAC-seq libraries generated from 143 hiPSC lines (collected after expanding in mTeSR1 medium with ROCK inhibitor on Matrigel) from 133 individuals, Supplemental Data 6;
- Previously published[18,20,68] RNA-seq data from 213 hiPSC lines (collected after culturing in mTeSR1 medium on Matrigel) from 213 individuals, Supplementary Data 2.
- Previously published[18] RNA-seq data from 3 hiPSC lines (collected after expanding in mTeSR1 medium containing ROCK inhibitor on Matrigel) from 3 individuals. Technical triplicates were collected for each hiPSC line.

Of note, both the RNA-seq and ATAC-seq data were generated from the same hiPSCs in the iPSCORE collection. However, the RNA-seq data and ATAC-seq data were generated from different passages of the hiPSC lines cultured under different experimental conditions. The RNA-seq data was generated from earlier passage ROCK inhibitor-naïve hiPSCs and the ATAC-seq data was generated from later passage hiPSCs that had been cultured with ROCK inhibitor. All iPSCORE resource molecular data is publicly available: RNA-seq at dbGaP under the accession code "phs000924 [https://www.ncbi.nlm.nih.gov/projects/gap/cgi-bin/study.cgi?study_id=phs000924.v4.p1]"; ATAC-seq at GEO under the accession code "GSE203377 [https://www.ncbi.nlm.nih.gov/geo/query/acc.cgi?acc=GSE203377]"; gVCF files at dbGaP under the accession code "phs001325 [https://www.ncbi.nlm.nih.gov/projects/gap/cgi-bin/ study.cgi?study_id=phs001325.v5.p1]". The TOBIAS transcription factor binding predictions, the TMM-normalized counts, the kinship matrix, and the reference narrow peak file have been deposited on "Figshare [https://figshare.com/projects/iPSC_coaccessibility/136585]".

We used the following publicly available datasets:
- 28 RNA-seq samples corresponding to either the formative state (early postimplantation epiblast-like; EPE) or the general population[1].
- 6 ATAC-seq samples corresponding to either the formative state (GCTM-2$^{high}$CD9$^{high}$EPCAM$^{high}$) or the primed state (GCTM-2$^{mid}$-CD9$^{mid}$)[1].

- ChIP-seq data for 18 Transcription Factors from ENCODE[37,41,70] (Supplementary Data 9);
- Seven hiPSC cell state gene sets obtained and curated from 3 published studies:
  - Two gene sets for the primed and formative states[1]
  - Four gene sets for epiblast, conventional PSCs, naïve PSCs, primitive endoderm[29]
  - One gene set for 8-cell like cells[3]
- Fifty-four sets of fetal tissue-specific ATAC-seq peaks from the Descartes; Human Chromatin Accessibility During Development Atlas[53]
- ChromHMM chromatin states in the ROADMAP Epigenomics hiPSC-18 line[33]

### Human iPSC (hiPSC) generation

Generation of the 219 hiPSC lines has previously been described in detail[18]. Briefly, cultures of primary dermal fibroblast cells were generated from a punch biopsy tissue, expanded for approximately 3 passages and cryopreserved. In batch, the fibroblasts were thawed and plated at a density of 250 K cells/well of 6-well plate and infected with the Cytotune Sendai virus (Life Technologies) per manufacturer's protocol to initiate reprogramming. The Sendai infected cells were maintained with 10% FBS/DMEM (Invitrogen) for Days 4–7 until the cells recovered and repopulated the well. These cells were then enzymatically dissociated using TrypLE (Life Technologies) and seeded onto a 10-cm dish pre-coated with mitotically inactive-mouse embryonic fibroblasts (MEFs) at a density of 500 K/dish and maintained with hESC medium, as previously described[71]. Emerging hiPSC colonies were manually picked after Day 21 and maintained on Matrigel (BD Corning) with mTeSR1 medium (Stem Cell Technologies). Multiple independently established hiPSC clones (i.e. referred to as lines) were derived from each individual, which were cultured typically to passage 12, and at least ten stock vials were frozen from each cell line. hiPSC pellets collected from each cell line were frozen in RTL plus buffer (Qiagen) and used for total RNA isolation. Sendai virus clearance typically occurred at or before P9 and was not detected in the hiPSC lines at the P12 stage of cryopreservation. A subset of the hiPSC lines was evaluated by flow cytometry for the expression of two pluripotent markers: Tra-1-81 (Alexa Fluor 488 anti-human, Biolegend) and SSEA-4 (PE anti-human, Biolegend). Pluripotency was also examined using PluriTest-RNAseq[18]. This iPSCORE resource was established as part of the Next Generation Consortium of the National Heart, Lung, and Blood Institute and is available to researchers through the biorepository at WiCell Research Institute (www.wicell.org; NHLBI Next Gen Collection). For-profit organizations can contact the corresponding author directly to discuss line availability.

### iPSCORE data generation and processing

**Generation of RNA-seq data for 213 hiPSC lines.** We analyzed previously published[20] RNA-seq data from 213 hiPSC lines from 213 iPSCORE individuals (Supplementary Data 2). Briefly, using the All-Prep DNA/RNA Mini Kit (QIAGEN) we extracted total RNA from lysed pellets frozen in RLT Plus buffer (collected after culturing hiPSC lines in mTeSR1 medium on Matrigel), assessed quality to make sure RNA integrity number (RIN) was 7.5 or greater, prepared libraries using the Illumina TruSeq stranded mRNA kit and sequenced with 100 bp paired end reads on HiSeq2500 (~22 M reads/per sample).

**Generation of RNA-seq for 3 hiPSC lines expanded with ROCK inhibitor.** While the RNA-seq and ATAC-seq data were generated from same hiPSC lines, there were batch effects because of differences in culture conditions (ie, the ATAC-seq data were generated from hiPSCs

that had been cultured with ROCK inhibitor). Therefore, the following data were used to examine the correlation between ATAC-seq peak co-accessibility network modules and gene co-expression network modules (see below: *Identifying Associations Between Gene and Regulatory Networks*). As previously described[18], for three individuals (iPSCORE_2_2, iPSCORE_2_3, and iPSCORE_2_9) one vial of a frozen hiPSC line was thawed into mTeSR1 medium containing 10 µM Y27632 (ROCK Inhibitor), plated onto one well of a Matrigel-coated six-well plate and incubated overnight. The media was replaced daily with mTeSR1 until the hiPSCs were visually estimated to be 80% confluent. The hiPSCs were then expanded by passaging from one well onto three wells of a six-well plate using Versene (Lonza) in mTeSR1 medium containing 5 µM ROCK inhibitor. When cells reached 80% confluency, the three wells of hiPSC line were individually dissociated using Accutase (Innovative Cell Technologies Inc.) in mTeSR1 medium containing 5 µM ROCK inhibitor, collected, counted, and $1 \times 10^6$ cells frozen in RLT plus buffer (three technical replicates per hiPSC line). Total RNA was extracted using AllPrep DNA/RNA Mini Kit (QIAGEN), RNA quality was assessed based on RNA integrity number (RIN) and nine libraries (three per hiPSC line) were prepared using the Illumina Tru-Seq stranded mRNA kit and sequenced with 100-bp paired end reads on a HiSeq2500.

**RNA-seq data processing.** We obtained transcript per million bp (TPM) as previously described[20]. Briefly, FASTQ files were aligned to the hg19 reference genome using STAR 2.5.0a[72] and Gencode V.34lift37[73] with parameters: *outFilterMultimapNmax 20, –outFilterMismatchNmax 999, –alignIntronMin 20, –alignIntronMax 1000000, –alignMates-GapMax 1000000*. We sorted and indexed the BAM files using Sambamba 0.6.7[74] and marked duplicates using bio-bambam2 (2.0.95) bammarkduplicates[75] and then re-indexed. To quantify gene expression (TPM), we used RSEM v1.2.20[76] with the following parameters: *--rsem-calculate-expression, --bam, --num-threads 16, --no-bam-output, --seed 3272015, --estimate-rspd, --paired-end, --forward-prob 0*. We identified 16,110 expressed genes on autosomes (TPM ≥ 1 in at least 20% of the 213 hiPSC lines) using rowQuantiles from the *matrixStats R package* (version 0.52.2).

**RNA-seq gene expression transformation.** To normalize gene expression and usage across samples, we performed inverse normal transformation using normalize.quantiles (preprocessCore package) and qnorm functions in R, in order to obtain mean expression = 0 and standard deviation = 1, as we previously described[19,77].

**Generation of ATAC-seq data for 143 hiPSC lines.** 150 ATAC-seq libraries were prepared from 143 hiPSC lines (for 7 lines we prepared two libraries) from 133 individuals (5 individuals each had two or three independent clones). The hiPSCs were harvested on Day 0 of our previously published large-scale iPSC-derived cardiovascular progenitor cells study prior to the initiation of WNT activation[78]. In brief, for each of the 150 ATAC-seq libraries, one vial of a frozen hiPSC line was thawed into mTeSR1 medium containing 10 µM ROCK Inhibitor (Sigma), plated on one well of a six-well plate coated with Matrigel, and incubated overnight. When hiPSCs were visually estimated to be 80% confluent they were passaged 1–2 times using Dispase II (2 mg/ml; Gibco/Life technologies) in mTeSR1 and plated onto Matrigel coated plates. The hiPSCs were then expanded by passaging using Versene (Lonza) in mTeSR1 medium containing 5 µM ROCK inhibitor. Finally, the hiPSCs were dissociated using Accutase (Innovative Cell Technologies Inc.) in mTeSR1 medium containing 5 µM ROCK inhibitor, collected, counted, and frozen as nuclear pellets of $2.5 \times 10^4$ cells for the ATAC-seq assay.

We performed ATAC-seq on the 150 hiPSC samples using a modified version of the Buenrostro et al.[79] protocol to: (1) selectively permeabilize the nuclear envelope of the cells without disrupting mitochondrial membrane, and (2) increase the number of reads that do not contain sequences spanning multiple nucleosomes. Frozen nuclear pellets were thawed on ice and tagmented in total volume of 25 µl in permeabilization buffer containing digitonin (10 mM Tris-HCl pH 7.5, 10 mM NaCl, 3 mM MgCl2, 0.01% digitonin) and 2.5 µl of Tn5 from Nextera DNA Library Preparation Kit (Illumina) for 45–75 min at 37 °C in a thermomixer (500 RPM shaking). Inclusion of digitonin, which is a mild detergent capable of selective permeabilization of cholesterol-rich bilayers[80], permeabilized the nuclear membranes (containing 20–35% cholesterol and cholesterol esters[80,81] without disrupting the mitochondrial membranes (0.5–3% cholesterol)[82]. To enrich for reads that span a single nucleosome, we included a double size selection step during purification using AMPure XP DNA beads (Beckman Coulter). This step enriched for reads containing inserts under 140 bp in length (shorter than the 146 bp wrapper around a single nucleosome). To eliminate confounding effects due to index hopping, all libraries within a pool were indexed with unique i7 and i5 barcodes. Libraries were amplified for 12 cycles using NEBNext® High-Fidelity 2X PCR Master Mix (NEB) in total volume of 25 µl in the presence of 800 nM of barcoded primers (400 nM each) custom synthesized by Integrated DNA Technologies (IDT).

**Sequencing of ATAC-seq.** The 150 ATAC-seq libraries were batched and sequenced with 100-bp paired-end reads on a HiSeq4000. To improve overall read depth 113 ATAC-seq libraries were sequenced twice (the second time with 150-bp paired end reads) resulting in 263 FASTQ files (Supplementary Data 6). The 263 FASTQ files were aligned to the hg19 reference genome with STAR[72] with the following flags: *--outFilterMultimapNmax 20, --outFilterMismatchNmax 999, --outFilterMismatchNoverLmax 0.04, --seedSearchStartLmax 20, --outFilterScoreMinOverLread 0.1, outFilterMatchNminOverLread 0.1*. We then filled in mate coordinates using samtools fixmate, marked duplicates using samtools markdup[83]. We used samtools merge to combine BAM files from the same library and indexed the merged BAM files with samtools index. We then filtered poorly mapped reads (MAPQ < 20), duplicates and reads less than 38 bp and greater than 2000 bp with samtools view to obtain reads passing filters (PF). We re-indexed the filtered merged BAM file with samtools index (version v6.7)[83]. After quality control, all BAM files from the same library were combined (1–2 paired files per hiPSC line) resulting in 150 BAM files (Supplementary Data 6).

**ATAC-seq peak calling and quality control.** MACS2 v.2.2.7[84] was used to call broad peaks for all 150 ATAC-seq libraries individually with settings: *--nomodel --shift −100 --extsize 200 -f BAMPE -g hs --broad*. To obtain one single set of high-quality peaks, 34 reference libraries were selected from unrelated individuals with 20–35 million reads passing filters, 60,000–90,000 broad peaks, and 100–225 bp mean fragment size to establish discrete regions of accessible chromatin (i.e., a reference set of ATAC-seq peaks) (Supplementary Fig. 5B, Supplementary Data 6). MACS2 v.2.2.7 was used to call narrow peaks on the 34 reference libraries jointly with settings: *-f BAMPE -g hs -t --nomodel --shift −100 --extsize 200 −narrow*. Narrow peaks were filtered by MACS2 score (<100), resulting in 136,333 peaks (including 132,225 autosomal peaks and 4108 peaks on sex chromosomes). For each of the 150 individual ATAC-seq samples coverage on the 136,333 peaks were obtained using *featureCounts* in Subread package v1.5.0[85]. Next, counts were trimmed mean of *M* value (TMM)-normalized for each peak across all 150 individual samples using the *cpm* function from edgeR v3.30.3[86].

**ATAC-seq peak transformation.** TMM values for all 136,333 ATAC-seq peaks were inverse normal transformed using the *normalize.quantiles* from the *preprocessCore* package and qnorm functions in R, in order to obtain mean expression = 0 and standard deviation = 1 for each peak across all 150 ATAC-seq samples.

**Identification of 56,978 autosomal reference peaks.** To reduce the computational burden in downstream analyses, we selected a subset of 56,978 reference peaks based on their MACS2 peak score and enrichment in active chromatin. We filtered ATAC-seq peaks that overlapped blacklisted regions[87], peaks in ZNF genes & repeats, heterochromatin, quiescent chromatin states from the ChromHMM model for iPSC-18[33], and peaks on sex chromosomes. We binned the remaining 87,659 ATAC-seq peaks into 20 MACS2 score quantiles each containing 4383 peaks, where quantile 20 consisted of the peaks with the highest scores. We then examined the enrichment of the 87,659 ATAC-seq peaks in iPSC-18 ChromHMM chromatin states by MACS2 score quantile (Supplementary Fig. 5C). We created 20 bed files containing the coordinates of the peaks in each quantile and calculated their enrichment (Odds Ratio) in each of the 12 chromatin states for iPSC-18, using *bedtools fisher*. The higher quantiles tended to be enriched for active (TssA, TssAFlnk, TxFlnk), bivalent (TssBiv, BivFlnk, EnhBiv), and repressed polycomb (ReprPC, ReprPCWk) chromatin. Based on these observations, we calculated pairwise co-accessibility for the 56,978 ATAC-seq peaks in MACS2 quantiles 8–20 (see *Identifying co-accessible peaks* section below), which alleviated the computational burden by eliminating $7 \times 10^9$ pairwise tests between peaks with low accessibility and peaks in inactive regions of the genome. For downstream analyses, we resolved a single chromatin state for each peak by intersecting the maximum summits with the iPS-18 chromatin states (described below in *Collapsing into 5 hiPSC chromatin states*) (Supplementary Data 8).

## Cellular deconvolution of pluripotency states using gene expression signatures

From GEO (GSE119324)[1] we downloaded 28 paired RNA-seq FASTQ files, including, fourteen from five unique human embryonic stem cells (hESCs) sorted (GCTM-2[high]CD9[high]EPCAM[high]) for the formative (EPE) population and fourteen files for five unique samples generated from the primed population (unsorted). We aligned and filtered the 28 FASTQ files, with STAR 2.5.0a[72] and RSEM v1.2.20[76] as described above in *RNA-seq data processing* section. We performed differential expression analysis between the formative (EPE) and general population RNA-seq samples using DESeq2 (v.1.34) *R* package[88] and created a signature matrix containing 300 of the most differentially expressed genes (150 upregulated in the GCTM-2[high]CD9[high]EPCAM[high] cells and 150 downregulated relative to the unfractionated cells; Supplementary Data 3). We deconvoluted the 213 hiPSC RNA-seq samples by supplying the signature matrix and TPM expression matrix as input to the *CIBERSORT.R* script[21].

## Covariates for co-expression and co-accessibility analyses

To perform the gene co-expression analysis, we used the following covariates: (1) sex; (2) age; (3) hiPSC passage number; (4) number of properly paired reads; (5) 20 genotype principal components to account for global ancestry; and (6) kinship matrix. All covariates are available in Supplementary Data 1–2.

To perform the ATAC-seq peak co-accessibility analysis, we used the following covariates: (1) sex; (2) age; (3) hiPSC passage number; (4) number of reads passing filters; (5) the mean fragment size; (6) the number of broad peaks; (7) the ratio of 100 bp reads to 150 bp reads; (8) 20 genotype principal components to account for global ancestry; and (9) kinship matrix. All covariates are available in Supplementary Data 1 and Supplementary Data 6.

**Sex.** To account for sex-dependent chromatin, we assigned binary values to each sex and included it as a covariate.

**Biological and technical covariates.** We scaled age, hiPSC passage number, number of reads passing filters, and mean fragment size for each library to the mean across libraries and included these normalized values as covariates.

**Broad peaks.** As described above in the *ATAC-seq Peak Calling and Quality Control section*, we used MACS2 v.2.2.7 to call peaks on all 150 ATAC-seq libraries independently to assess sample quality. We normalized the number of broad ATAC-seq peaks for each sample to mean across all 150 samples.

**Read length ratio.** As described above in the *Sequencing of ATAC-seq section*, several ATAC-seq libraries were sequenced multiple times with 100 or 150 bp paired end reads and merged. To account for the different proportions of read lengths in merged libraries, we included the ratio of reads passing filters from 100 and 150 bp sequencing runs for each library as a covariate.

**Genotype principal component analysis.** We previously performed principal component analysis (PCA) on WGS variants to determine the global ancestry of each individual in this study[19]. Briefly, we used the genotypes of 1,634,010 SNPs that had allele frequencies between 30% and 60% in the 1000 Genomes Phase 3 Project and genotyped in both iPSCORE and GTEx. We merged the VCF files from 1000 Genomes, iPSCORE, and GTEx, and performed PCA using the *pca* function in *plink 1.90b3x*[89]. The top 20 genotype principal components used as covariates to account for global ancestry for all 219 individuals in this study can be found in Supplementary Data 1.

**Kinship matrix.** We included a kinship matrix generated for a previous iPSCORE study[19] as a random effects term to account for the genetic relatedness between individuals. The matrix was constructed using the kinship function in plink 1.90b3x[89] and the same set of 1,634,010 SNPs employed in the genotype PCA.

## Gene co-expression analyses

We leveraged the previously published RNA-seq dataset of 213 hiPSC lines from iPSCORE individuals[18,20,68] to identify gene networks that are active in stem cells. We integrated and curated gene sets of defined pluripotency cell states from external sources[1,3,18,29] to annotate gene networks with stem cell states.

**Identification and characterization of gene co-expression.** To identify pairwise correlations between the 16,110 expressed autosomal genes across the 213 hiPSC lines, we performed a gene co-expression analysis using a Linear Mixed-effects Model (LMM) with the *lmekin* function in the *coxme* R package v.2.2-17[24], which incorporates a kinship matrix to control for random effects from genetic relatedness. We included normalized values for age, hiPSC passage number, sex, number of properly paired reads, and the top 20 PCs from ancestry as fixed effect covariates (see above: *Covariates for co-expression and co-accessibility analyses*).

$$Y_{ij} = \beta_k Y_{ik} + \sum_{m=1}^{M} \gamma_m PC_{im} + \sum_{p=1}^{P} \gamma_p C_{ip} + u_i + \epsilon_{ik} \qquad (1)$$

Where $Y_{ij}$ is the normalized expression value for gene $j$ in sample $i$, $Y_{ik}$ is the normalized expression value for gene $k$ in sample $i$, $\beta_k$ is the effect size (fixed effect) of gene $k$, $PC_{im}$ is the value of the $m^{th}$ genotype principal component for the individual associated with sample $i$, $\gamma_m$ is the effect size of the $m^{th}$ genotype principal component, $M$ is the number of genotype principal components used ($M = 20$), $C_{ip}$ is the covariate of the $p^{th}$ covariate for sample $i$, $\gamma_p$ the effect size of the $p^{th}$ covariate, $P$ is the number of covariates used ($P = 5$), $u_i$ is a vector of random effects for the individual associated with sample $i$ defined from the kinship matrix, and $\epsilon_{ik}$ is the error term for individual $i$ at gene $k$.

**Construction of the genome-wide gene co-expression network (GN).** We created the genome-wide co-expression network

(GN) using the *graph_from_data_frame* function from the *igraph* R package[26] by assigning the 16,110 expressed genes as nodes, the 3,533,609 co-expressed genes (Bonferroni-corrected *p*-value < 0.05 and Effect Size > 0) as edges. We extracted the genome-wide degree of each gene using the *degree* function on the GN.

To identify gene co-expression network modules (GNMs), we applied the unsupervised Leiden community detection algorithm (from the *igraph* R package[26]) to the GN. We optimized module detection by analyzing 1,700 combinations of three parameters: resolution (range: 0–5), beta (range: 0–0.1), and n_iterations (range 5–50). For each combination of parameters, we permuted the nodes to confirm the Leiden algorithm was clustering co-expressed genes better than NULL background. We calculated modularity, fractions of genes in major GNMs (membership >100) and the number of modules for each combination of parameters. We selected the modules obtained by resolution = 2.25, beta = 0.05, n_iterations = 45, which exhibited a decent modularity (Q = 0.41) and a high fraction of genes captured by major GNMs (91.8%). Under these parameters, genes within the same GNM were significantly more co-expressed than genes randomly connected through permutation (*P*-value = 0). As there is no consensus on modularity thresholds or network validation methods[90], we used downstream analyses, such as co-expression and gene set enrichment for validation. For each gene, we calculated its intra-modular degree (the number of co-expressed genes within the module), using the same functions described above, for each GNM independently, excluding inter-module edges (Supplementary Data 4).

**Gene module identification using weighted gene correlation network analysis (WGCNA).** To benchmark our gene module detection approach (Supplementary Note 1), we processed the 213 RNA-seq samples using the standard workflow (https://horvath.genetics.ucla.edu/html/CoexpressionNetwork/Rpackages/WGCNA/Tutorials/) for the WGCNA R package[6], which cannot account for covariates and kinship. We used the WGCNA *corPvalueStudent* function to calculate the associations with the 24 covariates used in the co-accessibility LMM, the estimated formative fraction (Supplementary Fig. 2A), and binary annotations for samples from 6 families with 5 or more individuals (Supplementary Fig. 3). We then compared our 13 GNMs with the 17 WGCNA modules by calculating the enrichment of the 7 cell state gene sets (Fig. 1J, Supplementary Fig. 3).

**PCA and UMAP analyses of GNMs.** To assess whether genes within a GNM had similar expression profiles across hiPSCs, we performed PCA and UMAP analyses on the 213 RNA-seq samples. We first identified the most interconnected genes within each GNM by ranking the intramodular degree for each GNM independently. Biological networks are scale-free[27] and follow the Pareto Principle[28], which states that 20% of the nodes are responsible for 80% of a network's connectivity, therefore we defined intramodular Pareto genes as the top 20% interconnected genes within each GNM. We performed a PC analysis on the inverse normal transformed TPM expression of the intramodular Pareto genes in the 13 major GNMs (membership >100 genes) with the *prcomp* function in base *R*. We performed UMAP dimensionality reduction using the *umap* function from the *umap* R package.

**Calculation of GNM score.** For each of the 213 RNA-seq samples we calculated a GNM score for each of the 13 GNMs by summing the inverse normal transformed TPM expression of the corresponding GNM-specific Pareto genes (i.e., each RNA-seq sample had 13 GNM scores).

**Functional characterization: gene co-expression network modules**
**Calculation of GNM co-expression enrichment.** To assess whether genes within each GNM were more co-expressed with each other

across the 213 hiPSC lines than with genes in different GNMs, we enumerated the number of co-expressed genes shared between all pairwise combinations of GNMs. We then performed a Fisher's Exact test to calculate the enrichment of genes showing co-expression within each GNM, using the genes co-expressed between each set of paired GNMs as background.

**GNM enrichment in stem cell states.** We determined if the GNMs were enriched for expressing marker genes from three published studies describing 7 hiPSC subpopulations representing different pluripotency cell states including: (1) 199 8-cell like cell (8CLC)-associated genes from Mazid et al.[3]; (2) 123 primitive endoderm (PrE)-associated genes, 248 epiblast-associated genes, 175 trophectoderm genes; 96 Naïve-associated PSCs genes from Stirparo et al.[29]; and (3) 266 EPE-associated genes and 452 general population-associated genes that we identified by reprocessing data from Lau et al.[1] (See *Cellular deconvolution of pluripotency states using gene expression signatures*). We performed a Fisher's Exact test to calculate the enrichment of hiPSC subpopulation or cell state associated genes in each GNM, using the genes in the remaining 12 major GNMs as background.

**Cellular deconvolution of pluripotency states using ATAC-seq peak accessibility signatures**
We downloaded 6 ATAC-seq samples that were performed on human embryonic stem cells (hESCs) that had been sorted for the formative (GCTM-2$^{high}$CD9$^{high}$EPCAM$^{high}$) and the primed (GCTM-2$^{mid}$-CD9$^{mid}$) cell states from GEO (GSE147338)[1]. We aligned and filtered the FASTQ files, with STAR 2.5.0a[72] as described in the *Sequencing of ATAC-seq* section. MACS2 v.2.2.7 was used to establish a reference set of narrow peaks on the 6 ATAC-seq samples simultaneously, using the parameters; *-f BAMPE -g hs -t --nomodel --shift −100 --extsize 200 −narrow*. For each of the 6 individual ATAC-seq samples, read counts in the 193,147 reference peaks were obtained using *featureCounts* in Subread package v1.5.0[85]. We performed differential accessibility analysis on the counts using DESeq2 (v.1.34) *R* package[88]. We used *bedtools intersect -r 0.25* to identify 938 formative-associated peaks that overlapped iPSCORE ATAC-seq peaks with a 25% reciprocal overlap (Supplementary Data 8). We used the ATAC-seq peaks that were differentially accessible in the GCTM-2$^{mid}$-CD9$^{mid}$ hiPSC subpopulation[1] to annotate 2,981 non-formative associated peaks using the same approach. We then deconvoluted the 150 ATAC-seq samples using the *CIBERSORT.R* script[21] with a signature matrix containing 200 of the most differentially accessible formative and primed peaks (Supplementary Data 7).

**ATAC-seq co-accessibility analyses**
We leveraged our newly generated 150 ATAC-seq samples to profile co-accessibility of open chromatin across the hiPSC epigenome and discovery regulatory networks that are active in different pluripotency cell states.

**Identifying co-accessible peaks.** Since pairwise calculations for all 132,225 autosomal ATAC-seq peaks would have been computationally intensive, requiring ~8.74 × 10⁹ tests, we focused our analyses on the 56,978 reference peaks (see above: *Identification of 56,978 autosomal reference peaks*) which reduced the number of tests to 1.62 × 10⁹. To identify pairwise correlations of accessibility between the 56,978 peaks across the 150 ATAC-seq samples, we performed a genome-wide analysis using a Linear Mixed-effects Model (LMM) with the *lmekin* function in the *coxme* R package v.2.2-17[24], which incorporates a kinship matrix to control for random effects from genetic relatedness. The following covariates were included: (1) sex; (2) age; (3) hiPSC passage number; (4) number of reads passing filters; (5) the mean fragment size; (6) the number of broad peaks; (7) the ratio of 100 bp reads to 150 bp reads; (8) 20 genotype principal components to account

for global ancestry (see above: Covariates for co-expression and co-accessibility analyses).

**Formula #2:**

$$X_{ij} = \beta_k X_{ik} + \sum_{m=1}^{M} \gamma_m PC_{im} + \sum_{p=1}^{P} \gamma_p C_{ip} + u_i + \epsilon_{ik} \qquad (2)$$

Specifically, we utilized inverse normal transformed TMMs across the 150 ATAC-seq samples for each of the 56,978 peaks. In Formula #2 co-accessibility, $X_{ij}$ is the normalized accessibility value for peak $j$ in sample $i$, $X_{ik}$ is the normalized accessibility value for peak $k$ in sample $j$, $\beta_k$ is the effect size (fixed effect) of peak $k$ and the remaining terms were consistent with co-expression variable. Of the 56,978 reference peaks, 47,761 were present in the 13 major RNMs. In total, we identified 8,696,814 pairs of co-accessible peaks ($P$-value $< 5 \times 10^{-8}$, Effect Size $> 0$).

**Construction of the genome-wide regulatory co-accessibility network (RN).** We created the genome-wide co-accessibility network (RN) using the *graph_from_data_frame* function from the *igraph* R package[26] by assigning the 56,978 accessible peaks as nodes, the 8,696,814 co-accessible peaks ($P$-value $< 5 \times 10^{-8}$, Effect Size $> 0$) as edges. We extracted the genome-wide degree of each peak using the *degree* function on the RN. To find the optimal Leiden community detection model, we followed the same approach as described in the *Construction of the Genome-wide Gene Co-expression Network* section. We selected modules obtained from the model using the following parameters; resolution = 2.5, beta = 0.09, n_iterations = 30, which had a modularity of 0.36 and 47,761 peaks (83.8%) in 13 major RNMs (membership ≥ 500 ATAC-seq peaks). For each combination of parameters, we permuted the nodes to confirm the Leiden algorithm was clustering co-expressed genes better than the null background. For each peak, we calculated its intramodular degree (the number of co-accessible peaks within the module), using the same functions described above, excluding intermodule edges and considering each RNM independently (Supplementary Data 8).

**PCA and UMAP analyses of RNMs.** To assess whether ATAC-seq peaks within an RNM had similar accessibility profiles across hiPSCs, we performed PCA and UMAP analyses on the 150 ATAC-seq samples. We first identified the most interconnected ATAC-seq peaks within each RNM by ranking the intramodular degree for each RNM independently. We performed a PC analysis on the inverse normal transformed TMM of the peaks with the top 10% intramodular degree (top 10%) from the 13 major RNMs (membership ≥ 500 peaks) with the *prcomp* function in base R. We performed UMAP dimensionality reduction using the *umap* function from the *umap* R package.

**Calculation of ATAC-seq peak co-accessibility enrichment.** To assess whether peaks within each RNM were more co-accessible with each other than peaks in different RNMs, we enumerated the number of co-accessible peaks shared between all pairwise RNMs. We then performed a Fisher's Exact test to calculate the enrichment of co-accessibility between RNM pairs.

**Calculation of RNM score.** For each of the 150 ATAC-seq samples we calculated an RNM score for each of the 13 RNMs by summing the inverse normal transformed TPM matrix of the corresponding RNM-specific Pareto (top 20% intramodular degree) peaks (i.e., each ATAC-seq sample had 13 RNM scores).

**Correlation of GNMs and RNMs**
We examined the correlation between ATAC-seq peak co-accessibility network modules and gene co-expression network modules.

**Identifying associations between gene and regulatory networks.** The GNMs and RNMs were identified using overlapping hiPSC lines from the iPSCORE collection. However, the RNA-seq data and ATAC-seq data were generated from different passages of the hiPSCs cultured under different experimental conditions. The RNA-seq data was generated from earlier passage ROCK inhibitor-naïve hiPSCs and the ATAC-seq data was generated from later passage hiPSCs after culturing with ROCK inhibitor. Therefore, to annotate peaks with putative gene targets we: (1) identified genes expressed after culturing with 3 hiPSC lines with ROCK inhibitor (see above: *Generation of RNA-seq for 3 hiPSC lines expanded with ROCK inhibitor*), and then (2) annotated each peak with a single gene (distance <100 kb and highest expressed gene). Specifically, to identify candidate target genes for the 47,761 ATAC-seq peaks in the 13 major RNMs, we generated a bed file of the TSSs for the 16,110 autosomal genes expressed (TPM > 1 in at least one of the nine samples (triplicates of each hiPSC line) cultured with 5 μM ROCK inhibitor and performed *bedtools closest* to identify the closest TSS within 100 kb of each ATAC-seq peak. For ATAC-seq peaks that overlapped the TSSs of multiple genes, we calculated the maximum TPM expression across all 3 hiPSC lines cultured with ROCK inhibitor and annotated the ATAC-peak with the gene with the maximum expression. Finally, to identify associations between the GNMs and RNMs we only used genes: (1) expressed in both ROCK inhibitor-naïve hiPSCs and ROCK inhibitor-exposed hiPSCs, (2) in one of the 13 major GNMs, and (3) annotated as a putative target in one of the 13 major RNMs. In total, 12,078 unique genes corresponding to 32,327 peaks were used for the association test (Supplementary Data 8). We calculated the number of genes in common between all GNM-RNM pairwise combinations and performed Fisher's Exact tests to calculate enrichments.

**Functional annotation of hiPSC ATAC-seq peaks**
To functionally characterize the hiPSC epigenome, we annotated the 47,761 ATAC-seq peaks in the 13 major RNMs with three epigenetic annotations: (1) hiPSC-specific chromatin states, (2) TF binding, (3) pluripotency cell state.

**Collapsing into 5 hiPSC chromatin states.** Using the single chromatin state annotation (described above in Identification of 56,978 autosomal reference peaks), we binned the 12 chromatin states into 5 collapsed states by molecular similarities (Supplementary Fig. 6B). "Active promoters" (TssA) were not collapsed, the collapsed "Enhancer" annotation consisted of peaks in enhancers (Enh), genic enhancers (EnhG), and flanking active promoters (TssAFlnk), "Bivalent Chromatin" consisted of bivalent promoters (TssBiv), bivalent enhancers (EnhBiv), and regions flanking bivalent chromatin (BivFlnk), "Transcription" consists of strong (Tx) and weak (TxWk) transcription, and flanking transcription (TxFlnk), and "Repressed Polycomb" consists of both polycomb states (ReprPC, ReprPCWk, Supplementary Data 8).

**Prediction of transcription factor binding with TOBIAS.** The TOBIAS algorithm[35] leverages distribution of reads across the genome for a given sample, therefore to profile TF occupancy, we ran TOBIAS to predict binding at 187 motifs across all 136,333 ATAC-seq peaks (GEO: GSE203377, Supplementary Data 11). First, we identified 187 transcription factors with experimentally validated high-confidence motifs (Quality A and B) included in the HOCOMOCO V 11 collection[36] that were expressed (TPM > 1 in >20% samples) in the 213 hiPSC lines. We used samtools 1.9 to merge, sort and index the 34 BAM files of the reference hiPSC samples (see above: *ATAC-seq Peak Calling and Quality Control*). We then ran *TOBIAS ATACorrect* on the merged reference BAM file to correct for cut site biases introduced by the Tn5 transposase within the 136,333 ATAC-seq peaks, using the following parameters: --genome hg19 fasta (http://hgdownload.soe.ucsc.edu/goldenPath/hg19/bigZips/) and --blacklist hg19-blacklist.v2.bed

(http://github.com/Boyle-Lab/Blacklist/tree/hg19-blacklist.v2.bed.gz). Next, we calculated footprints scores with TOBIAS *ScoreBigwig*, using the narrowPeak file with the 136,333 ATAC-seq peaks (MACS2 score >100) for −regions. Finally, to identify the predicted transcription factor binding sites, we ran *TOBIAS BINDetect* with all 187 expressed TFs, using hg19 fasta file and narrowPeak file as the genome and regions, respectively. TOBIAS predicted a total of 2,349,030 TFBSs across all 187 motifs on 49,070 ATAC-seq peaks which represented 37.1% of the 132,225 peaks on autosomes. We annotated the 14,208 ATAC-seq peaks without a TFBS association as "Not Bound".

**Validation of predicted TF binding sites using experimental TF ChIP-seq data.** To validate the TOBIAS TF binding predictions, we evaluated the concordance with binding profiles that have been experimentally validated via TF ChIP-seq[37,41,70] (Supplementary Data 9). We obtained peaks from 19 sets of transcription factor ChIP-seq peak files for 18 TFs in H1 embryonic stem cells from ENCODE[37,41,70] (Supplementary Data 9). The REST TF had two independent ChIP-seq experiments, which we merged using *bedtools merge*. We intersected the TOBIAS predicted TF binding sites for the 18 TFs with corresponding TF ChIP-seq peaks using *bedtools intersect -loj*. We performed a Fisher's Exact test to calculate whether TOBIAS predicted TF binding sites were enriched in corresponding TF ChIP-seq peaks (Supplementary Fig. 7).

**Collapsing into 92 transcription factor groups.** Many different transcription factors have very similar binding motifs, and TOBIAS often predicted that TFs with similar motifs bound at the same site. For example, NANOG.0.A and PO5F1.0.A are the same length and have nearly identical position weight matrices (PWMs, Supplementary Fig. 8B); hence, the identity of the exact TF bound could not be resolved. To reduce the effects of motif sequence similarity, we collapsed the 187 motifs into TF groups by hierarchically clustering the Euclidean distances based on the overlap of bound sites generated by TOBIAS thresholds using *cutree* (Supplementary Data 10). We selected a 0.75 threshold and obtained 92 TF groups, of which 49 consisted of a single motif, 24 were composed of motifs from the same TF family ($n = 73$ TFs) and 19 consisted of multiple TFs from different families ($n = 65$ TFs), hereby referred to as complexes (Supplementary Fig. 8, Supplementary Data 10). The 187 TF motifs and their corresponding collapsed TF groups are in Supplementary Data 10.

**Functional characterization of RNMs**
**Epigenetic feature enrichment within hiPSC regulatory networks.** To molecularly characterize hiPSC regulatory networks, for each of the 13 RNMs we calculated enrichment of the formative and primed cell states, 5 collapsed chromatin states and 93 TF groups (including "Not Bound"). We performed a Fisher's Exact test to calculate enrichment for each epigenetic annotation in the Pareto peaks for each major RNM by using the Pareto peaks for the 12 remaining major RNMs as background (Supplementary Data 12). We considered enrichments with a Bonferroni-corrected $p$-value < 0.05 significant.

**hiPSC regulatory networks conserved in fetal cell types**
**Annotating hiPSC ATAC-seq peaks with REs with fetal cell type-specific peaks.** To identify hiPSC ATAC-seq peaks that correspond to active chromatin in fetal tissue, we integrated single cell ATAC-seq peaks from 54 fetal cell types in the Descartes; Human Chromatin Accessibility During Development Atlas[53]. To obtain cell type-specific peaks, we used the $Z$-score corrected single cell ATAC-seq peaks for the 54 fetal cell types ($n = $ ~9500 peaks on autosomes per cell type) (https://atlas.brotmanbaty.org/bbi/human-chromatin-during-development/). We performed *bedtools intersect -f -r 0.25* on the 47,761

reference ATAC-seq peaks in the 13 major RNMs and identified 11,830 hiPSC ATAC-seq peaks with a 25% reciprocal overlap with at least one fetal cell type-specific peak. To calculate the enrichment of each RNM for each of the 54 fetal cell types, we performed Fisher's Exact tests of the overlap with these 11,830 hiPSC ATAC-seq peaks, using the remaining 12 RNMs as background (Supplementary Data 13). To calculate the enrichment of hiPSC TFBSs in the 54 fetal cell type-specific ATAC-seq peaks, we performed Fisher's Exact tests with *bedtools fisher* (Supplementary Data 14). For background, we merged the bed files for all 54 fetal cell type-specific peaks and calculated the number of base pairs in the merged peaks for each chromosome.

**Allele-specific chromatin accessibility (ASCA) analyses**
To identify regulatory variants that impact transcription factor binding we performed allele-specific chromatin accessibility (ASCA) using the 150 ATAC-seq samples from the 133 iPSCORE individuals.

**Calculation of allele-specific chromatin accessibility (ASCA).** A VCF file from WGS data of 273 iPSCORE individuals[68] was obtained from dbGaP (phs001325). We extracted SNPs in the 47,761 ATAC-seq peaks in the 13 major RNMs: (1) with minor allele frequency > 0.05 in all 273 iPSCORE individuals[68] using bcftools view with parameters: --*types snps*, --*f PASS*, -*q 0.05:minor*; and (2) in Hardy-Weinberg equilibrium ($p > 1 \times 10^{-6}$) in the 133 iPSCORE individuals with ATAC-seq data using *vcftools --hwe 0.000001*. To increase the power to detect allele-specific chromatin accessibility, we performed phasing on these variants using the Michigan Imputation Server with the 1000 Genomes Phase 3 as a reference panel and converted them into the hdf5 format using snp2hd5 in WASP (version 0.2.2), as suggested by the original developers. We realigned the BAM files after WASP correction and applied the same filters as described above in the *Sequencing of ATAC-seq* section, except for removing duplicates. Specifically, we excluded poorly mapped reads (MAPQ < 20%), and reads less than 38 bp and greater than 2000 bp with samtools view. We then identified allele mapping bias at heterozygous sites in each sample using the WASP mapping pipeline with default parameters[91] and duplicates were removed in a non-biased manner using the rmdup_pe.py script in WASP. Coverage of bi-allelic heterozygous variants was calculated using GATK ASEReadCounter (version 3.4-46) with parameters: -*overlap COUNT_FRAGMENTS_REQUIRE_SAME_BASE*, -*U ALLOW_N_CIGAR_READ*[92]. To maximize detection of ASCA, we aggregated allele read counts for each SNP across all heterozygous individuals and required: (1) a minimum of 5 heterozygous individuals were tested, and (2) at least 50 total reads mapped to the position, and (3) at least 10 reads mapped to each of the reference and alternate alleles. This resulted in 105,055 bi-allelic SNPs in 35,614 accessible ATAC-seq peaks used for ASCA analysis. We annotated 104,938 SNPs with their corresponding rsid from the gnomAD database (v2)[93], and used the chromosome, position, reference, and alternate allele to annotate the 117 SNPs that were not in gnomAD. ASCA was determined using a two-sided binomial test with equal probability (probability = 0.5) for each allele being accessible. $P$-values were corrected using Benjamini-Hochberg. SNPs with an adjusted $p$-value < 0.05 were considered to display allele-specific chromatin accessibility (ASCA-SNPs) (Supplementary Data 15). We annotated the 6,323 ASCA SNPs with the RNMs associated with the ATAC-seq peak and the overlapping TF group(s). To identify RNMs and TF groups with enriched allelic imbalance fractions (AIFs), we performed a one-sided Mann-Whitney $U$ test, using the 12 other RNMs or 22 significant TF groups (Fig. 5C) as background. To calculate the enrichment of ASCA SNPs in predicted TFBSs, we performed a Fisher's Exact test using bedtools fisher.

**Reporting summary**
Further information on research design is available in the Nature Portfolio Reporting Summary linked to this article.

## Data availability

The 222 iPSCORE hiPSC lines are available through WiCell Research Institute (www.wicell.org; NHLBI Next Gen Collection). The previously published FASTQ sequencing data for bulk RNA-seq and WGS have been deposited in dbGaP under the accession codes "phs000924 [https://www.ncbi.nlm.nih.gov/projects/gap/cgi-bin/study.cgi?study_id=phs000924.v4.p1]" and "phs001325 [https://www.ncbi.nlm.nih.gov/projects/gap/cgi-bin/study.cgi?study_id=phs001325.v5.p1]", respectively. The bulk ATAC-seq FASTQ files generated in this study have been deposited in Gene Expression Omnibus (GEO) under the accession code "GSE203377". The processed TOBIAS transcription factor binding predictions, the TMM-normalized counts, and the reference narrow peak file have been deposited on "Figshare [https://figshare.com/projects/iPSC_coaccessibility/136585]". Source data are provided with this paper.

## Code availability

All scripts developed to perform this study are available in "GitHub [https://github.com/tdarthur40/ipsc_coaccessibility]" and citable through "Zenodo [https://zenodo.org/records/10481265]"[94].

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

## Acknowledgements

This work was supported by a California Institute for Regenerative Medicine grant GC1R-06673-B (K.A.F.), NIH grants HG008118 (K.A.F.), HL107442 (K.A.F.) and HG011558 (K.A.F.), F31DK131867 (J.P.N.), F31HL158198 (T.D.A.) and T15LM011271 (J.P.N. and T.D.A.), and by the Coordenação de Aperfeiçoamento de Pessoal de Nível Superior - Brasil (CAPES) - Finance Code 001 (N.S.S. and A.D.L.).

## Author contributions

T.D.A., W.W.Y.G., K.A.F., and iPSCORE consortium members conceived the study. T.D.A., J.P.N., M.D., H.M., and iPSCORE consortium members performed the computational analyses. A.D.C., N.S., and iPSCORE consortium members generated molecular data. K.A.F., A.D.L., and iPSCORE consortium members oversaw the study. T.D.A., M.F.P., and K.A.F. prepared the manuscript.

## Competing interests

W.W.Y.G. and K.A.F. are co-founders of Synthalogy Therapeutics. The remaining authors declare no competing interests.

## Additional information

## iPSCORE Consortium

Lana Ribeiro Aguiar[5], Angelo D. Arias[4], Timothy D. Arthur[1,2], Paola Benaglio[4], W. Travis Berggren[9], Juan Carlos Izpisua Belmonte[10], Victor Borja[5], Megan Cook[5], Matteo D'Antonio[2,5], Agnieszka D'Antonio-Chronowska[4], Christopher DeBoever[3], Kenneth E. Diffenderfer[9], Margaret K. R. Donovan[2,3], KathyJean Farnam[5], Kelly A. Frazer [4,5]✉, Kyohei Fujita[4], Melvin Garcia[5], Benjamin A. Henson[5], Olivier Harismendy[2], David Jakubosky[1,2], Kristen Jepsen[5], He Li[4], Hiroko Matsui[5], Naoki Nariai[4], Jennifer P. Nguyen[2,3], Daniel T. O'Connor[11], Jonathan Okubo[5], Athanasia D. Panopoulos[10], Fengwen Rao[11], Joaquin Reyna[5], Bianca M. Salgado[5], Erin N. Smith[4], Josh Sohmer[5], Shawn Yost[3] & William W. Young Greenwald[3]

[9]Stem Cell Core, Salk Institute for Biological Studies, La Jolla, CA 92037, USA. [10]Gene Expression Laboratory, Salk Institute for Biological Studies, La Jolla, CA 92037, USA. [11]Department of Medicine, University of California, San Diego, La Jolla, CA 92093, USA.

