## [Peer Review File · Nature Communications]

Complex regulatory networks influence pluripotent cell state transitions in human iPSCsREVIEWER COMMENTS

Reviewer #1 (Remarks to the Author):

The manuscript by Arthur and colleagues provides a detailed analysis and integration of transcriptional and chromatin data sets in a large cohort of human iPSC lines. The important advance is that they use network-scale construction of chromatin co-accessibility information to infer regulatory modules each comprising of hundreds of open chromatin regions, with the prediction that the coordinated behaviours of these modules are informative for understanding upstream and downstream regulatory events. In this way, by compartmentalising the complex data, it makes it easier for researchers to isolate properties and make predictions. The authors have done an impressive job in integrating a large number of data sets and distilling down some key features. They highlight several interesting properties in the manuscript, for instance independent modules that may be driven by specific sets of transcription factors. Overall, the study raises many interesting new hypotheses and predictions for future testing, and the data and analysis framework should provide a useful resource for the field and will be of broad interest.

I have a few specific comments that are aimed at further improving the manuscript:

I felt that at several points in the text, there are quite strong claims that are based on inferred associations and effects, but that lack direct testing. I suggest that some of these claims are toned down to better reflect the limits of the conclusions. As an example, in the abstract: “Genetic analyses identified regulatory variants that disrupted transcription factor binding” but this was not tested experimentally. Also, it would be more accurate to say that only a small number of genetic variants were associated with altered accessibility, as I think this is an important part of the message. I also think the statement that the findings “uncover novel pluripotency regulatory mechanisms” should be toned down to better reflect the main findings of the work, even it is just to say that the work leads to new predictions about potential pluripotency regulatory mechanisms.

In the introduction, the authors write: “However, network-based algorithms have not yet been used to analyze ATAC-seq data to identify co-accessible regulatory elements distributed across the genome.” It would be helpful for the reader if the authors could give further background here about what has been done previously in terms of network analysis of co-accessible regions (such as, DOIs: [10.1093/bib/bbaa120](https://doi.org/10.1093/bib/bbaa120); [10.1016/j.molcel.2018.06.044](https://doi.org/10.1016/j.molcel.2018.06.044); [10.1016/j.xgen.2022.100166](https://doi.org/10.1016/j.xgen.2022.100166); [10.1371/journal.pcbi.1009670](https://doi.org/10.1371/journal.pcbi.1009670); [10.1038/s41467-020-18638-8](https://doi.org/10.1038/s41467-020-18638-8)), and where the gaps are that the current study then addresses.

Figure 2G is an important panel as it uses marker genes from various cell types to understand and annotate the GNMs. But the panel contains a bit of an odd mix of samples. It’s not clear to me why these particular data sets were chosen? It might be better to present this panel in a more logical way, for example to separate out the in vivo from in vitro samples, also consider adding a trophectoderm or trophoblast data set to complement the epiblast and primitive endoderm data that are already there.

The conclusion: “Altogether, these findings show that the profiled hiPSCs were composed of subpopulations of naive to primed pluripotency states (Figure S1)” is inferred from changes in the network modules and lacks experimental validation. I suggest that the text is altered to present the conclusion more accurately.

On similar lines, the sentence: “These data indicate that culturing with a ROCK inhibitor shifts the composition of pluripotency cell states decreasing the number of cells in a primed-like state and increasing the number in the formative-like state.” Again, this is inferred and not directly shown and the data do not address whether the proportion of cells is changing. And it is unclear why the formative state is specifically highlighted here, when the GNM is also associated with naïve PSCs, PrE and conventional PSCs.

Following integration of ATAC peaks with chromatin states, it was especially interesting to see that about half of the ATAC peaks overlapped with active chromatin states. One prediction might be that constructing RNMs from these active-associated ATAC peaks could have some benefit, i.e. it might lead to more specific networks and integration with GNMs. If it is possible to check this then that would be very interesting to do; but if it is a lot of work, then instead perhaps just add to the discussion that this is one future outlook of the work that could be done to further improve the network predictions.

Several of the figures are complex and dense, and I think tweaking the presentation of some of them could improve readability. In particular, I found the ordering and abbreviations of the chromatin states in Figures 4 and 5 not that logical, and perhaps by changing the ordering and/or using other visual cues could make this easier to read.

Page 12, the sentence: “Here, we show that rs9350250 is associated with decreased accessibility of three primed-associated RNMs (21, 24, and 37), and increased accessibility of formative-associated RNMs 3 and 8 (Figure 7G)”. I don’t see the association with RNM21.

Also in Figure 7, is there a link between rs9350250 genotype and the expression levels of E2F3 (or other nearby genes)? It would be informative to comment on this.

[minor] In the sentence: “We analyzed the expression of 8,972 Pareto peaks (top 20% intra-RNM degree connectivity) and observed RNM specific clustering in the UMAP space (Figure 3C)”, I don’t think “expression” is the right word here, as you are talking about accessibility.

Reviewer #2 (Remarks to the Author):

The authors presented an unsupervised machine learning algorithm to discover genome-wide regulatory network modules (RNMs) and gene network module (GNMs) from ATAC-seq and RNA-seq datasets, and revealed that the GNMs and

RNMs were highly correlated with each other. Their findings uncovered novel pluripotency regulatory mechanisms and provided a resource for future stem cell research.

The manuscript can be further improved through addressing the following points:

1. The analysis considered in this paper was accomplished by applying some well established methods. This is a weak point for this paper.

The considered samples and validation cases are not challenging. The novelty and contribution of this paper are limited.

2. The authors constructed the gene co-expression network and chromatin co-accessibility network from RNA-seq data and

ATAC-seq data respectively. They used the same model (i.e., Linear Mixed effects Model, LMM) to construct these two networks and used the same model (i.e., Leiden community detection algorithm) to detect network modules. Why the co-accessibility network can reflect the regulatory relationships?

3. It will be better if the authors could provide more discussion about why is this particular method suitable for this particular computational problem (and type of data) from statistical and biological perspectives.

4. Is it possible to use both chromatin accessibility and gene expression data to construct regulatory networks? As there are many methods being proposed for gene regulatory network inference, the more accurate the constructed network, the better for downstream analysis.

Reviewer #3 (Remarks to the Author):

This paper presents accessibility and gene expression data and analysis of nearly 200 human iPSC lines (150 ATAC-seq and 213 RNA-seq) from the PSCORE consortium. The ATAC-seq is new to this study, the RNA-seq data was previously published. Using these data, the authors infer gene co-expression networks clustered into Gene Network Modules (GNMs) and also co-accessibility networks clustered into Regulatory Network Modules (RNMs). The paper characterizes the GNMs based on their enrichment of Gene Ontology and Msigdb genesets and overlap with existing genes associated with different states of pluripotency. The major finding from this analysis is that the hiPSCs capture a range of pluripotency cell states. The paper focuses more on the analysis of the ATAC-seq data and the inferred RNMs. The RNMs are analyzed based on overlap with ChromHMM states, enrichment in GNMs, overlap with existing sets of peaks representing different cell types or pluripotency cell states, and finally examination of allele-specific SNPs and their impact on accessibility. The overall resource generated by the paper, namely the 150 ATAC-seq profiles is very useful for the community, however, the overall paper is quite descriptive and it is unclear what the major novel findings of this work are. There are a number of observations

made but in many cases they are consistent with what is known or it is not apparent what the major significance of the observation is. Another major limitation is that the RNA-seq and ATAC-seq came from different lines which makes it difficult to more directly observe the impact of change in accessibility to expression. There is also no validation of the GNM or RNMs so it is hard to know how accurate these networks and modules are. More details follow:

Strengths

1. The paper generates a useful resource, ATAC-seq data for 150 samples from different 133 human iPSC lines.
2. Presents one approach to analyze and interpret the data to gain insight into the transcriptome and accessibility state of the induced pluripotency state.
3. The TF grouping analysis and the preference of different groups for different genomic region types was interesting (second last para of "Functional annotation of hiPSC ATAC-seq peaks").

Weaknesses

1. The paper is quite descriptive and it is unclear what the major novel findings of this work are. For example based on the GNM analysis, the authors conclude that the hiPSC lines represent a mix of different known pluripotency cell states. However, this seems to have been already known. A similar finding is stated with accessibility as well as that there is some overlap with fetal cell line networks. These all seem like general expected observations.
2. As such the integration of the GNM and RNMs or the gene expression and accessibility was quite limited. Everything is based on enrichment analysis, however this is likely because the data were collected on different cohorts so we are unable to see the impact of changes in accessibility, including the allele-specific binding disruption on expression.
3. For the GNM analysis, the authors decided to focus on Modules 14 and 32 however there were other modules that also exhibit the pluripotency-state specific signatures.
4. It was hard to see how the analysis and resource was achieving the biggest goal of the paper as stated in the title that this analysis was identifying Regulatory networks that define cell state transitions. There is overlap with known pluripotency cell states and with accessibility of fetal cell lines but it is unclear what the transitions are and what determines them.
5. I found the section "Integration of hiPSC ATAC-seq peak functional annotations" to be not very useful. It can be removed or integrated with some other section.
6. The Allele-specific SNP analysis is interesting, but the section ends with "Taken together, these data show that specific TF groups, chromatin states and the RNMs which they comprise show relative enrichment or depletion for ASCA-SNPs.". Isn't this obvious that there will be some modules with enrichment and some with depletion and some of them will have some overlap with other annotation categories?
7. The paper is very descriptive and there is little or no validation (computational or experimental) of the

results. It is hard to know how reliable the GNMs are and how much they agree with gene regulatory networks associated with the pluripotency state. Similarly, the RNM network could have been evaluated with Hi-C data but this is not considered. Also although a lot of analyses have been shown, it is difficult to see if there any generalizable concepts that could be used to identify key genes or interactions that could be experimentally validated. Much of what they propose that could be tested is very specific to the pluripotency state (e.g. the ROCK inhibitor analysis).

We thank the reviewers for their valuable feedback on our manuscript. We appreciate their expertise and constructive criticism, which have allowed us to enhance the quality and clarity of our work. Many of our responses are structured with a narrative followed by a table with text pasted from the revised manuscript. The references in the narratives are organized in the bibliography at the end of this document while the references within the table correspond to the reference numbers in the main text.

Reviewer #1 (Remarks to the Author):

The manuscript by Arthur and colleagues provides a detailed analysis and integration of transcriptional and chromatin data sets in a large cohort of human iPSC lines. The important advance is that they use network-scale construction of chromatin co-accessibility information to infer regulatory modules each comprising of hundreds of open chromatin regions, with the prediction that the coordinated behaviours of these modules are informative for understanding upstream and downstream regulatory events. In this way, by compartmentalising the complex data, it makes it easier for researchers to isolate properties and make predictions. The authors have done an impressive job in integrating a large number of data sets and distilling down some key features. They highlight several interesting properties in the manuscript, for instance independent modules that may be driven by specific sets of transcription factors. Overall, the study raises many interesting new hypotheses and predictions for future testing, and the data and analysis framework should provide a useful resource for the field and will be of broad interest.

We are pleased that the reviewer recognizes the scientific advancements of our work as well as the fact that our unique iPSC chromatin accessibility data set provides an important resource for the field and affords future opportunities for further examining gene regulation in human-induced stem cell states.

I have a few specific comments that are aimed at further improving the manuscript:

1. I felt that at several points in the text, there are quite strong claims that are based on inferred associations and effects, but that lack direct testing. I suggest that some of these claims are toned down to better reflect the limits of the conclusions. As an example, in the abstract: “Genetic analyses identified regulatory variants that disrupted transcription factor binding” but this was not tested experimentally. Also, it would be more accurate to say that only a small number of genetic variants were associated with altered accessibility, as I think this is an important part of the message. I also think the statement that the findings “uncover novel pluripotency regulatory mechanisms” should be toned down to better reflect the main findings of the work, even it is just to say that the work leads to new predictions about potential pluripotency regulatory mechanisms.

We thank the reviewer for these comments. We have addressed the reviewer’s concerns by both revising both the tone of the manuscript and by conducting additional analyses to support some of the claims. We have substantially changed the genetic analysis section. The original

manuscript tested variants that intersected predicted TFBSs and evaluated their effect on co-accessibility. The approach was conceptually complex and due to a lack of suitable datasets, there was no way to test our hypotheses, therefore we decided to simplify the analysis. We no longer discuss specific variants although interested readers could find variants with strong effects by looking in the supplemental tables. In the revised analysis, we tested 105,055 SNPs in peaks in the 13 RNMs for allele-specific chromatin accessibility. We characterized the 6,323 SNPs with exhibited ASCA, by examining which RNMs they were in, and which predicted TFBSs they overlapped. We calculated differential allelic imbalance between RNMs and TF groups, independently. Although we do not experimentally validate TFBS disruption, we believe that the validation of the TOBIAS predictions (Figure S8) are sufficient to support our toned back claims about our findings.

The table below provides detailed insights into these edits.

Section	Revised Text
Abstract	Genetic analyses identified thousands of regulatory variants that overlapped predicted transcription factor binding sites and were associated with chromatin accessibility in the hiPSCs.
Abstract	Our work captures the coordinated activity of tens of thousands of regulatory elements in hiPSCs and bins these elements into discrete functionally characterized regulatory networks, shows that regulatory elements in pluripotency networks harbor variants with large effects, and provides a rich resource for future pluripotent stem cell research.
Introduction Page 3	Understanding how pluripotent subpopulations vary across hiPSC lines under conventional culturing could improve their utility for studying developmental processes and regenerative medicine.
Results Characterization of allele-specific chromatin accessibility SNPs (ASCA-SNPs) Pages 10-11	Previous studies have shown that both genetic and epigenetic variation affect hiPSC phenotypes ⁵⁵⁻⁵⁷ , however, the underlying mechanisms are poorly understood. One likely mechanism through which genetic variation influences hiPSC phenotypes ⁵⁵⁻⁵⁷ is by affecting chromatin accessibility. We examined 105,055 SNPs (MAF ≥ 0.05 , HWE p-value $> 1 \times 10^{-6}$) present in 35,614 ATAC-seq peaks in the 13 major RNMs and determined that 6,323 displayed ASCA. To determine if specific RNMs harbored ASCA SNPs with large effects, we performed Mann-Whitney U tests on the allelic imbalance fraction (AIF) of ASCA SNPs in each of the 13 RNMs using the other 12 RNMs as background (Figure 5A). We observed that four RNMs 1, 2, 3, and 9 (all enriched for enhancers Figure 3B), contained ASCA SNPs with greater allelic imbalance (AIF) compared to the other RNMs (Figure 5A). Of note, RNMs 2 and 3 were associated with the formative state (Figure 3A) and represent the NANOG-OCT4 and TEAD-mediated (Figure 3C) regulatory networks, respectively. While RNMs 1 and 9 were depleted for formative state regulatory elements, they

	represented NANOG-OCT4-mediated regulatory networks most likely in different pluripotent states. Our findings show that compared with the other regulatory networks active in hiPSCs, the regulatory elements in the pluripotency-associated networks harbor genetic variants that exert large effects on chromatin accessibility. Genetic variation can influence chromatin accessibility by affecting the binding specificity of TF motifs in regulatory elements⁵⁸⁻⁶¹. We examined if any specific group of TF groups were enriched for having ASCA SNPs overlapping their binding sites. Of the 6,323 ASCA SNPs, 1,933 (30.6%) overlapped at least one TFBS, while the majority did not overlap any TFBSs (n = 4,299, Figure 5B). We calculated the enrichment of the ASCA SNPs in TFBSs by performing a Fisher's Exact test for all 92 TF groups (Figure 5C). Regulatory elements predicted to bind 23 TF groups were enriched with ASCA SNPs (adjusted P-value < 0.05 and Odds Ratio > 1). These 23 groups included many pluripotency-associated TFs, such as NANOG-OCT4, TEAD Family, POU2F2, and SOX-LEF1 Complex (Figure 5C). We examined the allelic imbalance fraction (AIF) of the ASCA SNPs in each of the 23 TF groups and observed that NANOG-OCT4 contained ASCA SNPs with significantly higher AIF indicating that they had large effects on chromatin accessibility (Figure 5D). Six other TF groups (AR-FOXJ3, TEAD Family, CTCF Family, SALL4-ZIC1 Complex, and SOX-LEF1 Complex) exhibited increased AIF, albeit not significantly. Taken together, these results show that pluripotency-associated TF binding sites and regulatory networks are enriched with genetic variants that have large effects on chromatin accessibility. While previous studies have shown that pluripotency TFs have a remarkably high degree of evolutionary constraint⁶², our findings show that the regulatory elements to which they bind have a high amount of variability suggesting that they could play an important role in the observed pluripotent state differences between hiPSCs.
Figure 5A-D	Shows that RNMs and TFBS are differentially affected by allelic imbalance.
Methods Calculation of Allele-Specific Chromatin Accessibility (ASCA) Page 25	A VCF file from WGS data of 273 iPSCORE individuals⁶⁵ was obtained from dbGaP (phs001325). We extracted SNPs in the 47,761 ATAC-seq peaks in the 13 major RNMs: 1) with minor allele frequency > 0.05 in all 273 iPSCORE individuals⁶⁵ using bcftools view with parameters: <code>--types snps, --f PASS, -q 0.05:minor</code>; and 2) in Hardy-Weinberg equilibrium ($p > 1 \times 10^{-6}$) in the 133 iPSCORE individuals with ATAC-seq data using vcftools <code>--hwe 0.000001</code>. To increase the power to detect allele-specific chromatin accessibility, we performed phasing on these variants using the Michigan Imputation Server with the 1000 Genomes Phase 3 as a reference panel and converted them into the hdf5 format using snp2hd5 in WASP, as suggested by the original developers. We realigned the BAM files after WASP correction and applied the same filters as described above in the Sequencing of ATAC-seq section, except for

removing duplicates. Specifically, we excluded poorly mapped reads (MAPQ < 20%), and reads less than 38bp and greater than 2000bp with samtools view. We then identified allele mapping bias at heterozygous sites in each sample using the WASP mapping pipeline with default parameters⁸⁹ and duplicates were removed in a non-biased manner using the rmdup_pe.py script in WASP. Coverage of bi-allelic heterozygous variants was calculated using GATK ASEReadCounter with parameters: *-overlap COUNT_FRAGMENTS_REQUIRE_SAME_BASE, -U ALLOW_N_CIGAR_READ*⁹⁰. To maximize detection of ASCA, we aggregated allele read counts for each SNP across all heterozygous individuals and required: 1) a minimum of 5 heterozygous individuals were tested, and 2) at least 50 total reads mapped to the position, and 3) at least 10 reads mapped to each of the reference and alternate alleles. This resulted in 105,055 bi-allelic SNPs in 35,614 accessible ATAC-seq peaks used for ASCA analysis. We annotated 104,938 SNPs with their corresponding rsid from the gnomAD database (v2)⁹¹, and used the chromosome, position, reference, and alternate allele to annotate the 117 SNPs that were not in gnomAD. ASCA was determined using a two-sided binomial test with equal probability (probability = 0.5) for each allele being accessible. P-values were corrected using Benjamini-Hochberg. SNPs with an adjusted p-value < 0.05 were considered to display allele-specific chromatin accessibility (ASCA-SNPs) (Table S15). We annotated the 6,323 ASCA SNPs with the RNMs associated with the ATAC-seq peak and the overlapping TF group(s). To identify RNMs and TF groups with enriched allelic imbalance fractions (AIFs), we performed a Mann-Whitney U test, using the 12 other RNMs or 91 TF groups as background. To calculate the enrichment of ASCA SNPs in predicted TFBSs, we performed a Fisher's Exact test using bedtools fisher.

2. In the introduction, the authors write: “However, network-based algorithms have not yet been used to analyze ATAC-seq data to identify co-accessible regulatory elements distributed across the genome.” It would be helpful for the reader if the authors could give further background here about what has been done previously in terms of network analysis of co-accessible regions (such as, DOIs: 10.1093/bib/bbaa120; 10.1016/j.molcel.2018.06.044; 10.1016/j.xgen.2022.100166; 10.1371/journal.pcbi.1009670; 10.1038/s41467-020-18638-8), and where the gaps are that the current study then addresses.

The reviewer provides an excellent suggestion to highlight some of the work that has previously been done to characterize cis-regulatory interactions. To address the reviewer’s comment, we provided more context on the current state of the field in the introduction. See the edit that we made to the introduction section of the manuscript below.

Section	Edit
Introduction	But thus far, applications of these algorithms to analyze ATAC-seq data have been more limited. For example, studies have used paired

Pages 3-4	bulk ATAC-seq and RNA-seq to examine how regulatory elements impact gene expression networks under dynamic conditions, including comparing cells at baseline and post-stimulation⁹ or undergoing differentiation¹⁰. There has also been considerable effort studying patterns of co-accessibility in single-cell ATAC-seq under static conditions to explore local cis interactions (~500kb) between regulatory elements both in single tissues¹¹ and simultaneously across multiple tissues¹²; and a recent study analyzed paired single-cell ATAC-seq and RNA-seq data to study local cis-regulation of gene expression under environmental stimuli¹³. However, genes that are members of the same biological pathways are frequently encoded on different chromosomes, and there have been limited studies aimed at examining co-accessible ATAC-seq peaks distributed across the human genome to understand regulatory processes underlying the co-expression of gene modules. For example, in hiPSCs, the maintenance of pluripotency relies on the expression of pluripotency-related transcription factors, such as NANOG, OCT4, and SOX2, which create global epigenomic regulatory networks that enable self-renewal through the repression of developmental genes¹⁴, regulation of cell cycle transitions^{1,15}, and promotion of autoregulation^{16,17}.
---

3. Figure 2G is an important panel as it uses marker genes from various cell types to understand and annotate the GNMs. But the panel contains a bit of an odd mix of samples. It's not clear to me why these particular data sets were chosen? It might be better to present this panel in a more logical way, for example to separate out the in vivo from in vitro samples, also consider adding a trophoctoderm or trophoblast data set to complement the epiblast and primitive endoderm data that are already there.

Thank you for this suggestion. To address this comment, we updated the figure by restricting the gene sets from three sources (Lau et al., 2020; Mazid et al., 2022; Stirparo et al., 2018). We color-coded the gene set names to clarify which were obtained from *ex vivo* (in red) or *in vitro* (in blue) samples. The current panel is now Figure 1J and we annotated a new supplemental figure (Figure S4) in the same manner.

4. The conclusion: "Altogether, these findings show that the profiled hiPSCs were composed of subpopulations of naive to primed pluripotency states (Figure S1)" is inferred from changes in the network modules and lacks experimental validation. I suggest that the text is altered to present the conclusion more accurately.

The reviewer expressed a valid concern about our claim that the iPSC lines were composed of varying proportions of pluripotent subpopulations. We felt that this was an important message of the paper and hence conducted additional analysis to validate this claim. We outline these modifications in detail below and hope that they satisfy the reviewer's concerns.

We set out to determine if the relative proportions of the pluripotent cell states varied across 213 hiPSC lines with bulk RNA-seq data using cellular deconvolution. Deconvolution typically uses gene signatures obtained from cell type clusters in single-cell data; however, due to the high similarity between hiPSC pluripotency states, they don't separate into distinct clusters. Therefore, we generated gene signatures using bulk RNA-seq data for FACS-sorted formative and paired unsorted (e.g. primed) cells downloaded from (GSE119324) (Lau et al., 2020), for which co-author Dr. Martin F. Pera is the corresponding author. We applied the CIBERSORT deconvolution algorithm with a signature matrix containing the 100 most differentially expressed genes between the two subpopulations and observed that the estimated fraction of cells in the formative state exhibited a wide range across the 213 hiPSC lines. We examined the expression of formative-specific and primed-specific genes in hiPSCs with the highest and lowest estimated proportions of formative cells and observed notable expression differences of key regulators of pluripotency, such as DUSP5, LEFTY1, FST, and FZD5. Taken together, these results show that remarkable variation in pluripotency cell state composition exists across the 213 hiPSC lines under conventional culture conditions.

We next sought to determine if the GNMs were enriched for different pluripotency cell states. We first examined whether certain GNMs were associated with the estimated fraction of formative state cells. For each of the 213 RNA-seq samples, we calculated a GNM score for each of the 13 GNMs by summing the inverse normal transformed TPM expression of the corresponding GNM-specific Pareto genes (each RNA-seq sample had 13 GNM scores). To identify associations between the estimated fraction of formative state cells and GNM scores across the 213 hiPSC lines, we ran a linear model and observed that GNM 5 was positively associated with the estimated proportion of formative state cells, while GNM 10 exhibited a strong negative association.

We also downloaded ATAC-seq samples (GSE147338)(Lau et al., 2020). We used the most differentially accessible peaks between the formative and primed populations to perform cellular deconvolution using CIBERSORT on the 150 iPSCORE ATAC-seq samples. Only a small fraction of cells (range 0-36.6%) in each iPSCORE ATAC-seq sample were estimated to be in the formative state. We calculated RNM scores by summing the inverse normal transformed accessibility of 9,545 Pareto peaks (top 20% intramodular connectivity) for each ATAC-seq sample (each sample had 13 RNM scores). We then ran a linear model to test for associations between the estimated proportion of formative state cells and RNM scores. We showed that

RNMs 2, 3, and 8 are associated with the estimated formative proportion. Finally, we examined whether genome-wide ATAC-seq peak co-accessibility is associated with the coordinated gene expression in different pluripotency states. We annotated 32,327 ATAC-seq peaks in the 13 RNMs with 12,078 neighboring candidate target genes and then performed a Fisher's Exact test to calculate enrichments of RNMs in GNMs. We found that all RNMs had an association with at least one GNM. For example, five RNMs (2, 3, 5, 8, and 13) were positively enriched for the formative GNM 5. Altogether, we show that the RNMs are associated with different pluripotency states and demonstrate considerable variability in their proportions between hiPSC lines.

These new analyses have been added to the text as follows:

Section	Edit
Results Relative proportions of pluripotent subpopulations vary across 213 hiPSC lines Page 4	Previous studies have demonstrated that human induced pluripotent stem cell (hiPSC) lines are composed of subpopulations of interconvertible pluripotent cell states (Figure 1A, Figure S1)^{1,4}. We set out to determine if the relative proportion of these cell states varied across 213 hiPSC lines with bulk RNA-seq data (Table S1) using cellular deconvolution²⁰. Deconvolution typically uses gene signatures obtained from cell type clusters in single-cell data; however, due to the high similarity between hiPSC pluripotency states, they don't separate into distinct clusters. Therefore, we generated gene signatures using bulk RNA-seq data for FACS-sorted formative and paired unsorted (e.g. primed) cells¹. We applied the CIBERSORT deconvolution algorithm with a signature matrix containing the 100 most differentially expressed genes between the two populations (Table S3) and observed that the estimated fraction of cells in the formative state exhibited a wide range across the 213 hiPSC lines (Figure S2A). We examined the expression of formative-specific and primed-specific genes in hiPSCs with the highest and lowest estimated proportions of formative cells and observed notable expression differences of key regulators of pluripotency, such as DUSP5, LEFTY1, FST, and FZD5 (Figure 1B-E, Figure S2B-C). Taken together, these results show that remarkable variation in pluripotency cell state composition exists across the 213 hiPSC lines under conventional culture conditions.
Figures 1B-E	Shows gene expression profiles of DUSP5, LEFTY1, FST, and FZD5 in the samples with the highest estimated formative and primed proportions.
Figure S2A	Shows a stacked bar plot with the cell state estimates for all 213 RNA-seq samples.
Figure S2B-C	Shows the expression of formative-specific and primed-specific genes in hiPSCs with the highest and lowest estimated proportions of formative cells.
Methods	From GEO (GSE119324)¹ we downloaded 28 paired RNA-seq FASTQ files, including, fourteen from five unique human embryonic

Cellular deconvolution of pluripotency states using gene expression signatures Page 17	stem cells (hESCs) sorted (GCTM-2^{high}CD9^{high}EPCAM^{high}) for the formative (EPE) population and fourteen files for five unique samples generated from the primed population (unsorted). We aligned and filtered the 28 FASTQ files, with STAR 2.5.0a⁷⁰ and RSEM v1.2.20⁷⁴ as described above in RNA-seq data processing section. We performed differential expression analysis between the formative (EPE) and general population RNA-seq samples using DESeq2 (v.1.34) R package⁸⁶ and created a signature matrix containing 100 of the most differentially expressed genes (50 upregulated in the GCTM-2^{high}CD9^{high}EPCAM^{high} cells and 50 downregulated relative to the unfractionated cells; Table S3). We deconvoluted the 213 hiPSC RNA-seq samples by supplying the signature matrix and TPM expression matrix as input to the CIBERSORT.R script²⁰.
Results Identification of gene networks associated with pluripotency state Page 5	We next sought to determine if the GNMs were enriched for different pluripotency cell states. We first examined whether certain GNMs were associated with the estimated fraction of formative state cells (Figure S2A). For each of the 213 RNA-seq samples, we calculated a GNM score for each of the 13 GNMs by summing the inverse normal transformed TPM expression of the corresponding GNM-specific Pareto genes (each RNA-seq sample had 13 GNM scores). To identify associations between the estimated fraction of formative state cells and GNM scores across the 213 hiPSC lines, we ran a linear model and observed that GNM 5 was positively associated with the estimated proportion of formative state cells, while GNM 10 exhibited a strong negative association (Figure 1H-I). Next, we utilized marker genes for hiPSC pluripotency states defined in previous studies (Figure 1J). We discovered that three GNMs (1, 10, and 11) were associated with genes upregulated in the 8-cell like cells (8CLC) totipotent cell state³; GNM 5 was enriched for gene sets upregulated in the naïve²⁹, formative (EPE)¹, epiblast²⁹ states; GNM 9 and GNM 13 were both enriched for genes upregulated in the primitive endoderm (PrE)²⁹; while GNM 10 was strongly depleted for genes associated with the formative state¹ and enriched for genes associated with the primed¹, 8CLC³, and trophectoderm²⁹ states (Figure 1J).
Figure 1H-I	Shows the association between the estimated cell state and GNMs 5 and 10
Figure S5	Shows the association between the estimated formative proportion with remaining 11 GNMs
Methods GNM Enrichment in Stem Cell States Page 20	We determined if the GNMs were enriched for expressing marker genes from three published studies describing 7 hiPSC subpopulations representing different pluripotency cell states including: (1) 199 8-cell like cell (8CLC)-associated genes from Mazid et al³; (2) 123 primitive endoderm (PrE)-associated genes, 248 epiblast-associated genes, 175 trophectoderm genes ; 96 Naïve-associated PSCs genes from Stirparo et al.²⁹; and (3) 266 EPE-associated genes and 452 general population-associated genes that we identified by reprocessing data from Lau et al.¹ (See Cellular deconvolution of pluripotency states using gene expression signatures). We performed a Fisher's Exact test to calculate the

	enrichment of hiPSC subpopulation or cell state associated genes in each GNM, using the genes in the remaining 12 major GNMs as background.
Results Identification of genome-wide regulatory network underlying the co-expression of gene modules Page 6	We also calculated the intramodular co-accessibility enrichment between each pairwise combination of the 13 RNMs, which showed that peaks within an RNM were significantly more likely to be co-accessible compared with peaks in different RNMs (Figure 2D). These findings validate that the RNMs capture highly co-accessible peaks. We sought to determine whether the RNMs were associated with specific pluripotency states. We initially performed cellular deconvolution on the 150 ATAC-seq libraries using the most differentially accessible ATAC-seq peaks from FACS-sorted formative and primed cells¹. Only a small fraction of cells (range 0-36.6%) in each ATAC-seq sample were estimated to be in the formative state (Figure 2E, Table S6). We calculated RNM scores by summing the inverse normal transformed accessibility of 9,545 Pareto peaks (top 20% intramodular connectivity) for each ATAC-seq sample (each sample had 13 RNM scores) . We then ran a linear model to test for associations between the estimated proportion of formative state cells and RNM scores (Figure 2F-H). We observed that RNM 3 had a strong positive correlation with the estimated formative proportion, while RNMs 2 and 8 had weaker positive correlations (Figure 2F-H).
Figure 2D	Shows a heatmap showing the pairwise associations between 13 RNMs based on the co-accessibility enrichment
Figure 2E-H	Shows the estimated cell state proportions across 150 ATAC-seq samples and boxplots demonstrating RNM 2 (F), 3 (G), and 8 (H) associations with the estimated proportion of cells in the formative-state.
Methods Cellular deconvolution of pluripotency states using ATAC-seq peak accessibility signatures Page 20-21	We downloaded 6 ATAC-seq samples that were performed on human embryonic stem cells (hESCs) that had been sorted for the formative (GCTM-2^{high}CD9^{high}EPCAM^{high}) and the primed (GCTM-2^{mid}-CD9^{mid}) cell states from GEO (GSE147338)¹. We aligned and filtered the FASTQ files, with STAR 2.5.0a⁷⁰ as described in Sequencing of ATAC-seq section. MACS2 v.2.2.7 was used to establish a reference set of narrow peaks on the 6 ATAC-seq samples simultaneously, using the parameters; -f BAMPE -g hs -t --nomodel --shift -100 --extsize 200 --narrow. For each of the 6 individual ATAC-seq samples, read counts in the 193,147 reference peaks were obtained using featureCounts in Subread package v1.5.0⁸² . We performed differential accessibility analysis on the counts using DESeq2 (v.1.34) R package⁸⁶. We used bedtools intersect -r 0.25 to identify 938 formative-associated peaks that overlapped iPSCORE ATAC-seq peaks with a 25% reciprocal overlap (Table S8). We used the ATAC-seq peaks that were differentially accessible in the GCTM-2^{mid}-CD9^{mid} hiPSC subpopulation¹ to annotate 2,981 non-formative associated peaks using the same approach. We then deconvoluted the 150 ATAC-seq samples using the CIBERSORT.R script²⁰ with a signature

	matrix containing 200 of the most differentially accessible formative and primed peaks (Table S7).
Methods Calculation of ATAC-seq peak Co-accessibility Enrichment Page 22	To assess whether peaks within each RNM were more co-accessible with each other than peaks in different RNMs, we enumerated the number of co-accessible peaks shared between all pairwise RNMs. We then performed a Fisher's Exact test to calculate the enrichment of co-accessibility between RNM pairs.

5. On similar lines, the sentence: "These data indicate that culturing with a ROCK inhibitor shifts the composition of pluripotency cell states decreasing the number of cells in a primed-like state and increasing the number in the formative-like state." Again, this is inferred and not directly shown and the data do not address whether the proportion of cells is changing. And it is unclear why the formative state is specifically highlighted here, when the GNM is also associated with naïve PSCs, PrE and conventional PSCs.

We thank the reviewer for raising this concern. Our initial claims were based on GNM 32 enrichment with genes that were both formative-specific and upregulated by ROCKi (GNM 32 is now GNM 5 in the updated analyses conducted in response to Reviewer 1, comment 6). However, as the reviewer correctly points out, the association was an indirect inference. We did not have the appropriate data to directly test ROCKi's effect on cell state proportions in iPSCs and since these our claims about ROCKi are not a central conclusion of the study, we decided to remove sections detailing these analyses. The removal of these sections improved the narrative of the manuscript by allocating more focus on the conclusions about the complexity of pluripotency regulatory networks and the effect of genetic variation on NANOG-OCT4 binding.

6. Following integration of ATAC peaks with chromatin states, it was especially interesting to see that about half of the ATAC peaks overlapped with active chromatin states. One prediction might be that constructing RNMs from these active-associated ATAC peaks could have some benefit, i.e. it might lead to more specific networks and integration with GNMs. If it is possible to check this then that would be very interesting to do; but if it is a lot of work, then instead perhaps just add to the discussion that this is one future outlook of the work that could be done to further improve the network predictions.

We thank the reviewer for this important comment, as the implementation substantially strengthened the paper. While addressing Reviewer 2 (comment 1) about network validation, we observed that modules that consisted of higher proportions of peaks in quiescent chromatin, heterochromatin, and ZNF Repeat regions exhibited diminished intramodular co-accessibility. Based on this reviewer's comment, we hypothesized that removing the peaks in these three non-active chromatin states would improve the intramodular strength of the discovered network

modules. The remaining 12 chromatin states have well-known functions in hiPSC genome regulation. For example, repressed polycomb complexes and bivalent chromatin are known to repress the expression of developmental genes in pluripotent stem cells (Kashyap et al., 2009; Kumar et al., 2022). We removed ATAC-seq peaks in the three non-active chromatin states and calculated the pairwise co-accessibility for additional ATAC-seq peaks in the 12 remaining chromatin states. We included the newly calculated co-accessibility to regenerate the network modules and observed an improvement in: 1) the network validation by increasing intra-modular co-accessibility (Figure 2D) and, 2) integration with the GNMs (Figure 2I) because we were able to annotate more peaks with putative gene targets. We have updated the Results and the Methods to reflect these changes. Below is a table detailing the changes.

Section	Edit
Methods Identification of 56,978 autosomal reference peaks Page 17	To reduce the computational burden in downstream analyses, we selected a subset of 56,978 reference peaks based on their MACS2 peak score and enrichment in active chromatin. We filtered ATAC-seq peaks that overlapped blacklisted regions⁸⁴, peaks in ZNF genes & repeats, heterochromatin, quiescent chromatin states from the ChromHMM model for iPSC-18⁸⁵, and peaks on sex chromosomes. We binned the remaining 87,659 ATAC-seq peaks into 20 MACS2 score quantiles each containing 4,383 peaks, where quantile 20 consisted of the peaks with the highest scores. We then examined the enrichment of the 87,659 ATAC-seq peaks in iPSC-18 ChromHMM chromatin states by MACS2 score quantile (Figure S6C). We created 20 bed files containing the coordinates of the peaks in each quantile and calculated their enrichment (Odds Ratio) in each of the 12 chromatin states for iPSC-18, using bedtools fisher. The higher quantiles tended to be enriched for active (TssA, TssAFlnk, TxFlnk), bivalent (TssBiv, BivFlnk, EnhBiv), and repressed polycomb (ReprPC, ReprPCWk) chromatin. Based on these observations, we calculated pairwise co-accessibility for the 56,978 ATAC-seq peaks in MACS2 quantiles 8-20 (see Identifying co-accessible peaks section below), which alleviated the computational burden by eliminating 7×10^9 pairwise tests between peaks with low accessibility and peaks in inactive regions of the genome. For downstream analyses, we resolved a single chromatin state for each peak by intersecting the maximum summits with the iPSC-18 chromatin states (described below in Collapsing into 5 hiPSC chromatin states) (Table S8).
Figure S6C	Shows the enrichment of peaks the 12 active chromatin states by MACS2 score quantile.

7. Several of the figures are complex and dense, and I think tweaking the presentation of some of them could improve readability. In particular, I found the ordering and abbreviations of the chromatin states in Figures 4 and 5 not that logical, and perhaps by changing the ordering and/or using other visual cues could make this easier to read.

Thank you for providing feedback on the layout of the figures. To improve the readability of the manuscript, we have updated all figures. The table below details how the figures have been updated.

Original Figure Number	Updated Figure Number	Edits
1	NA	We removed the study overview figure because it was dense. We integrated much of the content detailed in Figure 1 into the bottom of the Introduction.
2	1	We extensively updated Figure 2 to address comments about the lack of validation (see Reviewer 1, Comment 4). Panels B-E report new results for the RNA-seq cellular deconvolution using external datasets. Panel G addresses a comment about the validation of gene network module discovery. Panels H and I show that modules are associated the estimated cell state proportion, which strengthens our claim that GNMs represent the cell state gene expression. For panel J, we restricted the analysis to include gene sets from Mazid et al, Lau et al, and Stirparo et al. We color-coded the gene set names to clarify which were obtained from ex vivo (in red) or in vitro (in blue) samples.
3	2	Several panels in Figure 3 were also updated. The cartoon in panel A was simplified for clarity. Like Figure 1 (formerly Figure 2), panel D is a validation of the RNM discovery method. Panels E through H correspond to the new results from the cellular deconvolution of the ATAC-seq data.
4	S7	Figure 4 was moved to the supplement as Figure S7. It includes the strategy for collapsing 12 active chromatin states in 5 groups. The corresponding Method Section Collapsing into 5 hiPSC chromatin states (page 23) is pasted below: “Using the single chromatin state annotation (described above in Identification of 56,978 autosomal reference peaks), we binned the 12 chromatin states into 5 collapsed states by molecular similarities (Figure S7B). “Active promoters” (TssA) were not collapsed, the collapsed “Enhancer” annotation consisted of peaks in enhancers (Enh), genic enhancers (EnhG), and flanking active promoters (TssAFlnk), “Bivalent Chromatin” consisted of bivalent promoters (TssBiv), bivalent enhancers (EnhBiv), and regions flanking bivalent chromatin (BivFlnk), “Transcription” consists of strong (Tx) and weak (TxWk) transcription, and flanking transcription (TxFlnk), and “Repressed Polycomb” consists of both polycomb states (ReprPC, ReprPCWk).”

5	3	We simplified Figure 5 by removing the qualitative Biocircos plots and adding panel B to Figure 3. This reduced the figure's size and improved the plot legibility.
6	4	We lightly edited Figure 6 to reflect the updated TFBS enrichment analysis.
7	5	We modified the approach to evaluate allele specific chromatin accessibility (see Reviewer 1, Comment 1). The results from the updated analysis are easier to interpret and the figures are simplified.

8. Page 12, the sentence: "Here, we show that rs9350250 is associated with decreased accessibility of three primed-associated RNMs (21, 24, and 37), and increased accessibility of formative-associated RNMs 3 and 8 (Figure 7G)". I don't see the association with RNM21.

We thank the reviewer for pointing out this discrepancy. As mentioned above (Reviewer 1, Comment 1), we took a different approach to evaluating the impact of genetic variants on allele-specific chromatin accessibility. Although we found the results from the original analysis very interesting, the claims were quite strong and because of a lack of suitable datasets there were no way to computationally validate our findings. The updated analysis showed that ASCA SNPs with higher allelic imbalance were differentially enriched in canonical pluripotency factors which is a novel observation that should be experimentally validated in future studies.

9. Also in Figure 7, is there a link between rs9350250 genotype and the expression levels of E2F3 (or other nearby genes)? It would be informative to comment on this.

The reviewer brings up an excellent question. As mentioned above (Reviewer 1, Comments 1 and 8), we no longer highlight rs9350250 in the manuscript. Prior to removing it, we did test whether rs9350250 influenced the expression of E2F3 by performing a targeted quantitative trait locus analysis, but did not observe an association. There are a few reasons why we were not able to detect an association. The RNA-seq and ATAC-seq were generated at different times and from iPSCs grown in slightly different media conditions (with and without ROCK inhibitor), which limited our ability to completely integrate the two datasets and address questions like the one posed. Ultimately, the inability to functionally validate our ASCA predictions contributed to our decision to modify the last section and tone back our claims.

10. [minor] In the sentence: "We analyzed the expression of 8,972 Pareto peaks (top 20% intra-RNM degree connectivity) and observed RNM specific clustering in the UMAP space (Figure 3C)", I don't think "expression" is the right word here, as you are talking about accessibility.

We thank the reviewer for identifying this discrepancy. We have changed "expression" to the appropriate term, "accessibility".

Reviewer #2 (Remarks to the Author):

The authors presented an unsupervised machine learning algorithm to discover genome-wide regulatory network modules (RNMs) and gene network module (GNMs) from ATAC-seq and RNA-seq datasets, and revealed that the GNMs and RNMs were highly correlated with each other. Their findings uncovered novel pluripotency regulatory mechanisms and provided a resource for future stem cell research. The manuscript can be further improved through addressing the following points:

We thank the reviewer for their encouraging comments about the value of our study. We hope that the modifications to the text we detail below satisfy their concerns.

1. The analysis considered in this paper was accomplished by applying some well-established methods. This is a weak point for this paper. The considered samples and validation cases are not challenging. The novelty and contribution of this paper are limited.

We have broken our response into three sections to respond to each of the reviewer's comments.

- *The analysis considered in this paper was accomplished by applying some well established methods.*

While network-based algorithms are commonly employed, they have not yet been used to analyze ATAC-seq data from hundreds of hiPSC lines derived from different individuals to identify co-accessible regulatory elements distributed across the genome. We have provided more context on previous studies and the current state of the field in the introduction and where the gaps are that our current study addresses. In response to Reviewer 2, Comment 3 we have also clarified that the network approach we used was novel (see below).

See the additions we made to the manuscript in the table below.

Section	Edit
Introduction Pages 3-4	But thus far, applications of these algorithms to analyze ATAC-seq data have been more limited. For example, studies have used paired bulk ATAC-seq and RNA-seq to examine how regulatory elements impact gene expression networks under dynamic conditions, including comparing cells at baseline and post-stimulation ⁹ or undergoing differentiation ¹⁰ . There has also been considerable effort

	studying patterns of co-accessibility in single-cell ATAC-seq under static conditions to explore local cis interactions (~500kb) between regulatory elements both in single tissues¹¹ and simultaneously across multiple tissues¹²; and a recent study analyzed paired single-cell ATAC-seq and RNA-seq data to study local cis-regulation of gene expression under environmental stimuli¹³. However, genes that are members of the same biological pathways are frequently encoded on different chromosomes, and there have been limited studies aimed at examining co-accessible ATAC-seq peaks distributed across the human genome to understand regulatory processes underlying the co-expression of gene modules. For example, in hiPSCs, the maintenance of pluripotency relies on the expression of pluripotency-related transcription factors, such as NANOG, OCT4, and SOX2, which create global epigenomic regulatory networks that enable self-renewal through the repression of developmental genes¹⁴, regulation of cell cycle transitions^{1,15}, and promotion of autoregulation^{16,17}.
--	---

- *The considered samples and validation cases are not challenging.*

We thank the reviewer for requesting validation of the network modules, as the implementation substantially strengthened the paper. Reviewer 3, Comment 7 also requested validation. When we investigated this concern, we noted that there was a considerable amount of co-expression between genes in different modules. For this reason, we optimized the network discovery algorithm and validated our findings as follows:

Optimization

- First, we ran the Leiden community detection algorithm 1,700 times with different combinations of the three parameters (resolution, beta, and n_iterations) to identify the model that yields networks with the highest intramodularity and highest resolution modules consisting of biologically relevant gene membership.
- We permuted the nodes to confirm the Leiden algorithm was clustering co-expressed genes better than the null background (data not shown).
- Since there are no currently established standard modularity thresholds for biological networks due to limitations discussed in this review (Lancichinetti and Fortunato, 2011), we selected the combination that had a modularity greater than 0.4 and yielded the highest number of modules (element membership > 100 genes) and the highest fraction of elements in modules.

Validation

- A. We validated our networks and claims about cell state specificity by performing cellular deconvolution using gene signatures from FACS-sorted populations of formative and primed hiPSC subpopulations (Lau et al., 2020). We estimated the relative proportion of the formative and primed states across the 213 RNA-seq samples and 150 ATAC-seq samples. We demonstrated that GNMs and RNMs are associated with the estimated formative proportion. This observation indicates that the GNM and RNM modules are respectfully composed of co-expressed genes and co-accessible peaks that capture specific biological signals.
- B. We further validated the networks by showing that elements in the same module are more likely to be co-expressed or co-accessible compared to elements in other modules (Figure 1G, Figure 2D).
- C. The accuracy of biological networks is typically assessed by functional enrichment. We obtained five gene sets (8CLC and Naïve, epiblast, trophoctoderm, primitive endoderm) from differential expression analyses (Mazid et al., 2022; Stirparo et al., 2018). We reanalyzed the differential expression data to curate gene sets for the formative and primed cell states (Lau et al., 2020). We performed a functional enrichment analysis on these seven gene set to validate our gene network modules.
- D. We also benchmarked our approach against established tools, such as weighted gene co-expression analysis (WGCNA). Since our model accounts for kinship, it outperformed WGCNA in module detection precision. Reviewer 2, Comment 3 directly requests elaboration on why our approach is more appropriate compared to other established methods. We provide a detailed description on how we benchmarked our model against WGCNA in response to that comment below.

See the manuscript edits in the table below.

Section	Edit
Methods Construction of the Genome-wide Gene Co-expression Network (GN) Page 19	We optimized module detection by analyzing 1,700 combinations of three parameters: resolution (range: 0-5), beta (range: 0-0.1), and n_iterations (range 5-50). For each combination of parameters, we permuted the nodes to confirm the Leiden algorithm was clustering co-expressed genes better than NULL background. We calculated modularity, fractions of genes in major GNMs (membership > 100) and the number of modules for each combination of parameters. We selected the modules obtained by resolution =2.25, beta = 0.05, n_iterations = 45, which exhibited a decent modularity (Q=0.41) and a high fraction of genes captured by major GNMs (91.8%). Under

	these parameters, genes within the same GNM were significantly more co-expressed than genes randomly connected through permutation (P-value = 0). As there is no consensus on modularity thresholds or network validation methods⁸⁸, we used downstream analyses, such as co-expression and gene set enrichment for validation. For each gene, we calculated its intramodular degree (the number of co-expressed genes within the module), using the same functions described above, for each GNM independently, excluding inter-module edges (Table S4).
Results Relative proportions of pluripotent subpopulations vary across 213 hiPSC lines Page 4	Previous studies have demonstrated that human induced pluripotent stem cell (hiPSC) lines are composed of subpopulations of interconvertible pluripotent cell states (Figure 1A, Figure S1)^{1,4}. We set out to determine if the relative proportion of these cell states varied across 213 hiPSC lines with bulk RNA-seq data (Table S1) using cellular deconvolution²⁰. Deconvolution typically uses gene signatures obtained from cell type clusters in single-cell data; however, due to the high similarity between hiPSC pluripotency states, they don't separate into distinct clusters. Therefore, we generated gene signatures using bulk RNA-seq data for FACS-sorted formative and paired unsorted (e.g. primed) cells¹. We applied the CIBERSORT deconvolution algorithm with a signature matrix containing the 100 most differentially expressed genes between the two populations (Table S3) and observed that the estimated fraction of cells in the formative state exhibited a wide range across the 213 hiPSC lines (Figure S2A). We examined the expression of formative-specific and primed-specific genes in hiPSCs with the highest and lowest estimated proportions of formative cells and observed notable expression differences of key regulators of pluripotency, such as DUSP5, LEFTY1, FST, and FZD5 (Figure 1B-E, Figure S2B-C). Taken together, these results show that remarkable variation in pluripotency cell state composition exists across the 213 hiPSC lines under conventional culture conditions.
Figures 1B-E	Shows gene expression profiles of DUSP5 , LEFTY1 , FST , and FZD5 in the samples with the highest estimated formative and primed proportions.
Figure S2A	Shows a stacked bar plot with the cell state estimates for all 213 RNA-seq samples.
Methods Calculation of GNM Co-expression Enrichment Page 20	To assess whether genes within each GNM were more co-expressed with each other across the 213 hiPSC lines than with genes in different GNMs, we enumerated the number of co-expressed genes shared between all pairwise combinations of GNMs. We then performed a Fisher's Exact test to calculate the enrichment of genes showing co-expression within each GNM, using the genes co-expressed between each set of paired GNMs as background.
Results Identification of genome-wide regulatory network	We also calculated the intramodular co-accessibility enrichment between each pairwise combination of the 13 RNMs, which showed that peaks within an RNM were significantly more likely to be co-

underlying the co-expression of gene modules Page 6	accessible compared with peaks in different RNMs (Figure 2D, Table S13). These findings validate that the RNMs capture highly co-accessible peaks.
Figure 1G	Shows that genes within the same GNM are more co-expressed with each other than with genes in different GNMs.
Figure 2D	Shows that peaks within the same RNM are more co-accessible with each other than with peaks in different RNMs.
Methods GNM Enrichment in Stem Cell States Page 20	We determined if the GNMs were enriched for expressing marker genes from three published studies describing 7 hiPSC subpopulations representing different pluripotency cell states including: (1) 199 8-cell like cell (8CLC)-associated genes from Mazid et al.³; (2) 123 primitive endoderm (PrE)-associated genes, 248 epiblast-associated genes, 175 trophectoderm genes ; 96 Naïve-associated PSCs genes from Stirparo et al.²⁹; and (3) 266 EPE-associated genes and 452 general population-associated genes that we identified by reprocessing data from Lau et al.¹ (See Cellular deconvolution of pluripotency states using gene expression signatures). We performed a Fisher's Exact test to calculate the enrichment of hiPSC subpopulation or cell state associated genes in each GNM, using the genes in the remaining 12 major GNMs as background.
Results Identification of gene networks associated with pluripotency state Page 5	To identify associations between the estimated fraction of formative state cells and GNM scores across the 213 hiPSC lines, we ran a linear model and observed that GNM 5 was positively associated with the estimated proportion of formative state cells, while GNM 10 exhibited a strong negative association (Figure 1H-I, Figure S5). Next, we utilized marker genes for hiPSC pluripotency states defined in previous studies (Figure 1J). We discovered that three GNMs (1, 10, and 11) were associated with genes upregulated in the 8-cell like cells (8CLC) totipotent cell state³; GNM 5 was enriched for gene sets upregulated in the naïve²⁹, formative (EPE)¹, epiblast²⁹ states; GNM 9 and GNM 13 were both enriched for genes upregulated in the primitive endoderm (PrE)²⁹; while GNM 10 was strongly depleted for genes associated with the formative state¹ and enriched for genes associated with the primed¹, 8CLC³, and trophectoderm²⁹ states (Figure 1J).
Figure 1J	Shows that the GNMs are differentially enriched with gene sets from early embryonic cell states.
Results Identification of gene networks associated with pluripotency state	Supplemental Note 1 shows our approach for detecting gene modules works better for the iPSCORE cohort, which contains related individuals, than the commonly used WGCNA⁶ approach; Figure S3, Figure S4)

- *The novelty and contribution of this paper are limited.*

Reviewer 3, Comment 4 also requested additional clarification of the novelty our findings. For this reason, we have extensively re-written the manuscript to emphasize the following points.

Although the existence of a spectrum of cell states of pluripotency (formative or early post-implantation and primed or late post-implantation states) within hiPSC culture grown under conventional culture conditions has been described previously, this study provides important new insight into these cellular states.

- This is the first study to perform ATAC-seq on hundreds of hiPSCs lines with deep whole genome sequencing. We believe that the release of this data will be a valuable resource and contribution to the field.
- The reviewer suggests that this is a common method, however we were unable to find any other reports of genome-wide networks of regulatory elements (ATAC-seq peaks). Most studies that analyze RNA and ATAC for a given cell type, characterize cis-regulatory elements (CREs) within ~500kb of a putative gene target (Pliner et al., 2018). Genome-wide networks enable the characterization of cell state regulatory mechanisms, as they connect all CREs that are simultaneously active. This was particularly useful in characterizing this dataset because differences between formative and primed pluripotency are poorly characterized which limits the ability to perform functional annotations with external datasets.
- In the revisions, we applied cellular deconvolution to hiPSCs to estimate the proportion of formative and primed cell states in RNA-seq and ATAC-seq samples for hundreds of genetically diverse lines. Our findings imply that there are different compositions of cell states between hiPSC lines, and this has important implications for the use of diverse cell lines in embryo and disease modeling. There are four reasons why our deconvolution analysis is novel:
 - Unlike mature tissues and cell types, signature genes and regulatory elements of hiPSC cell states are poorly characterized. We performed differential expression and differential accessibility between RNA-seq and ATAC-seq samples from

FACS-sorted formative and primed hiPSC subpopulations to create a signature matrix for cellular deconvolution. We show that bulk RNA samples have an incredibly variable expression of these signature genes (**Figure 1B-E**). The signature matrices are a valuable resource for evaluating the composition of the hiPSCs.

- To our knowledge, this is the first application of cellular deconvolution of cell states rather than cell types.
- Most studies that use deconvolution apply it on bulk RNA-seq samples. This study is among the first to apply cellular deconvolution to bulk ATAC-seq samples.
- Conventional views of genome regulation in pluripotent stem cells consider NANOG-OCT4 mediated regulatory networks, bivalent chromatin, repressed polycomb complexes, and cell cycle regulation to be uniformly active (Lau et al., 2020; Li and Belmonte, 2017). Our study demonstrates that these processes are distinguishable from each other and are differentially active in the formative and primed cell states indicating that they may be important for cell state transitions. Below are a few examples:
 - Polycomb repressed complexes are only enriched in a primed-associated regulatory network (RNM 6, **Figure 3C**), suggesting that they are a cell state-specific mechanism for repressing differentiation. **Page 8**
 - Formative associated RNM 8 is enriched for bivalent chromatin and G1 phase specific TFs. Previous reports show that formative state have abbreviated G1 phases, suggesting that RNM 8 may underlie a transition between G1 and S phase in formative pluripotency. **Page 8-9**
 - Emerging evidence has implicated TEAD4 plays an important role in naïve and formative pluripotency and that it might be a pioneering factor for placental lineage commitment. We identify RNM 3, a formative-associated network, that is primarily enriched for TEAD4 binding sites (there is a weak enrichment with 1 other TF group, ATF4-CEBPG Complex) and enhancers (**Figure 3C**). This discrete classification could provide a valuable resource for the development of differentiation protocols for trophoblast and placental lineages. **Page 8-9**
- Our study is the first to annotate the stem cell epigenome with regulatory elements that become active during fetal development (**Figure 4**). Our analysis shows that RNMs are differentially enriched with 49 fetal cell type-specific peaks. These results indicate that distinct types of early embryonic regulatory elements are reused in later stages of fetal

development. It also indicates that lineage-specific regulatory elements are regulated by distinct mechanisms during pluripotency. It is well known that NANOG regulation is associated with neuronal development (Deb-Rinker et al., 2005) however our study is the first to provide predictions about TFs regulate genes associated with fate commitment to different lineages during pluripotency.

- We show that TF binding sites are differentially enriched for regulatory variation. To our knowledge, allele-specific chromatin accessibility has never been conducted on hundreds of hiPSC lines from genetically diverse individuals. We demonstrate that despite its importance for regulating pluripotency NANOG-OCT4 binding sites are enriched for regulatory variants that impact chromatin accessibility. This novel observation has remarkable implications for evolution and speciation. Future experimentation can further characterize the effects of these regulatory variants.

We appreciate the Reviewer requesting clarification on the novelty of our findings. We have extensively edited the manuscript to highlight these novel aspects of our work.

2. The authors constructed the gene co-expression network and chromatin co-accessibility network from RNA-seq data and ATAC-seq data respectively. They used the same model (i.e., Linear Mixed effects Model, LMM) to construct these two networks and used the same model (i.e., Leiden community detection algorithm) to detect network modules. Why the co-accessibility network can reflect the regulatory relationships?

We thank the reviewer for the question. We have broken the response into two sections; 1) an explanation of the underlying biological assumptions, and 2) an explanation of our approach.

1. It has been hypothesized that the mechanism underlying the coordination of gene co-expression is co-accessible regulatory elements with similar epigenetic profiles (i.e. transcription factor binding and chromatin state). From a biological perspective, genes that are co-expressed are involved in shared biological processes. These genes are distributed across the genome, thus the regulatory elements that coordinate their expression must be co-accessible and distributed across the genome. To test this hypothesis, we show that the gene and regulatory networks are highly correlated in **Figure 2I**.
2. As we indicate in the manuscript, although the RNA-seq and ATAC-seq data were generated from the same hiPSC lines, they were generated from samples collected at different times and in slightly different culture conditions. This limited a complete

integration of the two datasets. In response to the reviewer’s comment, we independently generated the GNMs and RNMs, and then independently calculated the association between the estimated formative proportion and the GNMs and RNMs. Observing that formative GNM 5 and RNMs 2, 3, and 8 are associated with each other indicates that the regulatory networks reflect the gene networks. The functional characterization of the GNMs and RNMs are concordant with previous characterizations of the stem cell epigenome. For example, the formative state is defined by an upregulation of Nodal signaling, which is concordant with GNM 5 (Figure 1J-K)(Lau et al., 2020). It is also defined by differential TEAD binding sites, which is concordant with RNM 3 (Figure 3C)(Lau et al., 2020).

Please look at the table below to see the locations of these analyses in our manuscript.

Section	Edit
Results Identification of gene networks associated with pluripotency state Page 5	We first examined whether certain GNMs were associated with the estimated fraction of formative state cells (Figure S2A). For each of the 213 RNA-seq samples, we calculated a GNM score for each of the 13 GNMs by summing the inverse normal transformed TPM expression of the corresponding GNM-specific Pareto genes (each RNA-seq sample had 13 GNM scores). To identify associations between the estimated fraction of formative state cells and GNM scores across the 213 hiPSC lines, we ran a linear model and observed that GNM 5 was positively associated with the estimated proportion of formative state cells, while GNM 10 exhibited a strong negative association (Figure 1H-I).
Results Identification of genome-wide regulatory network underlying the co-expression of gene modules Page 6	We calculated RNM scores by summing the inverse normal transformed accessibility of 9,545 Pareto peaks (top 20% intramodular connectivity) for each ATAC-seq sample (each sample had 13 RNM scores) . We then ran a linear model to test for associations between the estimated proportion of formative state cells and RNM scores (Figure 2F-H). We observed that RNM 3 had a strong positive correlation with the estimated formative proportion, while RNMs 2 and 8 had weaker positive correlations (Figure 2F-H).
Results Identification of genome-wide regulatory network underlying the co-	We annotated 32,327 ATAC-seq peaks in the 13 RNMs with 12,078 neighboring candidate target genes and then performed a Fisher’s Exact test (see Methods) to calculate enrichments of RNMs in GNMs. We found that all RNMs had an association with at least one GNM (Figure 2I). For example, five RNMs (2, 3, 5, 8, and 13) were positively enriched for the formative GNM 5 (Figure 1G-H, J), suggesting that co-accessibility of ATAC-seq peaks across the genome

expression of gene modules Page 6-7	mechanistically underlie the differential expression of genes between pluripotency states.
Figure 2f	Shows the associations between gene and regulatory networks.
Methods Identifying Associations Between Gene and Regulatory Networks Page 22-23	The GNM were identified using ROCK inhibitor-naïve hiPSCs but the ATAC-seq peaks were identified in hiPSCs after culturing with ROCK inhibitor. To annotate peaks with putative gene targets we: 1) identified genes expressed after culturing with 3 hiPSC lines with ROCK inhibitor (see above: Generation of RNA-seq for 3 hiPSC lines expanded with ROCK inhibitor), and then 2) annotated each peak with a single gene (distance < 100 kb and highest expressed gene). Specifically, to identify candidate target genes for the 47,761 ATAC-seq peaks in the 13 major RNMs, we generated a bed file of the TSSs for the 16,110 autosomal genes expressed (TPM > 1 in at least one of the nine samples (triplicates of each hiPSC line) cultured with 5µM ROCK inhibitor and performed bedtools closest to identify the closest TSS within 100 kb of each ATAC-seq peak. For ATAC-seq peaks that overlapped the TSSs of multiple genes, we calculated the maximum TPM expression across all 3 hiPSC lines cultured with ROCK inhibitor and annotated the ATAC-peak with the gene with the maximum expression. Finally, to identify associations between the GNMs and RNMs we only used genes: 1) expressed in both ROCK inhibitor-naïve hiPSCs and ROCK inhibitor-exposed hiPSCs, 2) in one of the 13 major GNMs, and 3) annotated as a putative target in one of the 13 major RNMs. In total, 12,078 unique genes corresponding to 32,327 peaks were used for the association test (Table S8). We calculated the number of genes in common between all GNM-RNM pairwise combinations and performed Fisher's Exact tests to calculate enrichments.

3. It will be better if the authors could provide more discussion about why is this particular method suitable for this particular computational problem (and type of data) from statistical and biological perspectives.

We thank the reviewer for this comment. We agree with the reviewer that further explanation is needed to convey the utility of the approach. We have added text highlighting why this method was appropriate for the type of data from statistical and biological perspectives.

From a statistical perspective, the rationale for using a linear mixed model is to accommodate RNA-seq and ATAC-seq samples from hundreds of individuals of different ancestries and some of whom are related. Existing tools, such as WGCNA (Langfelder and Horvath, 2008), assume that the samples are independent, therefore it should not be used on samples from individuals of different ancestries or related individuals. Linear mixed models can account for kinship as

random effects. To demonstrate that our approach was more suitable for our dataset, we performed WGCNA (Langfelder and Horvath, 2008) on the 213 RNA-seq samples using the standard tutorial pipeline. We observed the following:

- We demonstrated that covariates and kinship that were corrected for in our LMM model are correlated with modules identified by WGCNA (Figure S3). These correlations may influence the module identifications.
- We demonstrated that our LMM network approach more precisely identifies gene modules (Figure 1J, Figure S4). Our model identifies a single GNM enriched with the formative-associated genes, while WGCNA identified two such modules (Figure S4).

This analysis supported that our linear mixed model network approach was more appropriate for our dataset compared to WGCNA.

Section	Edit
Supplemental Note 1	Weighted gene co-expression network analysis (WGCNA) ⁶ is the most commonly used gene module detection method, however, it cannot account for kinship (genetically related individuals). In this study, we used hiPSC lines from 219 individuals (Table S1) recruited as part of the iPSCORE resource, of which 140 belonged to families composed of two or more subjects (range: 2–14 subjects). To address this confounding factor, we first applied an LMM to calculate gene co-expression and ATAC-seq peak co-accessibility, using kinship as the random effects term (See Methods). We loaded the edges of the significantly co-expressed genes and co-accessibility ATAC-seq peaks into a network and applied the Leiden community detection algorithm to detect modules. To determine if our approach or WGCNA is more suitable for module detection using iPSCORE resource samples, we applied WGCNA to calculate gene co-expression in the 213 hiPSC RNA-seq samples. Downstream analyses showed that the WGCNA modules were correlated with biological and technical covariates, as well as family structure (Figure S3). We also determined that the LMM-Leiden module detection approach was more precise at identifying modules associated with formative-state-specific gene expression than WGCNA (Figure S4). These results show that the conventional WGCNA module detection approach can be affected by donor relatedness and that accounting for kinship leads to more accurate module membership.
Figure S3	Shows that WGCNA modules are correlated with biological and technical covariates that are corrected for in the linear mixed model approach.
Figure S4	Shows that multiple WGCNA modules are enriched with formative-specific genes, compared with Figure 1J which shows that only one GNM is enriched with formative-specific genes.

Methods Gene module identification using weighted correlation network analysis (WGCNA) Page 19-20	To benchmark our gene module detection approach (Supplemental Note 1), we processed the 213 RNA-seq samples using the standard workflow (https://horvath.genetics.ucla.edu/html/CoexpressionNetwork/Rpackages/WGCNA/Tutorials/) for the WGCNA R package⁶, which cannot account for covariates and kinship. We used WGCNA to calculate the associations with the 24 covariates used in the co-accessibility LMM, the estimated formative fraction (Figure S2A), and binary annotations for samples from 6 families with 5 or more individuals (Figure S3). We then compared our 13 GNMs with the 17 WGCNA modules by calculating the enrichment of the 7 cell state gene sets (Figure 1J, Figure S3).
---	--

4. Is it possible to use both chromatin accessibility and gene expression data to construct regulatory networks? As there are many methods being proposed for gene regulatory network inference, the more accurate the constructed network, the better for downstream analysis.

The reviewer provides an excellent suggestion. The RNA-seq and ATAC-seq were generated using the same hiPSCs; however, they were grown under different culture conditions several years apart. ATAC-seq was generated from samples that were cultured with ROCK inhibitor, while the RNA-seq was generated from earlier passage samples cultured in media not containing ROCK inhibitor. Therefore, we did not directly integrate both molecular datasets. Despite these differences in culture conditions, the gene and regulatory networks are highly correlated. See response above to Reviewer 2, Comment 2 for a detailed explanation.

The most common method to determine the accuracy of biological networks is by performing functional enrichment (Langfelder and Horvath, 2008). In our study, we demonstrate that the gene modules precisely capture the formative and primed cell states (**Figure 1J**). Additionally, we demonstrate that our regulatory networks (RNMs 2, 3, and 4, **Figure 3A-C**) exhibit well-known characteristics of formative pluripotency, such as NANOG-OCT4, TEAD, SOX, enhancer, and bivalent chromatin enrichments which supports the accuracy of modules. Moreover, we demonstrate that our method outperforms WGCNA in module precision (**Figure S3**) when the data contains samples from related individuals, as discussed in detail in our response to Reviewer 2, Comment 3.

Section	Edit
Results	We initially annotated each of the 56,978 reference ATAC-seq peaks with chromatin states collapsed into five main categories (See Methods, Table S8) and observed that 46.8% of the ATAC-seq peaks

Functional annotation of hiPSC ATAC-seq peaks Page 7	were in enhancers, 22.0% were in active promoters, 13.9% were in bivalent or poised chromatin, 5.8% were in repressed polycomb regions, and 11.45% were in transcribed regions (Figure S7B). Our annotations were consistent with previous characterizations of hiPSC regulatory elements^{1,36-38}, specifically; 1) the relatively large fraction of the peaks in bivalent chromatin indicating open but inactive regulatory elements, and 2) the presence of peaks in polycomb regions.
Results Functional characterization of regulatory network modules Page 8	RNM 2 was enriched for pluripotency TFs (NANOG-OCT4 complex, POU2F2/OCT2, SOX-LEF1 complex, TEAD Family) in enhancers, suggesting that it represents the hallmark hiPSC pluripotency regulatory network active in the formative state (Figure 3A-C). RNM 3 was highly enriched for enhancers and the TEAD Family (TEAD1 and TEAD4) (Figure 3A-C). TEAD signaling is strongly implicated in the differentiation of hiPSC to the trophoblast lineage⁴¹; and recently Dattani et al.⁴² showed that suppression of YAP/TEAD signaling was critical to the insulation of naive hiPSC from trophoblast differentiation. Though the capacity for conventional hiPSC to undergo trophoblast differentiation has been the subject of considerable controversy^{41,43}, it is certainly possible that formative state cells, closer to the naïve-state, might be capable of entering this differentiation pathway, and that regulatory elements in RNM 3 are in some way primed for activation in naïve and formative state cells⁴⁴. RNM 8 was strongly enriched for G₁ cell cycle-associated TFs (E2F Family, E2F2, E2F5, SP-E2F Complex) which is consistent with the fact that the formative state has a highly proliferative phenotype and an abbreviated G₁ phase^{1,45} (Figure 3A-C).

Reviewer #3 (Remarks to the Author):

This paper presents accessibility and gene expression data and analysis of nearly 200 human iPSC lines (150 ATAC-seq and 213 RNA-seq) from the iPSCORE consortium. The ATAC-seq is new to this study, the RNA-seq data was previously published. Using these data, the authors infer gene co-expression networks clustered into Gene Network Modules (GNMs) and also co-accessibility networks clustered into Regulatory Network Modules (RNMs). The paper characterizes the GNMs based on their enrichment of Gene Ontology and Msigdb genesets and overlap with existing genes associated with different states of pluripotency. The major finding from this analysis is that the hiPSCs capture a range of pluripotency cell states. The paper focuses more on the analysis of the ATAC-seq data and the inferred RNMs. The RNMs are analyzed based on overlap with ChromHMM states, enrichment in GNMs, overlap with existing sets of peaks representing different cell types or pluripotency cell states, and finally examination of allele-specific SNPs and their impact on accessibility. The overall resource generated by the paper, namely the 150 ATAC-seq profiles is very useful for the community, however, the overall paper is quite descriptive and it is unclear what the major novel findings of this work are. There are a number of observations made but in many cases they are consistent with what is known or it is not apparent what the major significance of the observation is. Another major limitation is

that the RNA-seq and ATAC-seq came from different lines which makes it difficult to more directly observe the impact of change in accessibility to expression. There is also no validation of the GNM or RNMs so it is hard to know how accurate these networks and modules are. More details follow:

We thank the reviewer for recognizing the novel aspects of our study, the usefulness of our ATAC-seq data as a community resource, and their thorough critique.

Strengths

1. The paper generates a useful resource, ATAC-seq data for 150 samples from different 133 human iPSC lines.
2. Presents one approach to analyze and interpret the data to gain insight into the transcriptome and accessibility state of the induced pluripotency state.
3. The TF grouping analysis and the preference of different groups for different genomic region types was interesting (second last para of "Functional annotation of hiPSC ATAC-seq peaks").

Weaknesses

1. The paper is quite descriptive and it is unclear what the major novel findings of this work are. For example based on the GNM analysis, the authors conclude that the hiPSC lines represent a mix of different known pluripotency cell states. However, this seems to have been already known. A similar finding is stated with accessibility as well as that there is some overlap with fetal cell line networks. These all seem like general expected observations.

We thank the reviewer for this comment. Reviewer 2, Comment 1 also requested additional clarification of the novelty of our findings. For this reason, we have extensively re-written the manuscript to emphasize our novel findings.

Below we list the novel findings of our study:

Although the existence of a spectrum of cell states of pluripotency (formative or early post-implantation and primed or late post-implantation states) within hiPSC culture grown under conventional culture conditions has been described previously, this study provides important new insight into these cellular states.

- This is the first study to perform ATAC-seq on hundreds of hiPSCs lines with deep whole genome sequencing. We believe that the release of this data will be a valuable resource and contribution to the field.
- The reviewer suggests that this is a common method, however we were unable to find any other reports of genome-wide networks of regulatory elements (ATAC-seq peaks). Most studies that analyze RNA and ATAC for a given cell type, characterize cis-regulatory elements (CREs) within ~500kb of a putative gene target (Pliner et al., 2018).

Genome-wide networks enable the characterization of cell state regulatory mechanisms, as they connect all CREs that are simultaneously active. This was particularly useful in characterizing this dataset because differences between formative and primed pluripotency are poorly characterized which limit the ability to perform functional annotations with external datasets.

- In the revisions, we applied cellular deconvolution to hiPSCs to estimate the proportion of formative and primed cell states in RNA-seq and ATAC-seq samples for hundreds of genetically diverse lines. Our findings imply that there are different compositions of cell states between hiPSC lines, and this has important implications for the use of diverse cell lines in embryo and disease modeling. There are four reasons why our deconvolution analysis is novel:
 - Unlike mature tissues and cell types, signature genes and regulatory elements of hiPSC cell states are poorly characterized. We performed differential expression and differential accessibility between RNA-seq and ATAC-seq samples from FACS-sorted formative and primed hiPSC subpopulations to create a signature matrix for cellular deconvolution. We show that bulk RNA samples have an incredibly variable expression of these signature genes (**Figure 1B-E**). The signature matrices are a valuable resource for evaluating the composition of the hiPSCs.
 - To our knowledge, this is the first application of cellular deconvolution of cell states rather than cell types.
 - Most studies that use deconvolution apply it on bulk RNA-seq samples. This study is among the first to apply cellular deconvolution to bulk ATAC-seq samples.
- Conventional views of genome regulation in pluripotent stem cells consider NANOG-OCT4 mediated regulatory networks, bivalent chromatin, repressed polycomb complexes, and cell cycle regulation to be uniformly active (Lau et al., 2020; Li and Belmonte, 2017). Our study demonstrates that these processes are distinguishable from each other and are differentially active in the formative and primed cell states indicating that they may be important for cell state transitions. Below are a few examples:
 - Polycomb repressed complexes are only enriched in a primed-associated regulatory network (RNM 6, **Figure 3C**), suggesting that they are a cell state-specific mechanism for repressing differentiation. **Page 8**

- Formative associated RNM 8 is enriched for bivalent chromatin and G1 phase specific TFs. Previous reports show that formative state have abbreviated G1 phases, suggesting that RNM 8 may underlie a transition between G1 and S phase in formative pluripotency. **Page 8-9**
- Emerging evidence has implicated TEAD4 plays an important role in naïve and formative pluripotency and that it might be a pioneering factor for placental lineage commitment. We identify RNM 3, a formative-associated network, that is primarily enriched for TEAD4 binding sites (there is a weak enrichment with 1 other TF group, ATF4-CEBPG Complex) and enhancers (**Figure 3C**). This discrete classification could provide a valuable resource for the development of differentiation protocols for trophoblast and placental lineages. **Page 8-9**
- We show that TF binding sites are differentially enriched for regulatory variation. To our knowledge, allele-specific chromatin accessibility has never been conducted on hundreds of hiPSC lines from genetically diverse individuals. We demonstrate that despite its importance for regulating pluripotency NANOG-OCT4 binding sites are enriched for regulatory variants that impact chromatin accessibility. This novel observation has remarkable implications for evolution and speciation. Future experimentation can further characterize the effects of these regulatory variants.

Regarding: *“A similar finding is stated with accessibility as well as that there is some overlap with fetal cell line networks. These all seem like general expected observations.”*

- Our study is the first to annotate the stem cell epigenome with regulatory elements that become active during fetal development (**Figure 4**). Our analysis shows that RNMs are differentially enriched with 49 fetal cell type-specific peaks. These results indicate that distinct types of early embryonic regulatory elements are reused in later stages of fetal development. It also indicates that lineage-specific regulatory elements are regulated by distinct mechanisms during pluripotency. It is well known that NANOG regulation is associated with neuronal development (Deb-Rinker et al., 2005) however our study is the first to provide predictions about TFs regulate genes associated with fate commitment to different lineages during pluripotency.

2. As such the integration of the GNMs and RNMs or the gene expression and accessibility was quite limited. Everything is based on enrichment analysis, however this is likely because the data were collected on different cohorts so we are unable to see the impact of changes in accessibility, including the allele-specific binding disruption on expression.

We thank the reviewer for this comment. We note that the RNA-seq and ATAC-seq were both generated on the same hiPSC lines from the iPSCORE cohort, but were collected at different times and under slightly different culture conditions. We agree that the integration of the RNA-seq and ATAC-seq is limited by this fact. However, in Figure 2I, we highlight the associations between the GNMs and RNMs. Despite this limitation, we believe that this study is a valuable resource for the stem cell biology field.

Below is a table highlighting how we determined the associations between gene and regulatory networks.

Section	Edit
Results Identification of genome-wide regulatory network underlying the co-expression of gene modules Page 6-7	We annotated 32,327 ATAC-seq peaks in the 13 RNMs with 12,078 neighboring candidate target genes and then performed a Fisher's Exact test (see Methods) to calculate enrichments of RNMs in GNMs. We found that all RNMs had an association with at least one GNM (Figure 2I). For example, five RNMs (2, 3, 5, 8, and 13) were positively enriched for the formative GNM 5 (Figure 1G-H, J), suggesting that co-accessibility of ATAC-seq peaks across the genome mechanistically underlie the differential expression of genes between pluripotency states.
Figure 2I	Shows the associations between gene and regulatory networks.
Methods Molecular Data Sources Page 13	Of note, both the RNA-seq and ATAC-seq data were generated from the same hiPSCs in the iPSCORE collection. However, the RNA-seq data and ATAC-seq data were generated from different passages of the hiPSC lines cultured under different experimental conditions. The RNA-seq data was generated from earlier passage ROCK inhibitor-naïve hiPSCs and the ATAC-seq data was generated from later passage hiPSCs that had been cultured with ROCK inhibitor.
Methods Identifying Associations Between Gene and Regulatory Networks Page 22-23	The GNMs were identified using ROCK inhibitor-naïve hiPSCs but the ATAC-seq peaks were identified in hiPSCs after culturing with ROCK inhibitor. To annotate peaks with putative gene targets we: 1) identified genes expressed after culturing with 3 hiPSC lines with ROCK inhibitor (see above: Generation of RNA-seq for 3 hiPSC lines expanded with ROCK inhibitor), and then 2) annotated each peak with a single gene (distance < 100 kb and highest expressed gene). Specifically, to identify candidate target genes for the 47,761 ATAC-seq peaks in the 13 major RNMs, we generated a bed file of the TSSs for the 16,110 autosomal genes expressed (TPM > 1 in at least one of the nine samples (triplicates of each hiPSC line) cultured with 5µM ROCK inhibitor and performed bedtools closest to identify the closest TSS within 100 kb of each ATAC-seq peak. For ATAC-seq peaks that overlapped the TSSs of multiple genes, we calculated the maximum TPM expression across all 3 hiPSC lines

	cultured with ROCK inhibitor and annotated the ATAC-peak with the gene with the maximum expression. Finally, to identify associations between the GNMs and RNMs we only used genes: 1) expressed in both ROCK inhibitor-naïve hiPSCs and ROCK inhibitor-exposed hiPSCs, 2) in one of the 13 major GNMs, and 3) annotated as a putative target in one of the 13 major RNMs. In total 12,078 unique genes corresponding to 32,327 peaks were used for the association test (Table S8). We calculated the number of genes in common between all GNM-RNM pairwise combinations and performed Fisher’s Exact tests to calculate enrichments.
--	--

3. For the GNM analysis, the authors decided to focus on Modules 14 and 32 however there were other modules that also exhibit the pluripotency-state specific signatures.

We thank the reviewer for their comment. The suggestion to focus on all the pluripotency-state modules led to interesting insights. Particularly, the epigenetic analyses of the RNMs revealed a surprising level of complexity underlying self-renewal and pluripotency.

The table below shows the edited text on pluripotent GNMs and RNMs.

Section	Edit
Results Identification of gene networks associated with pluripotency state Page 5	We first examined whether certain GNMs were associated with the estimated fraction of formative state cells (Figure S2A). For each of the 213 RNA-seq samples, we calculated a GNM score for each of the 13 GNMs by summing the inverse normal transformed TPM expression of the corresponding GNM-specific Pareto genes (each RNA-seq sample had 13 GNM scores). To identify associations between the estimated fraction of formative state cells and GNM scores across the 213 hiPSC lines, we ran a linear model and observed that GNM 5 was positively associated with the estimated proportion of formative state cells, while GNM 10 exhibited a strong negative association (Figure 1H-I). Next, we utilized marker genes for hiPSC pluripotency states defined in previous studies (Figure 1J). We discovered that three GNMs (1, 10, and 11) were associated with genes upregulated in the 8-cell like cells (8CLC) totipotent cell state³; GNM 5 was enriched for gene sets upregulated in the naïve²⁹, formative (EPE)¹, epiblast²⁹ states; GNM 9 and GNM 13 were both enriched for genes upregulated in the primitive endoderm (PrE)²⁹; while GNM 10 was strongly depleted for genes associated with the formative state¹ and enriched for genes associated with the primed¹, 8CLC³, and trophoctoderm²⁹ states (Figure 1J).
Figure 1J	Shows the enrichment of all 13 GNMs with pluripotency associated gene sets.
Figure 1H-I, Figure S5	Shows the associations with all 13 GNMs with the estimated formative proportions.

Results Functional characterization of regulatory network modules Page 8	We initially examined the three RNMs (2, 3, and 8) enriched for the formative-specific peaks (Figure 3A). RNM 2 was enriched for pluripotency TFs (NANOG-OCT4 complex, POU2F2/OCT2, SOX-LEF1 complex, TEAD Family) in enhancers, suggesting that it represents the hallmark hiPSC pluripotency regulatory network active in the formative state (Figure 3A-C). RNM 3 was highly enriched for enhancers and the TEAD Family (TEAD1 and TEAD4) (Figure 3A-C). TEAD signaling is strongly implicated in the differentiation of hiPSC to the trophoblast lineage⁴¹; and recently Dattani et al.⁴² showed that suppression of YAP/TEAD signaling was critical to the insulation of naive hiPSC from trophoblast differentiation. Though the capacity for conventional hiPSC to undergo trophoblast differentiation has been the subject of considerable controversy^{41,43}, it is certainly possible that formative state cells, closer to the naïve-state, might be capable of entering this differentiation pathway, and that regulatory elements in RNM 3 are in some way primed for activation in naïve and formative state cells⁴⁴. RNM 8 was strongly enriched for G₁ cell cycle-associated TFs (E2F Family, E2F2, E2F5, SP-E2F Complex) which is consistent with the fact that the formative state has a highly proliferative phenotype and an abbreviated G₁ phase^{1,45} (Figure 3A-C). This suggests that RNM 8 represents a network that may underlie cell cycle regulation in formative pluripotency. The three primed RNMs (6, 7, and 10) exhibited similar enrichments for E2F cell cycle associated TFBSs (Figure 3A-C), which is consistent with the primed state being in the G₁ phase¹. Interestingly, the primed associated RNMs displayed distinct chromatin state enrichments (Figure 3A-B). RNMs 6 and 7 were strongly enriched with active promoters and actively transcribed regions (Figure 3B), further supporting that the primed state is more metabolically active than the formative state¹. RNM 10 is enriched with both bivalent chromatin and repressed polycomb complexes (Figure 3B) and likely captures a primed-specific regulatory mechanism for the repression of developmental genes. These observations suggest that repressed polycomb complexes are likely activated at the transition from the formative to the primed state. Overall, these analyses show that the RNMs encompass well-known epigenomic and cell state-specific features present in formative to primed states and importantly captured the coordinated activity of the majority of regulatory elements underpinning the spectrum of pluripotency traits in hiPSCs.
Discussion Pages 11-12	Our findings highlight certain understudied regulatory networks that may be critical to hiPSC pluripotent state transitions. The formative cell state is enriched for self-renewal processes¹, compared to primed cells which can exhibit emergent features of lineage commitment²¹. Historically, epigenomic regulation in pluripotent cells has been characterized by a single regulatory network primarily mediated by NANOG, OCT4, and SOX2 binding (RNM 2)²². Our study suggests that there are several regulatory networks with distinct epigenetic profiles (chromatin state, TF binding), that mediate biological processes that are indispensable for maintaining the formative and

	primed pluripotency states (Figure 3). For example, a TEAD4-mediated regulatory network (RNM 3, Figure 3A-C) is present in the early postimplantation epiblast-like formative state and absent in the late postimplantation epiblast-like primed state. We demonstrate that trophoblast-specific peaks are enriched with RNM 3 peaks and TEAD Family binding sites (Figure 4C-D), which is consistent with previous observations that TEAD4 is a pioneering factor for the placental lineage commitment⁶³. Recently much effort has been put into developing efficient trophoblast differentiation protocols to study common pregnancy-related diseases, such as preeclampsia^{41,43,44,53}. RNM 3 may serve as a resource to identify regulatory elements that are early determinants of trophoblast cell fate commitment. Bivalent chromatin and repressed polycomb complexes are well-known pluripotency-associated epigenomic features that repress the expression of genes involved in differentiation^{14,38,64}. We show that the primed-associated RNM 10 is enriched in both polycomb repressed regions and bivalent chromatin, while the formative-associated RNM 8 is primarily enriched with bivalent chromatin, indicating that the two pluripotency states have discrete regulatory networks involving different epigenomic mechanisms for the repression of developmental genes (Figure 3A-B). These analyses revealed regulation in hiPSCs is more complex than the canonical NANOG, OCT4, and SOX2 regulatory network and that leveraging large sample sizes can resolve independent functional distinct pluripotency networks.
--	---

4. It was hard to see how the analysis and resource was achieving the biggest goal of the paper as stated in the title that this analysis was identifying Regulatory networks that define cell state transitions. There is overlap with known pluripotency cell states and with accessibility of fetal cell lines but it is unclear what the transitions are and what determines them.

We thank the reviewer for their comment about cell state transitions. We agree that this concept required further explanation, therefore we added clarifying statements in the introduction and results. Since pluripotent cell states are interconvertible, the field has defined “cell state transitions” as the molecular differences between closely related states.

We have changed the title of the paper because the definition of cell state transitions is not known outside the niche field that studies pluripotency states.

Below is a table highlighting the clarification and new Title:

Section	Edit
Introduction Page 1	“...unsupervised community detection algorithms could provide novel insights into pluripotency cell state transitions (i.e., differences between closely related states), and how genetic background contributes to variability in self-renewal and pluripotency cell state-specific network activity across cell lines.”

New Title	Analysis of regulatory network modules in hundreds of human stem cell lines reveals complex epigenetic and genetic factors contribute to pluripotency state differences between subpopulations
------------------	--

5. I found the section "Integration of hiPSC ATAC-seq peak functional annotations" to be not very useful. It can be removed or integrated with some other section.

We agree with the reviewer that the placement of this analysis diminished its usefulness. We have changed the panel to show the *SMAD1* loci (Figure 3D, 3E). *SMAD1* is an important developmental gene active in several different tissues. We feel that diagramming the data in this format aids in the interpretation of information provided the complex bioinformatic analyses and network-based approaches and insights into their utility. For these reasons, we have moved the panel to the bottom of the "Functional characterization of regulatory network modules" on **pages 8-9** and feel that it now serves a more useful message.

6. The Allele-specific SNP analysis is interesting, but the section ends with "Taken together, these data show that specific TF groups, chromatin states and the RNMs which they comprise show relative enrichment or depletion for ASCA-SNPs.". Isn't this obvious that there will be some modules with enrichment and some with depletion and some of them will have some overlap with other annotation categories?

We thank the reviewer for this insightful comment. Based on Reviewer 1, Comment 1 we have substantially changed the genetic analysis section. We believe that our study addresses a dearth of information about the role of genetic regulatory variation in stem cells because there are only a handful of labs actively investigating the topic.

The original manuscript tested variants that intersected predicted TFBSs and evaluated their effect on co-accessibility. The approach was conceptually complex and due to a lack of suitable datasets, it is challenging to test our hypotheses, therefore we decided to simplify the analysis. We no longer discuss specific variants although interested readers could find variants with strong effects by looking in the supplemental tables. In the revised analysis, we tested 105,055 SNPs in peaks in the 13 RNMs for allele-specific chromatin accessibility. We characterized the 6,323 SNPs with exhibited ASCA, by examining which RNMs they were in, and which predicted TFBSs they overlapped. We calculated differential allelic imbalance between RNMs and TF groups, independently. We observed enrichment for allelic imbalance in formative associated RNMs 2 and 3 and NANOG-OCT4 Complex binding sites. To the reviewer's point, these are concordant annotations, as RNM 2 is primarily enriched with NANOG-OCT4 Complex (Figure

3C). Despite this association between RNM 2 and NANOG-OCT4, the shared enrichment for higher allelic imbalance may suggest that the mechanism in which ASCA SNPs disrupt RNM 2 is through the disruption of NANOG-OCT4 TFBSs by SNPs that have large effects on chromatin accessibility.

Below is a table highlighting the interpretation of the RNM and TFBS association in the context of allelic imbalance enrichment:

Section	Edit
Results Characterization of allele-specific chromatin accessibility SNPs (ASCA-SNPs) Page 10	Previous studies have shown that both genetic and epigenetic variation affect hiPSC phenotypes⁵⁵⁻⁵⁷, however, the underlying mechanisms are poorly understood. One likely mechanism through which genetic variation influences hiPSC phenotypes⁵⁵⁻⁵⁷ is by affecting chromatin accessibility. We examined 105,055 SNPs (MAF ≥ 0.05, HWE p-value $> 1 \times 10^{-6}$) present in 35,614 ATAC-seq peaks in the 13 major RNMs and determined that 6,323 displayed ASCA. To determine if specific RNMs harbored ASCA SNPs with large effects, we performed Mann-Whitney U tests on the allelic imbalance fraction (AIF) of ASCA SNPs in each of the 13 RNMs using the other 12 RNMs as background (Figure 5A). We observed that four RNMs 1, 2, 3, and 9 (all enriched for enhancers Figure 3B), contained ASCA SNPs with greater allelic imbalance (AIF) compared to the other RNMs (Figure 5A). Of note, RNMs 2 and 3 were associated with the formative state (Figure 3A) and represent the NANOG-OCT4 and TEAD-mediated (Figure 3C) regulatory networks, respectively. While RNMs 1 and 9 were depleted for formative state regulatory elements, they represented NANOG-OCT4-mediated regulatory networks most likely in different pluripotent states. Our findings show that compared with the other regulatory networks active in hiPSCs, the regulatory elements in the pluripotency-associated networks harbor genetic variants that exert large effects on chromatin accessibility. Genetic variation can influence chromatin accessibility by affecting the binding specificity of TF motifs in regulatory elements⁵⁸⁻⁶¹. We examined if any specific group of TF groups were enriched for having ASCA SNPs overlapping their binding sites. Of the 6,323 ASCA SNPs, 1,933 (30.6%) overlapped at least one TFBS, while the majority did not overlap any TFBSs (n = 4,299, Figure 5B). We calculated the enrichment of the ASCA SNPs in TFBSs by performing a Fisher's Exact test for all 92 TF groups (Figure 5C). Regulatory elements predicted to bind 23 TF groups were enriched with ASCA SNPs (adjusted P-value < 0.05 and Odds Ratio > 1). These 23 groups included many pluripotency-associated TFs, such as NANOG-OCT4, TEAD Family, POU2F2, and SOX-LEF1 Complex (Figure 5C). We examined the allelic imbalance fraction (AIF) of the ASCA SNPs in each of the 23 TF groups and observed that NANOG-OCT4 contained

	ASCA SNPs with significantly higher AIF indicating that they had large effects on chromatin accessibility (Figure 5D). Six other TF groups (AR-FOXJ3, TEAD Family, CTCF Family, SALL4-ZIC1 Complex, and SOX-LEF1 Complex) exhibited increased AIF, albeit not significantly. Taken together, these results show that pluripotency-associated TF binding sites and regulatory networks are enriched with genetic variants that have large effects on chromatin accessibility. While previous studies have shown that pluripotency TFs have a remarkably high degree of evolutionary constraint⁶², our findings show that the regulatory elements to which they bind have a high amount of variability suggesting that they could play an important role in the observed pluripotent state differences between hiPSCs.
Figure 5A	Shows that RNMs 2, 3, 1, and 9 are enriched for ASCA SNPs with higher allelic imbalance.
Figure 5D	Shows that ASCA SNPs that intersect NANOG-OCT4 binding sites have higher allelic imbalance.
Discussion Page 12	Our study also addresses a dearth of information about the role of genetic regulatory variation in stem cells because there are only a handful of labs actively investigating the topic. In addition to heritable, complex diseases that have known developmental origins (i.e. autism spectrum disorder, schizophrenia), there is mounting evidence that common diseases, such as type 2 diabetes⁶⁵ and cardiovascular disease¹⁸, are influenced by regulatory variation that is active during fetal development. It is paramount to expand exploration into these areas of research to characterize developmental regulatory variation. In this study, we identify thousands of SNPs with allele-specific chromatin accessibility (ASCA). Despite the importance of maintaining pluripotency, we observed that the regulatory elements in the pluripotency-associated networks harbor genetic variants that exert large effects on chromatin accessibility. We also observed that compared with the other TF groups the binding sites for the NANOG-OCT4 complex were enriched for ASCA SNPs. Our findings suggest that variability in the regulatory elements in the pluripotency networks could play an important role in the observed varying proportions of pluripotency states between hiPSCs.

7. The paper is very descriptive and there is little or no validation (computational or experimental) of the results. It is hard to know how reliable the GNMs are and how much they agree with gene regulatory networks associated with the pluripotency state. Similarly, the RNM network could have been evaluated with Hi-C data but this is not considered. Also although a lot of analyses have been shown, it is difficult to see if there any generalizable concepts that could be used to identify key genes or interactions that could be experimentally validated. Much of

what they propose that could be tested is very specific to the pluripotency state (e.g. the ROCK inhibitor analysis).

We thank the reviewer for requesting validation of the network modules, as the implementation substantially strengthened the paper. Reviewer 2, Comment 1 also requested validation of the network modules. When we investigated this concern, we noted that there was a considerable amount of co-expression between genes in different modules. For this reason, we optimized the network discovery algorithm as follows:

Optimization

- D. First, we ran the Leiden community detection algorithm 1,700 times with different combinations of the three parameters (resolution, beta, and n_iterations) to identify the model that yields networks with the highest intramodularity and highest resolution modules consisting of biologically relevant gene membership.
- E. We permuted the nodes to confirm the Leiden algorithm was clustering co-expressed genes better than the null background (data not shown).
- F. Since there are no currently established standard modularity thresholds for biological networks due to limitations discussed in this review (Lancichinetti and Fortunato, 2011), we selected the combination that had a modularity greater than 0.4 and yielded the highest number of modules (element membership > 100 genes) and the highest fraction of elements in modules.

Validation

- E. We validated our networks and claims about cell state specificity by performing cellular deconvolution using gene signatures from FACS-sorted populations of formative and primed hiPSC subpopulations (Lau et al., 2020). We estimated the relative proportion of the formative and primed states across the 213 RNA-seq samples and 150 ATAC-seq samples. We demonstrated that GNMs and RNMs are associated with the estimated formative proportion. This observation indicates that the GNM and RNM modules are respectfully composed of co-expressed genes and co-accessible peaks that capture specific biological signals.

- F. We further validated the networks by showing that elements in the same module are more likely to be co-expressed or co-accessible compared to elements in other modules (Figure 1G, Figure 2D).
- G. The accuracy of biological networks is typically assessed by functional enrichment. We obtained five gene sets (8CLC and Naïve, epiblast, trophectoderm, primitive endoderm) from differential expression analyses (Mazid et al., 2022; Stirparo et al., 2018). We reanalyzed the differential expression data to curated gene sets for the formative and primed cell states (Lau et al., 2020). We performed a functional enrichment analysis on these seven gene set to validate our gene network modules.
- H. We also benchmarked our approach against established tools, such as weighted gene co-expression analysis (WGCNA). Since our model accounts for kinship, it outperformed WGCNA in module detection precision. Reviewer 2, Comment 3 directly requests elaboration on why our approach is more appropriate compared to other established methods. We provide a detailed description on how we benchmarked our model against WGCNA in response to that comment below.

The Reviewer 2, Comment 3 directly requests elaboration on why our approach is more appropriate compared to other established methods. We provide a detailed description on how we benchmarked our model against WGCNA in response to Reviewer 2, Comment 3.

Below is a table of additional text that highlights the utility of study:

Section	Edit
Discussion Page 12	In summary, our study suggests that epigenomic regulation of pluripotency and self-renewal processes are more complex than previously thought. It classifies hiPSC ATAC-seq peaks into 13 networks, 6 of which are associated with the formative or primed cell states, and 7 of which are likely associated with different pluripotent states. It also proposes potential mechanisms for how genetic variation influences TF binding and pluripotency regulatory mechanisms. These network classifications and ASCA SNPs could be a useful resource for researchers investigating, various aspects of pluripotency and development, such as; 1) epigenomic regulation in stem cells, 2) cell fate commitment, 3) fetal developmental processes, 4) embryonic-specific regulatory variation, 5) influence of regulatory variation on regulatory element co-accessibility, and 6) the development iPSC tissue derivation protocols.

Bibliography

- Deb-Rinker, P., Ly, D., Jezierski, A., Sikorska, M., Walker, P.R., 2005. Sequential DNA methylation of the Nanog and Oct-4 upstream regions in human NT2 cells during neuronal differentiation. *J Biol Chem* 280, 6257–6260. <https://doi.org/10.1074/jbc.C400479200>
- Kashyap, V., Rezende, N.C., Scotland, K.B., Shaffer, S.M., Persson, J.L., Gudas, L.J., Mongan, N.P., 2009. Regulation of stem cell pluripotency and differentiation involves a mutual regulatory circuit of the NANOG, OCT4, and SOX2 pluripotency transcription factors with polycomb repressive complexes and stem cell microRNAs. *Stem Cells Dev* 18, 1093–1108. <https://doi.org/10.1089/scd.2009.0113>
- Kumar, B., Navarro, C., Winblad, N., Schell, J.P., Zhao, C., Weltner, J., Baqué-Vidal, L., Salazar Mantero, A., Petropoulos, S., Lanner, F., Elsässer, S.J., 2022. Polycomb repressive complex 2 shields naïve human pluripotent cells from trophectoderm differentiation. *Nat Cell Biol* 24, 845–857. <https://doi.org/10.1038/s41556-022-00916-w>
- Lancichinetti, A., Fortunato, S., 2011. Limits of modularity maximization in community detection. *Phys Rev E Stat Nonlin Soft Matter Phys* 84, 066122. <https://doi.org/10.1103/PhysRevE.84.066122>
- Langfelder, P., Horvath, S., 2008. WGCNA: an R package for weighted correlation network analysis. *BMC Bioinformatics* 9, 559. <https://doi.org/10.1186/1471-2105-9-559>
- Lau, K.X., Mason, E.A., Kie, J., De Souza, D.P., Kloehn, J., Tull, D., McConville, M.J., Keniry, A., Beck, T., Blewitt, M.E., Ritchie, M.E., Naik, S.H., Zalcenstein, D., Korn, O., Su, S., Romero, I.G., Spruce, C., Baker, C.L., McGarr, T.C., Wells, C.A., Pera, M.F., 2020. Unique properties of a subset of human pluripotent stem cells with high capacity for self-renewal. *Nat Commun* 11, 2420. <https://doi.org/10.1038/s41467-020-16214-8>
- Li, M., Belmonte, J.C.I., 2017. Ground rules of the pluripotency gene regulatory network. *Nat Rev Genet* 18, 180–191. <https://doi.org/10.1038/nrg.2016.156>
- Mazid, M.A., Ward, C., Luo, Z., Liu, C., Li, Y., Lai, Y., Wu, L., Li, J., Jia, W., Jiang, Y., Liu, H., Fu, L., Yang, Y., Ibañez, D.P., Lai, J., Wei, X., An, J., Guo, P., Yuan, Y., Deng, Q., Wang, Y., Liu, Y., Gao, F., Wang, J., Zaman, S., Qin, B., Wu, G., Maxwell, P.H., Xu, X., Liu, L., Li, W., Esteban, M.A., 2022. Rolling back human pluripotent stem cells to an eight-cell embryo-like stage. *Nature* 605, 315–324. <https://doi.org/10.1038/s41586-022-04625-0>
- Pliner, H.A., Packer, J.S., McFaline-Figueroa, J.L., Cusanovich, D.A., Daza, R.M., Aghamirzaie, D., Srivatsan, S., Qiu, X., Jackson, D., Minkina, A., Adey, A.C., Steemers, F.J., Shendure, J., Trapnell, C., 2018. Cicero Predicts cis-Regulatory DNA Interactions from Single-Cell Chromatin Accessibility Data. *Mol Cell* 71, 858-871.e8. <https://doi.org/10.1016/j.molcel.2018.06.044>
- Stirparo, G.G., Boroviak, T., Guo, G., Nichols, J., Smith, A., Bertone, P., 2018. Integrated analysis of single-cell embryo data yields a unified transcriptome signature for the human pre-implantation epiblast. *Development* 145, dev158501. <https://doi.org/10.1242/dev.158501>

REVIEWER COMMENTS

Reviewer #1 (Remarks to the Author):

The authors have substantially re-worked the manuscript and this version is much improved. Most of my previous concerns have been addressed very well. But I have two remaining comments:

1) In the revised manuscript, the authors applied retrospective deconvolution to estimate the fraction of formative state cells in the original cell populations. These estimates are used to infer associations between specific GRNs and RNMs with different pluripotent states. Deconvolution based on transcriptomic data uses the CIBERSOFT method, which was well-validated in the original publication. One optional suggestion is for the authors to use the RNA-seq data from the Lau et al publication to benchmark their ability to estimate formative state fractions. Specifically, I am wondering if Dr. Pera might know the fraction of formative cells in the two “General population” samples from the Lau et al study, as presumably these samples were analysed at the time by flow cytometry for EPCAM-GCTM2-CD9 in order to generate the matched EPCAM-GCTM2-CD9 high samples. If that fraction is known, then the authors could run their CIBERSORT analysis on these “General populations”, and hopefully retrieve an estimate that matches the measured proportion. The same comment goes for the deconvolution approach using ATAC-seq data, but here I think the validation is more important because as far as I know, CIBERSOFT has not previously be applied to ATAC-seq data in this way (?). If that is the case, then I think it is important for the authors to demonstrate that estimating pluripotent state fraction using ATAC-seq data is legitimate and accurate. (Such validations were an important aspect of the original CIBERSOFT study, based on transcriptome data). Hopefully the authors might have access to existing ATAC-seq data where the proportion of formative cells in the samples are known (e.g. from the Lau study). But if not, then I think the authors should consider including new ATAC-seq profiling of a small number of samples with known proportions of formative cells. These results would validate their new approach and give confidence that the inferred connections between RNMs and formative state are solid.

2) Page 12: In the paragraph on the role of genetic regulatory variation in stem cells, the authors write: “Our findings suggest that variability in the regulatory elements in the pluripotency networks could play an important role in the observed varying proportions of pluripotency states between hiPSCs.” But if this was true, i.e. if genetic variability in the regulatory elements is a primary determinant of pluripotent state populations, then you would expect to see broad agreement between the proportion of formative cells in the same iPSC line as measured by RNA-seq and by ATAC-seq. But as far as I can tell from the Supplementary tables, that does not seem to be the case. Does the lack of concordance instead suggest that genetic variability in regulatory elements does not play an important role in determining formative state abundance within a population of hiPSCs?

Reviewer #2 (Remarks to the Author):

The authors have addressed all my concerns.

Reviewer #3 (Remarks to the Author):

The authors have done a reasonable job of addressing my comments.

We thank the reviewer for their diligence in assessing the revision of our manuscript. As highlighted below, we performed additional analyses to address their concerns. We have made edits to the manuscript to address the reviewer's comments which are highlighted in yellow. We also made minor edits to improve grammar and the flow of the manuscript which are not highlighted.

1) In the revised manuscript, the authors applied retrospective deconvolution to estimate the fraction of formative state cells in the original cell populations. These estimates are used to infer associations between specific GRNs and RNMs with different pluripotent states. Deconvolution based on transcriptomic data uses the CIBERSOFT method, which was well-validated in the original publication. One optional suggestion is for the authors to use the RNA-seq data from the Lau et al publication to benchmark their ability to estimate formative state fractions. Specifically, I am wondering if Dr. Pera might know the fraction of formative cells in the two "General population" samples from the Lau et al study, as presumably these samples were analysed at the time by flow cytometry for EPCAM-GCTM2-CD9 in order to generate the matched EPCAM-GCTM2-CD9 high samples. If that fraction is known, then the authors could run their CIBERSOFT analysis on these "General populations", and hopefully retrieve an estimate that matches the measured proportion. The same comment goes for the deconvolution approach using ATAC-seq data, but here I think the validation is more important because as far as I know, CIBERSOFT has not previously be applied to ATAC-seq data in this way (?). If that is the case, then I think it is important for the authors to demonstrate that estimating pluripotent state fraction using ATAC-seq data is legitimate and accurate. (Such validations were an important aspect of the original CIBERSOFT study, based on transcriptome data). Hopefully the authors might have access to existing ATAC-seq data where the proportion of formative cells in the samples are known (e.g. from the Lau study). But if not, then I think the authors should consider including new ATAC-seq profiling of a small number of samples with known proportions of formative cells. These results would validate their new approach and give confidence that the inferred connections between RNMs and formative state are solid.

We understand that the reviewer has concerns about the accuracy of the cell state estimates based on deconvolution using CIBERSOFT. In the first round of revisions, we performed cellular deconvolution on the bulk RNA and ATAC-seq for the hiPSCs to address a different reviewer's request to validate our claims that the GNMs and RNMs are associated with pluripotency cell states. In the Lau et al paper, a double high (GCTM2^{high}-CD9^{high}) population was obtained by harvesting the top 25% of cells with these markers, then the EPCAM^{high}-GCTM2^{high}-CD9^{high} population was harvested by collecting the top 10% subset of GCTM2^{high}-CD9^{high} cells with the highest EPCAM expression. The GCTM-2^{mid}-CD9^{mid} population was obtained by gating on mid-level expression of GCTM-2 and CD9, as indicated in Figure 1a in Lau et al. According to Dr. Pera, these populations are not collected in a manner that enables experimental quantification of the exact formative and primed fractions. For the RNA-seq deconvolution, we utilized the FACS sorted populations of the EPCAM^{high}-GCTM2^{high}-CD9^{high} and the unsorted "General population" to respectively represent the Formative and Primed cell states. For the ATAC-seq deconvolution, we used the FACS sorted GCTM-2^{high}CD9^{high}EPCAM^{high} and the GCTM-2^{mid}-CD9^{mid} to respectively represent the Formative and Primed cell states.

CIBERSOFT has been previously used to deconvolute bulk ATAC-seq data in published papers. Based on the reviewer's comment we realized that we did not properly cite these previous studies (Qu et al, 2017; Corces et al 2016). We have added references to these studies in our manuscript.

We acknowledge that the deconvoluted cell states are rough estimates, and more of a method to rank the samples by cell state proportion rather than quantifying exact formative and primed fractions.

We believe that the analyses in the manuscript conducted downstream of the CIBERSORT deconvolution validate that cell state differences across samples are accurately captured. Below are a few examples supporting our claim.

- 1) In Figure 1B and Figure S2B-C, we demonstrate that formative and primed-associated genes in used in the signature matrix are differentially expressed in the 15 samples with an estimated formative proportion over 80% and the 69 samples that had a 100% primed estimate. The observation that dozens of genes share the same gene expression patterns in these two sample groups, indicates that they are likely co-expressed and underlie cell state differences.
- 2) Figure 1H-I shows that the GNMs 5 and 10, which were defined in an unsupervised manner, are strongly correlated with the formative and primed estimates, respectively. In Figure 1J, we show that GNM 5 is strongly enriched with differentially expressed genes in the formative state, the epiblast and the naïve state. In Figure 1K, we demonstrate that GNM 5 is composed of several genes that are prominently highlighted as being characteristic of the formative cell state in Lau et al.
- 3) Figure 2F-H show that RNMs 2, 3 and 8 are associated the formative cellular estimates and Figure 2I shows that they are also associated with formative-associated GNM 5. This observation exhibits concordance between the RNA and ATAC-seq formative estimates.
- 4) Figure 3A-C shows that the formative-associated RNMs 2, 3, and 8 are enriched with the formative-associated peaks from Lau et al. Additionally, RNMs 2 and 3 are enriched at enhancers and bound by TEAD Family TFs and NANOG-OCT4, which are features of the formative state described by Lau et al.

Here, we performed two additional analyses to address the reviewer's concerns about the accuracy of the cell state estimates.

- 1) To determine if the CIBERSORT RNA-seq deconvolution conducted in our manuscript was optimal, we generated signature matrices that consisted of 100, 200, and 300 differentially expressed genes. CIBERSORT calculates the correlation between the expected values and the observed values for each sample which can be interpreted as a metric for the model's performance. We observed that the 300 gene signature matrix yielded the strongest mean sample correlation ($r = 0.821$; see histogram below), therefore we re-performed the downstream analyses with these new estimates and updated Figure 1B-E, H-I and Figure S2. We had omitted the CIBERSORT sample correlations from the supplemental material in the previous submission of this manuscript but have added them to this re-submission in Supplemental Table S2 (for RNA-seq deconvolutions) and Table S6 (for ATAC-seq deconvolution). We have also updated Table S3 to reflect the optimized 300-gene signature matrix.

2) To further validate the deconvolution estimates, we performed an additional deconvolution analysis using scRNA-seq data from 90 GCTM2^{low}-CD9^{low} cells, 90 GCTM2^{mid}-CD9^{mid} cells, 90 GCTM2^{high}-CD9^{high} cells, and 100 EPCAM^{high}-GCTM2^{high}-CD9^{high} cells from Lau et al (GSE119323). Pseudobulk RNA-seq was generated by summing the gene counts from cells from each population. The four aggregated samples were TMM normalized with the edgeR package. The same 300 gene signature matrix used to deconvolute the 213 iPSC RNA-seq samples was applied to the pseudobulk for the four cell populations. The double low and double mid populations were estimated to have no cells in the formative states, while the double high population exhibited a low fraction (5.5%), and the triple high population was composed of >25% cells in the formative state (table below). These findings further support that the iPSC lines are being appropriately ranked by the deconvolution estimates based on their relative cell state compositions.

Population	Formative Estimate	
	<fct>	<dbl>
GCTM2low-CD9low		0.00000000
GCTM2mid-CD9mid		0.00000000
GCTM2high-CD9high		0.05476091
EPCAMhigh-GCTM2high-CD9high		0.25863701

We have added clarifying edits to reiterate that the deconvolution estimates appropriately rank the iPSCs based on their relative cell state compositions, but not meant to be interpreted as precise measurements of relative cell state proportions.

2) Page 12: In the paragraph on the role of genetic regulatory variation in stem cells, the authors write: “Our findings suggest that variability in the regulatory elements in the pluripotency networks could play an important role in the observed varying proportions of pluripotency states between hiPSCs.” But if this was true, i.e. if genetic variability in the regulatory elements is a primary determinant of pluripotent state populations, then you would expect to see broad agreement between the proportion of formative cells in the same iPSC line as measured by RNA-seq and by ATAC-seq. But as far as I can tell from the Supplementary tables, that does not seem to be the case. Does the lack of concordance instead suggest that genetic variability in regulatory elements does not play an important role in determining formative state abundance within a population of hiPSCs?

We appreciate the reviewer’s astute observation. It is well known that environmental factors contribute to incomplete penetrance in polygenic traits. We believe that the lack of concordance between the formative estimates in the RNA and ATAC-seq data from the same iPSC lines arises from the different culture conditions under which the molecular samples were collected. The ATAC-seq samples were collected from iPSC lines cultured with ROCK inhibitor. In the

initial version of the manuscript, we demonstrated that ROCK inhibitor exerted large phenotypic effects on iPSC gene expression. It is likely that such large phenotypic effects make weak genetic effects undetectable and drive the majority of the differences between RNA and ATAC formative estimates. We removed this analysis during the first round of revisions because it only included three paired samples and we felt that it was underpowered to make a strong claim. Polygenic traits are influenced by many regulatory variants that exert small effects; thus, they are often masked by environmental factors.

Here, to address the Reviewer's concern about genetic variation playing a role in determining formative state abundance across iPSC lines cultured under similar conditions, we performed an additional analysis to demonstrate that formative-associated GNM 5 Pareto gene expression is more similar in iPSCs from monozygotic twins, than unrelated individuals. Briefly, we measured the pairwise correlation of the 213 samples based on the inverse normalized transformed TMM-expression of the 298 GNM 5 Pareto genes using the cor function in base R. We focused on 91 pairwise comparisons where individuals were members of a monozygotic twin pair. In total, there were 7 pairs of monozygotic twins and 84 pairs between unrelated individuals. In the plot below, we observe that monozygotic twins have more similar fractions of the formative associated GNM 5 compared to unrelated individuals ($P\text{-value} = 5.3 \times 10^{-5}$), supporting that genetic variants play a role in the relative cell state composition in iPSC lines.

REVIEWERS' COMMENTS

Reviewer #1 (Remarks to the Author):

The authors have nicely addressed the remaining comments that I had. This is a strong and complete article.